# Towards Steering without Sacrifice: Principled Training of Steering Vectors for Prompt-only Interventions

**Yuntai Bao** [1]   **Qinfeng Li** [1]   **Xinyan Yu** [1]   **Ge Su** [1 2]   **Wenqi Zhang** [2]   **Liu Yan** [3]   **Haiqin Weng** [3]   **Jianwei Yin** [1]
**Xuhong Zhang** [✉ 2]

## Abstract

Recently, *steering vectors (SVs)* have emerged as an effective and lightweight approach to steer behaviors of large language models (LLMs), among which fine-tuned SVs are more effective than optimization-free ones. However, current approaches to fine-tuned SVs suffer from two limitations. First, they require careful selection of steering factors on a per-SV basis to balance steering effectiveness and generation quality at inference time. Second, they operate as *full-sequence SVs (FSSVs)*, which can sacrifice generation quality regardless of factor selection due to excessive intervention on the model generation process. To address the first limitation, we propose *joint training* of steering factors and directions, such that post-hoc factor selection is no longer required. Using neural network scaling theory, we find that moderately large initialization sizes and learning rates for steering factors are essential for stability and efficiency of joint training. To tackle the second limitation, we draw inspiration from *representation fine-tuning* and introduce **Prompt-Only Steering Vector (PrOSV)**, an SV that intervenes only on a few prompt tokens. Our empirical results show that PrOSV outperforms traditional FSSVs on AXBENCH when using our joint training scheme. We also find that PrOSV achieves a better tradeoff between general model utility and adversarial robustness than FSSV.

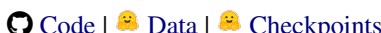

 Code | 🤗 Data | 🤗 Checkpoints

---

[1]Zhejiang University [2]Innovation and Management Center, School of Software Technology (Ningbo), Zhejiang University [3]Ant Group. ✉ Corresponding author. Correspondence to: Xuhong Zhang <zhangxuhong@zju.edu.cn>. Contact: Yuntai Bao <yuntaibao@zju.edu.cn>.

*Proceedings of the 43rd International Conference on Machine Learning*, Seoul, South Korea. PMLR 306, 2026. Copyright 2026 by the author(s).

## 1. Introduction

As large language models (LLMs) grow in capabilities and complexities (Google DeepMind, 2025), they also pose growing challenges in reliability and control (Bai et al., 2022; Anthropic, 2025; Schoen et al., 2025). Prompting and fine-tuning are common techniques to control LLMs, but both have limitations: prompting is versatile but also brittle and labor-intensive (Chang et al., 2024), while fine-tuning is powerful but expensive, producing uninterpretable, non-sparse changes to model weights (Wehner et al., 2025). An emergent alternative is the *steering vector (SV)* approach, which steers model behaviors by adding a fixed vector to representations (Subramani et al., 2022; Turner et al., 2023; Wu et al., 2025a). SVs are not only interpretable and reversible compared to fine-tuning, but also more efficient and robust than prompting.

However, the practical utility of SVs is bottlenecked by a lack of principled training and application protocols. We contend that for SVs to serve as a viable engineering tool, they should satisfy **two desiderata** beyond steering effectiveness: minimal hyperparameter tuning to ensure cross-concept scalability as well as minimal impact on model utility (Wehner et al., 2025).

Despite the remarkable progress in effectiveness (Wu et al., 2025b), current methods struggle to meet these criteria simultaneously. First, current SVs rely on post-hoc selection of steering factors for each SV instance to balance intervention strength and generation quality (Algorithm 3); however, the *factor selection* process is brittle since SVs are sensitive to variations in steering factors (Wu et al., 2025a). Second, traditional SVs are typically *full-sequence SVs (FSSVs)* and intervene on both prompts and generated tokens, which could severely degrade model capabilities even with careful factor selection (von Rütte et al., 2024; Braun et al., 2024).

Our work bridges these gaps by transitioning the fine-tuned SV from a heuristic-heavy experimental tool to a more theoretically grounded approach. On one hand, to reduce hyperparameter tuning costs, we argue that steering factors should be trained together with steering directions in an end-to-end fashion, instead of being regarded as an external constant.

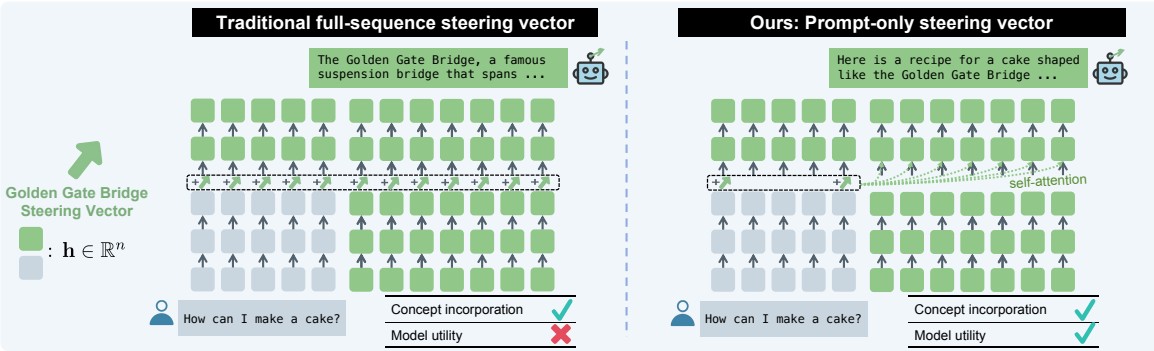

*Figure 1.* Traditional FSSV versus PrOSV. FSSV could compromise general model utility even after careful factor selection. In contrast, PrOSV achieves effective concept-based steering without sacrificing model utility.

Therefore, the **main technical challenge** we address is to enable *joint training* in a principled manner. We follow the approach of Hayou et al. (2024b;a) and Li et al. (2025) who study the impact of hyperparameters on *low-rank adaptation (LoRA)* (Hu et al., 2021) fine-tuning dynamics based on scaling theory of neural networks in the infinite-width limit. We utilize the same theoretical tool in the setting of SV training and derive the optimal learning rates and initialization strategies for steering factors and directions. With our joint training scheme, hyperparameter tuning is one-off before training and no longer required at inference time. On the other, we take inspiration from *representation fine-tuning (ReFT)* (Wu et al., 2024b) and introduce **Prompt-Only SV (PrOSV)**, which intervenes only at prefill stage and not at decode stage while achieving effective steering.

In summary, our work makes two **contributions** that challenge prevailing practices of SVs: (1) We show that inference-time factor selection is unnecessary when steering factors and directions are jointly trained with appropriate initialization schemes and learning rates; (2) We make the counterintuitive observation that PrOSV can outperform FSSVs on concept-based steering when trained properly. We also find that, PrOSV better handles the tension between preservation of general model utility and robustness to concept suppression attacks than FSSV, and it can be robust to extended contexts (~1K tokens) on Qwen2.5-32B.

## 2. Related Work

**Representation steering.** *Representation steering* encompasses methods that control model behaviors by intervening on representations at inference time (Wehner et al., 2025; Zou et al., 2023; Wu et al., 2024a;b). PrOSV is inspired by ReFT (Wu et al., 2024b), which shows that low-rank prompt-only interventions enable effective task adaptation. While ReFT focuses on fine-tuning, we extend this prompt-only design to the domain of SVs for concept-based steering.

**Steering vectors.** Prior work shows that adding a fixed vector to representations could enable effective model con-

trol (Subramani et al., 2022). This approach is termed the *steering vector (SV)*, and is one of the most lightweight forms of representation steering. Based on how SVs are obtained, we categorize them into three types: optimization-free SVs, sparse autoencoder (SAE) (Sharkey et al., 2022; Huben et al., 2024) SVs and fine-tuned SVs.

*Optimization-free SVs*, represented by *difference-in-means (DiffMean)*, are extracted from representations using contrastive inputs (Turner et al., 2023; Marks & Tegmark, 2023; Rimsky et al., 2024). While intuitive, they often lack the effectiveness of optimization-based methods (Wu et al., 2025a). *SAE SVs* emerge from the unsupervised decomposition of representations into tens of thousands of features with SAE (Templeton et al., 2024; Lieberum et al., 2024). However, a post-hoc selection process is required to identify relevant steering directions (Arad et al., 2025) and the SAE might not contain the desired concepts (Leask et al., 2025). *Fine-tuned SVs* are obtained by optimizing SVs to minimize objective functions (Wu et al., 2025a), and prior work has introduced preference optimization objectives to improve steering effectiveness (Cao et al., 2024; Wu et al., 2025b). Our work belongs in this category and addresses the training and inference protocols as well as intervention locations.

**Deriving optimal fine-tuning parameterization with scaling theory.** Scaling theory allows us to predict how training dynamics change as models grow, ensuring that learning remains stable regardless of model width (Yang et al., 2022). By deriving optimal initialization and learning rates, prior work has transformed the trial-and-error process of LoRA fine-tuning into a principled discipline (Hayou et al., 2024b;a; Li et al., 2025). Since fine-tuning SVs follows a similar optimization path on frozen models, this theoretical framework provides a rigorous foundation for selecting hyperparameters that guarantee both efficiency and stability.

## 3. Preliminaries and Task Formulation

**Language models (LMs).** In this paper, we focus on transformer LMs (Vaswani et al., 2017) denoted by $p(\cdot)$. Re-

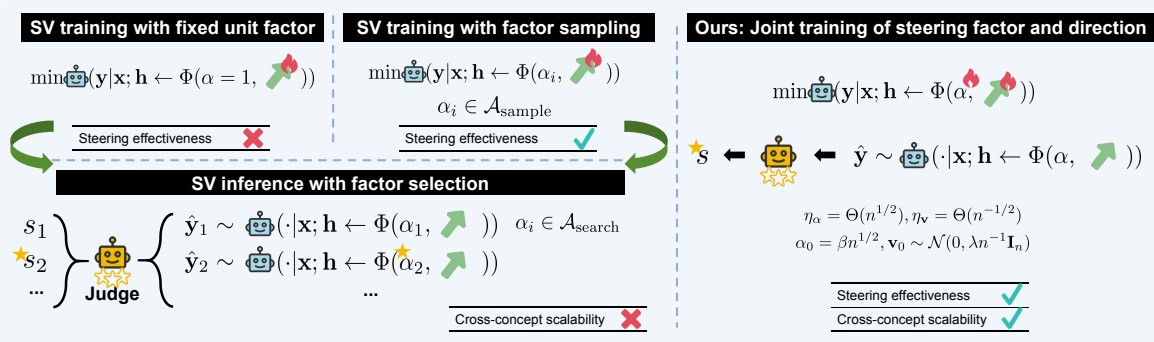

*Figure 2.* Comparison of SV training and inference strategies. Traditional fine-tuned FSSVs always require post-hoc factor selection, while our joint training scheme enables end-to-end cross-concept scalability and is compatible with both FSSV and PrOSV.

sponses are sampled from its output distribution in an autoregressive manner: $\hat{\mathbf{y}} \sim p(\cdot|\mathbf{x})$, where $\mathbf{x}$ is an input prompt. Let model width be $n$ and output residual stream representation of the $l$-th layer be $\mathbf{h}_l \in \mathbb{R}^n$. We denote representations of the $i$-th token at the $l$-th layer by $\mathbf{h} := \mathbf{h}_l^{(i)}$ for simplicity.

**Interventions.** According to Wu et al. (2024b; 2025a), an *intervention* is defined as a function $\Phi : \mathbb{R}^n \to \mathbb{R}^n$ that edits representations in-place during the forward pass: $\mathbf{h} \leftarrow \Phi(\mathbf{h})$. The model output distribution under intervention is written as $p_\Phi(\cdot|\mathbf{x}; \mathbf{h} \leftarrow \Phi(\mathbf{h}))$.

**Steering vectors (SVs).** An SV is an intervention parameterized by a vector $\mathbf{v} \in \mathbb{R}^n$, termed the *steering direction*. Following prevalent settings of previous work on SVs, we apply SV interventions to a single layer of a model. In this paper, we focus on the simplest intervention functional form, *addition intervention (AddInv)*. It adds a scaled vector to representations: $\Phi^{\mathrm{Add}}(\mathbf{h}; \alpha, \mathbf{v}) = \mathbf{h} + \alpha\mathbf{v}$, where $\alpha \in \mathbb{R}$ is the *steering factor*.

**Task setup.** Our final objective is to achieve *concept-based steering*. Following the setup of Wu et al. (2025a), the goal of concept-based steering is to incorporate a concept $c$ into response using intervention $\Phi$: $\hat{\mathbf{y}}^c \sim p_\Phi(\cdot|\mathbf{x}; \mathbf{h} \leftarrow \Phi)$. An example concept is the *Golden Gate Bridge concept*, i.e. "descriptions of or references to the Golden Gate Bridge" (Templeton et al., 2024). The steered response ($\hat{\mathbf{y}}^c$) should express the concept $c$ while fulfilling instruction $\mathbf{x}$; correspondingly, two common failure modes are failure of concept incorporation and ignoring instruction content.

**Current scheme for SV training and inference.** *Response-only language modeling (Lang.)* is a common objective for training SVs (see §F.4) (Subramani et al., 2022). As for *trainable parameters*, previous works view the steering direction ($\mathbf{v}$) as the only trainable parameter while treating the steering factor ($\alpha$) as an external constant. Regarding the *SV training process*, as is shown in Figure 2, earlier works use the unit factor ($\alpha = 1$) during training (Subramani et al., 2022); however it fails to yield effective SVs as is shown in

Table 2 and Wu et al. (2025a). Therefore Wu et al. (2025b) adopt a *factor sampling trick* (Algorithm 2) to improve steering performance and to decrease variance. At *inference* time, optimal factors are selected via grid search (*factor selection*; Algorithm 3). This process is expensive since it requires sampling hundreds of intervened responses for each SV and factors vary across instances of SVs (Rimsky et al., 2024; Wu et al., 2025a).

## 4. Training Dynamics of SVs with Adam

In this section, we study how to stably and efficiently train SVs when jointly training the steering direction $\mathbf{v}$ and steering factor $\alpha$. **Our goal** is to answer a practical question: *How should we choose initialization and learning rates so that both $\alpha$ and $\mathbf{v}$ learn effectively without destabilizing the model?* To answer this, we analyze the effect of learning rate and initialization strategy on the training dynamics of SVs in the infinite-width limit when using the Adam optimizer (Kingma & Ba, 2014). Our analysis is grounded in the general framework of scaling theory of neural networks (Yang & Littwin, 2023; Yang et al., 2022). We follow the notations, settings as well as definitions of stability and efficiency from prior work (Hayou et al., 2024b;a; Li et al., 2025), and we only utilize the same theoretical tools for SV training on a frozen pretrained neural network.

### 4.1. $\gamma$-operator notation

We study SV training dynamics in the infinite-width limit. Following prior work, we assume that only model width $n$ increases with all other aspects such as model depth and number of training steps held fixed. LLMs nowadays have large widths, e.g., $n \approx 2K$ for models with $\geq 2B$ parameters (Gemma Team, 2024). Therefore it makes sense to study training dynamics when the width goes to infinity.

The $\gamma$-*operator notation* describes how coordinate size of a vector $\mathbf{v} \in \mathbb{R}^n$ scales with network width $n$ as $n$ approaches infinity, where coordinates of $\mathbf{v}$ are asymptotically

*independent and identically distributed (IID)* (Hayou et al., 2024b). It is defined as $\mathbf{v} = \Theta(n^{\gamma[\mathbf{v}]})$, and it describes the *typical* coordinate size in the second moment: $\|\mathbf{v}\|_2^2/n = \Theta(n^{2\gamma[\mathbf{v}]}), n \to \infty$. For real-valued variables $u, v$, two basic computational rules apply: $\gamma[u+v] = \max(\gamma[u], \gamma[v])$ $(u + v \neq 0 \text{ as } n \to \infty)$ and $\gamma[uv] = \gamma[u] + \gamma[v]$. We refer readers to §E for more details.

## 4.2. Simplified settings and the optimizer

To simplify analysis, we adopt the same setup as Hayou et al. (2024b;a) and Li et al. (2025). Specifically, we assume each mini-batch has one data point $(\mathbf{x}, \mathbf{y})$ (which can be trivially extended to larger mini-batches), and the goal of SV training is to minimize the objective function $\ell(\cdot)$.

In this paper we focus on the Adam optimizer for its wide usage. We use a convenient assumption from prior work that gradients $(g)$ are processed by the Adam optimizer such that $\gamma[g] = 0$. This assumption is generally satisfied due to the entry-wise gradient normalization of the Adam optimizer (Yang & Littwin, 2023; Hayou et al., 2024b).

## 4.3. Stability and Efficiency of SVs

We now analyze the conditions that ensure stability and efficiency of SV training. We denote $\mathbf{z} = \Phi(\mathbf{h}) - \mathbf{h}$ as *SV features*, i.e. the contribution of an SV to representations, mirroring the definition of *LoRA features* (Hayou et al., 2024a). Then we have $\mathbf{z} = \alpha\mathbf{v}$ for AddInv. By default, we use subscript $t$ to denote the value of variables at the $t$-th training step $(t = 0, 1, \dots)$.

The notion of *stability* requires that SV features remain within reasonable range (i.e. neither explodes nor diminishes) as $n$ approaches infinity (Hayou et al., 2024b):

**Definition 4.1** (Stability). An SV training process is considered stable if, for all training steps $t \geq 1$, i.e. we have $\mathbf{h}, \mathbf{z}_t = \Theta(1)$ as model width $n \to \infty$.

Following Hayou et al. (2024b;a), we assume that the pre-training parameterization of the LLM already ensures stability, such that $\mathbf{h} = \Theta(1)$. Based on Definition 4.1, we obtain the following for AddInv:

$$\gamma[\mathbf{z}_t] = \gamma[\alpha_t\mathbf{v}_t] = \gamma[\alpha_t] + \gamma[\mathbf{v}_t] = 0. \quad (1)$$

Besides stability, we require that $\mathbf{z}_t$ should be sufficiently updated during SV training. The update to SV feature at the $t$-th step, $\Delta\mathbf{z}_t = \mathbf{z}_t - \mathbf{z}_{t-1}$, is expanded as follows:

$$\Delta\mathbf{z}_t = \underbrace{(\Delta\alpha_t)\mathbf{v}_{t-1}}_{\delta_t^1} + \underbrace{\alpha_{t-1}(\Delta\mathbf{v}_t)}_{\delta_t^2} + \underbrace{(\Delta\alpha_t)(\Delta\mathbf{v}_t)}_{\delta_t^3}, \quad (2)$$

where $\Delta\alpha_t = \alpha_t - \alpha_{t-1}, \Delta\mathbf{v}_t = \mathbf{v}_t - \mathbf{v}_{t-1}$, $\delta_t^1$ is the update obtained by fixing $\mathbf{v}$ and training only $\alpha$, $\delta_t^2$ is obtained by fixing $\alpha$ and training only $\mathbf{v}$, and $\delta_t^3$ describes

the composite update of both $\alpha$ and $\mathbf{v}$. Ideally, we should ensure $\Delta\mathbf{z}_t = \Theta(1)$ as $n \to \infty$ so that total SV feature updates are bounded and non-trivial. In the meantime, we enforce the same properties for individual parameter updates, so that both $\alpha$ and $\mathbf{v}$ are actively learned rather than one parameter dominating the total update. This leads to the following definition, which is also called the *feature learning* regime (Yang & Hu, 2020; Hayou et al., 2024b;a).

**Definition 4.2** (Efficiency). An SV training process is considered efficient if it is stable (Definition 4.1) and, for all $t \geq 1$, the additive components of $\Delta\mathbf{z}_t$ are all $\Theta(1)$. For AddInv, we require $\delta_t^i = \Theta(1), i = 1, 2, 3$.

Since update rule is gradient descent, the updates to $\alpha$ and $\mathbf{v}$ at the $t$-th step are $\Delta\alpha_t = -\eta_\alpha g_{t-1}^\alpha$ and $\Delta\mathbf{v}_t = -\eta_\mathbf{v} g_{t-1}^\mathbf{v}$, respectively, where $\eta_\alpha, \eta_\mathbf{v}$ are learning rates and $g_{t-1}^\alpha, g_{t-1}^\mathbf{v}$ are gradients processed by Adam optimizer. We thus obtain the values of $\alpha$ and $\mathbf{v}$ at the $t$-th step:

$$\alpha_t = \alpha_0 - \eta_\alpha \sum_{i=0}^{t-1} g_i^\alpha, \mathbf{v}_t = \mathbf{v}_0 - \eta_\mathbf{v} \sum_{i=0}^{t-1} g_i^\mathbf{v}. \quad (3)$$

Taking these into stability and efficiency requirements for AddInv (Definitions 4.1 and 4.2), we obtain the following:

$$\begin{cases} \gamma\left[\delta_t^1\right] = \gamma\left[-\eta_\alpha g_{t-1}^\alpha \mathbf{v}_{t-1}\right] = 0, \\ \gamma\left[\delta_t^2\right] = \gamma\left[-\eta_\mathbf{v} g_{t-1}^\mathbf{v} \alpha_{t-1}\right] = 0, \\ \gamma\left[\delta_t^3\right] = \gamma\left[(\Delta\alpha_t)(\Delta\mathbf{v}_t)\right] = 0, \\ \gamma\left[\mathbf{z}_t\right] = \gamma\left[\alpha_t\mathbf{v}_t\right] = 0, \end{cases} \quad (4)$$

which can be simplified into the following equations:

$$\begin{cases} \gamma[\eta_\alpha] + \max(\gamma[\mathbf{v}_0], \gamma[\eta_\mathbf{v}]) = 0, \\ \gamma[\eta_\mathbf{v}] + \max(\gamma[\alpha_0], \gamma[\eta_\alpha]) = 0, \\ \gamma[\eta_\mathbf{v}] + \gamma[\eta_\alpha] = 0, \\ \max(\gamma[\alpha_0], \gamma[\eta_\alpha]) + \max(\gamma[\mathbf{v}_0], \gamma[\eta_\mathbf{v}]) = 0. \end{cases} \quad (5)$$

The solution to the equations above is:

$$\begin{cases} \gamma[\eta_\mathbf{v}] + \gamma[\eta_\alpha] = 0, \\ \gamma[\mathbf{v}_0] \leq \gamma[\eta_\mathbf{v}], \gamma[\alpha_0] \leq \gamma[\eta_\alpha]. \end{cases} \quad (6)$$

This solution indicates that the learning rates of steering factors and directions should be their respective reciprocals to achieve stability and efficiency, i.e. $\eta_\mathbf{v}\eta_\alpha = \Theta(1)$. If one parameter learns faster, the other must learn proportionally slower. Meanwhile, initialization sizes of factors and directions should not exceed the scales of their respective learning rates.

## 4.4. Practical Suggestions on SV Training

Based on our theoretical results above, we provide actionable suggestions on improving the stability and efficiency of the training process for fine-tuned SVs with AddInv.

The solution of Equation (6) does not specify the precise scaling rules. To make choice of hyperparameters feasible, we start with taking all inequalities of Equation (6) as equalities and using Kaiming initialization (He et al., 2016) for steering directions due to its wide usage, with variance $\sigma_{\mathbf{v}}^2 = \lambda n^{-1}$ where the constant $\lambda$ is *direction initialization size*. According to Lemma E.4, $\sigma_{\mathbf{v}}^2 = \Theta(n^{-1})$ means that $\mathbf{v}_0$ has $\Theta(n^{-1/2})$-sized entries. We thus have $\alpha_0 = \Theta(n^{1/2})$ and initialize steering factors with $\alpha_0 = \beta n^{1/2}$ where $\beta$ is *factor initialization size*. Both $\beta$ and $\lambda$ are tuned via grid search. As for *learning rate*, we have $\eta_{\mathbf{v}} = \Theta(n^{-1/2}), \eta_\alpha = \Theta(n^{1/2})$ and tune learning rates via grid search.

Notably, our analysis above highlights the role of factorized parameterization in making SV training tractable. Without factorization, one needs to directly optimize the SV feature $\mathbf{z}$. In that case, the stability/efficiency requirements would reduce to $\mathbf{z}_0, \eta_{\mathbf{z}} = \Theta(1)$, which offers little practical guidance for hyperparameter selection. By decomposing the SV feature into steering factor and direction, we obtain nontrivial scaling rules formulated as polynomials with respect to $n$. This enables principled and informed choices of initialization and learning rates.

**Key advantage.** Although both initialization sizes and learning rates require tuning, the tuning process is one-off before training for a certain layer of a model, thus the cost is amortized across all future SVs. In contrast, traditional SVs require selection of steering factors for each instance of SV.

**Bridge to algorithm.** Based on these scaling laws, we present a joint training procedure (Algorithm 1) that directly implements these principles.

---

**Algorithm 1** Our scheme for joint training of SV factors and directions.

---

**input** Training set $\mathcal{D}$, direction learning rate $\eta_{\mathbf{v}} = \Theta(-\frac{1}{2})$, factor learning rate $\eta_\alpha = \Theta(\frac{1}{2})$, direction initialization size $\lambda$, factor initialization size $\beta$, training steps $T$, loss function $\ell(\cdot)$

**output** Steering factor $\alpha_T$, steering direction $\mathbf{v}_T$

$\quad \mathbf{v}_0 \sim \mathcal{N}(\mathbf{0}, \lambda n^{-1}\mathbf{I}_n)$ {To ensure: $\mathbf{v}_0 = \Theta(-\frac{1}{2})$}

$\quad \alpha_0 \leftarrow \beta n^{1/2}$ {To ensure: $\alpha_0 = \Theta(\frac{1}{2})$}

$\quad t \leftarrow 0$

$\quad$**while** $t < T$ **do**

$\quad\quad (\mathbf{x}, \mathbf{y}) \sim \mathcal{D}$

$\quad\quad l_t \leftarrow \ell\left(p_\Phi\left(\cdot | \mathbf{x}; \mathbf{h} \leftarrow \Phi(\mathbf{h}; \alpha_t, \mathbf{v}_t)\right), \mathbf{y}\right)$

$\quad\quad \{g_t^{\mathbf{v}}, g_t^\alpha\} \leftarrow \text{Adam}(\{\nabla_{\mathbf{v}} l_t, \nabla_\alpha l_t\})$

$\quad\quad \mathbf{v}_{t+1} \leftarrow \mathbf{v}_t - \eta_{\mathbf{v}} g_t^{\mathbf{v}}$

$\quad\quad \alpha_{t+1} \leftarrow \alpha_t - \eta_\alpha g_t^\alpha$

$\quad\quad t \leftarrow t + 1$

$\quad$**end while**

---

## 5. Prompt-Only Steering Vector

In this section, we introduce **Prompt-Only SV (PrOSV)** as an attempt to challenge traditional fine-tuned SVs for concept-based steering, with key design choices on intervention location, source of steering factors and training scheme.

**Intervention location.** As is shown in Figure 1, PrOSV intervenes only at prefill stage and *not* decode stage. Similar to ReFT, PrOSV functions mainly by implicitly editing the KV cache. Mechanistically, PrOSV minimizes disruption to attention patterns and better preserves model capabilities than FSSV (§N.1). By restricting intervention to a constant number of prompt tokens, PrOSV achieves a $37\times$ reduction in computational overhead compared to FSSV (§G.1).

Following Wu et al. (2024b), we apply interventions to prompt prefixes and suffixes; more general intervention location strategies are left for future work. Let a prompt be $\mathbf{x} \in \mathcal{V}^m$, then interventions are applied to $p$ prefix tokens and $s$ suffix tokens ($p, s \in \{0, 1, \ldots, m\}$). The intervention locations are thus: $\mathcal{I} = \{1, 2, \ldots, p\} \cup \{m-s+1, \ldots, m-1, m\}$. In what follows, we write $p1+s2$ to denote $p = 1, s = 2$.

**Steering factor and optimization scheme.** Unlike traditional SVs that rely on post-hoc factor selection, PrOSV employs our joint training scheme of Algorithm 1, such that the trained steering factors are directly used for inference.

We emphasize that joint training is essential for effectiveness and scalability. FSSVs (including optimization-free SVs like DiffMean) *cannot* be trivially converted into PrOSVs for two reasons: (1) FSSVs work via distinct mechanisms from PrOSVs and have different optimal directions (§N.2), and (2) factor selection is required during the attempt.

## 6. Experiments

In this section, we verify the practical usefulness of our joint training scheme (§6.1) and analyze how intervention locations of PrOSV affect steering performance (§6.2), We also evaluate PrOSV on AxBench (Wu et al., 2025a), a large-scale concept-based steering benchmark (§6.3). Finally, we investigate how PrOSV affects general model utility and robustness to extended contexts and adversarial attacks (§6.4).

### 6.1. Verification of Theoretical Results

We verify whether our joint training algorithm of Algorithm 1 could guide SV training in practice.

**Data.** We conduct experiments on the CONCEPT10 dataset of the AxBench evaluation framework. CONCEPT10 consists of 10 concepts for each subset, where a subset tests steering methods applied at a certain layer of a model. The training data for a concept $c$ is $\mathcal{D}^c = \{(\mathbf{x}_i, \mathbf{y}_i^c)\}_{i=1}^N$, where

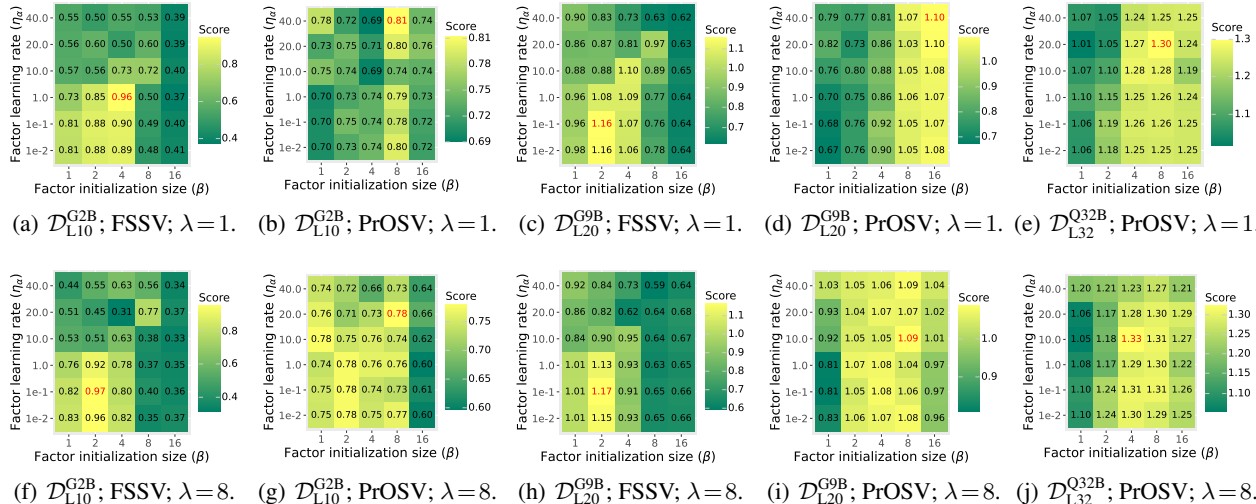

*Figure 3.* Visualization of *concept scores* using our joint training scheme. Highest scores are highlighted in red. Entries with relatively high scores all have moderately high factor initialization sizes and factor learning rates, and optimal performance is only attained when both are selected properly.

$N = 72$, $\mathbf{x}_i$ is a concept-neutral instruction and $\mathbf{y}_i$ is a steered response generated by gpt-4o-mini (OpenAI, 2024).

**Models.** We use instruction-following models: Gemma2-2B, Gemma2-9B (Gemma Team, 2024) and Qwen2.5-32B (Qwen Team, 2024). Intervention layers are the 10th layer of Gemma2-2B ($\mathcal{D}_{L10}^{G2B}$), 20th layer of Gemma2-9B ($\mathcal{D}_{L20}^{G9B}$) and 32nd layer of Qwen2.5-32B ($\mathcal{D}_{L32}^{Q32B}$). We select these layers since SVs usually perform best at middle layers (Wu et al., 2025a; Sun et al., 2025a).

**Metrics.** We report *concept scores* (0–2) of AXBENCH (Wu et al., 2025a), i.e. how well the concept is incorporated into a response. This is because AXBENCH training data models only the concepts, not the tradeoff between concept incorporation and instruction fulfillment. For each concept, we sample SV-intervened responses using 10 random instructions from AlpacaEval (Li et al., 2023). Concept scores are evaluated with an LLM judge (gpt-4o-mini; OpenAI (2024)) and averaged across three random seeds.

**Hyperparameters.** We focus primarily on tuning hyperparameters for steering factors since steering factors are not trained in prior work. We vary learning rates of factors ($\eta_\alpha$) and initialization sizes of factors and directions ($\beta, \lambda$). Meanwhile we fix training steps, batch size and direction learning rates ($\eta_\mathbf{v} = 0.04$) (details in §J). For PrOSV, we use $p4+s4$ on Gemma2-2B/9B and $p2+s2$ on Qwen2.5-32B.

**Results.** Results are shown in Figure 3 (more results in §J.2), from which we make the following observations. (1) SVs are highly sensitive to hyperparameters. (2) FSSV and PrOSV achieve highest concept scores with $\beta > 1$ and $\eta_\alpha > \eta_\mathbf{v}$, which highlights the importance of using larger initialization sizes and learning rates for steering factors.

(3) A larger direction initialization size does not improve FSSVs, but it increases the concept scores of PrOSV across the majority of search grid ((b) vs. (g), (d) vs. (i) and (e) vs. (j)). (4) PrOSV is robust to factor learning rates at certain factor initialization sizes, for example, $\beta = 8$ on $\mathcal{D}_{L10}^{G2B}$ with $\lambda = 1$, $\beta = 8$ on $\mathcal{D}_{L20}^{G9B}$ with $\lambda = 8$ and $\beta = 8$ on $\mathcal{D}_{L32}^{Q32B}$ with $\lambda = 1$. Similar phenomena can be seen for FSSV, but FSSV has a smaller robustness range.

> **Takeaway.**
> - Choices of hyperparameters crucially determine the training performance of SVs;
> - Moderately high learning rates and initialization sizes for steering factors are necessary for SVs to achieve optimal steering performance.

## 6.2. Effect of PrOSV Intervention Location

In this experiment, we investigate how PrOSV intervention location affects the performance of SVs trained with our joint training scheme.

**Data.** We use CONCEPT10, with average prompt lengths of 21 on $\mathcal{D}_{L10}^{G2B}$, 17 on $\mathcal{D}_{L20}^{G9B}$ and 13 on $\mathcal{D}_{L32}^{Q32B}$.

**Models.** We use Gemma2-2B/9B and Qwen2.5-32B.

**Metrics.** We report the best overall score and the best concept scores, each selected independently from the search grid. *Overall score* (0–2), the main metric of AXBENCH, is the harmonic mean of *concept score*, *instruct score* (how well a response is related to the instruction) and *fluency score* (how fluent a response is) (Wu et al., 2025a). Results are averaged across three random seeds.

*Table 1.* Highest grid-selected overall (O) and concept (C) scores on CONCEPT10. Best results are highlighted in bold.

| Location | G2B; L10 O | G2B; L10 C | G9B; L20 O | G9B; L20 C | Q32B; L32 O | Q32B; L32 C |
|---|---|---|---|---|---|---|
| FSSV | 0.65 | 0.97 | 0.86 | 1.17 | 0.93 | 1.27 |
| Full prompt | 0.54 | **1.12** | 0.71 | **1.41** | 0.88 | **1.58** |
| $|\mathcal{I}| = 2$ | | | | | | |
| $p2$ | 0.65 | 0.69 | 0.78 | 0.90 | 0.91 | 1.00 |
| $s2$ | 0.68 | 0.82 | 0.71 | 0.94 | 1.11 | 1.28 |
| $p1+s1$ | 0.67 | 0.79 | 0.91 | 1.12 | 1.10 | 1.24 |
| $|\mathcal{I}| = 4$ | | | | | | |
| $p4$ | 0.59 | 0.63 | 0.73 | 0.85 | 0.94 | 0.97 |
| $s4$ | 0.69 | 0.83 | 0.77 | 1.03 | 1.08 | 1.24 |
| $p2+s2$ | **0.70** | 0.82 | **0.92** | 1.14 | **1.16** | 1.33 |
| $|\mathcal{I}| = 8$ | | | | | | |
| $p8$ | 0.58 | 0.64 | 0.75 | 0.85 | 0.90 | 0.94 |
| $s8$ | 0.61 | 0.85 | 0.74 | 1.12 | 0.92 | 1.24 |
| $p4+s4$ | 0.69 | 0.85 | 0.89 | 1.09 | 1.13 | 1.30 |

*Table 2.* Overall steering scores (0–2; ↑) on AXBENCH. * results are taken from Wu et al. (2025a), † from Wu et al. (2025b) and ‡ from Arad et al. (2025). Best results are highlighted in bold.

| Method | $\mathcal{D}^{G2B}_{L10}$ | $\mathcal{D}^{G9B}_{L20}$ | $\mathcal{D}^{Q32B}_{L32}$ |
|---|---|---|---|
| Prompt | 0.698* | 1.075* | 1.060 |
| *Objective: Lang.* | | | |
| FSSV | 0.663† | 0.788† | 0.798 |
| + Fixed unit factor* | 0.072 | 0.024 | — |
| + Joint training | 0.736 | 0.821 | 0.919 |
| PrOSV | 0.758 | 0.859 | 1.049 |
| *Objective: SimPO* | | | |
| FSSV (RePS) | 0.756† | 0.892† | 0.947 |
| + Joint training | 0.769 | 0.886 | 0.982 |
| PrOSV | **0.803** | **0.905** | **1.102** |
| LoReFT* | 0.701 | 0.777 | — |
| DiffMean* | 0.297 | 0.322 | — |
| SAE‡ | — | 0.546 | — |

**Hyperparameters.** We use the setup of §6.1 but fix $\lambda = 8$ since it yields higher overall scores for PrOSV while having little impact for FSSV (see §J.2). For PrOSV, we vary both the intervention budget and location. Besides *full-prompt intervention* with dynamic budgets, we test interventions with fixed budgets: $|\mathcal{I}| = 2, 4, 8$, where the largest fixed budget is around half the average prompt length. For PrOSVs with fixed budgets, we use three variants: *prefix-only* (e.g., $p4$), *suffix-only* (e.g., $s4$) and *prefix-suffix* (e.g., $p2+s2$).

**Results.** Results are shown in Table 1, from which we make the following observations. (1) Full-prompt interventions achieve highest concept scores but lowest overall scores, meaning they successfully incorporate target concepts at the cost of generation quality. (2) PrOSV generally achieves lower concept scores than FSSV, which is expected; however, PrOSV could obtain comparable concept scores to FSSV on Gemma2-9B and Qwen2.5-32B, and sometimes yields higher overall scores than FSSV. These results indicate that concept incorporation does not require full-sequence interventions. (3) Prefix-suffix interventions generally achieve higher overall scores than prefix-only and suffix-only interventions with the same budgets, and $p2+s2$ strikes the best tradeoff between concept incorporation and generation quality; (4) Steering performance does not always scale with computational budget, since $p2+s2$ yields higher overall scores than both $p1+s1$ and $p4+s4$. (5) Results are consistent on Gemma2-2B, Gemma2-9B and Qwen2.5-32B, indicating that our findings above are likely transferrable across model families and model scales.

> **Takeaway.**
> - PrOSV achieves a better tradeoff between concept incorporation and generation quality than FSSV;
> - Four tokens of intervention is sufficient for concept-

based steering with PrOSV, and $p2+s2$ is a model-agnostic choice that achieves the best tradeoff within our selected search space.

### 6.3. Evaluating Concept-based Steering at Scale

In this experiment, we study whether our joint training scheme helps SVs attain better steering performance than factor sampling with factor selection and how PrOSV compares to FSSV on the concept-based steering task. To this end, we use AXBENCH, a large-scale benchmark for evaluating model control methods (Wu et al., 2025a).

**Data.** For Gemma2-2B/9B, we train SVs on the CONCEPT500 dataset from AXBENCH with 500 concepts. Since preference optimization objectives require contrastive training examples, we follow Wu et al. (2025b) and augment the training set by generating concept-neutral responses ($\mathbf{y}_i$) with gpt-4o-mini and obtain $\mathcal{D}^{c+} = \{(\mathbf{x}_i, \mathbf{y}_i, \mathbf{y}_i^c)\}_{i=1}^{N}$. Due to limited computing resources, we only test on 100 concepts for Qwen2.5-32B. Details are shown in §L.1.

**Metrics.** We report average overall scores across concepts; standard deviation is reported in §L.2.

**Models.** We test on three setups: $\mathcal{D}^{G2B}_{L10}$, $\mathcal{D}^{G9B}_{L20}$ and $\mathcal{D}^{Q32B}_{L32}$.

**Methods.** We report results on three types of model control methods: prompting, fine-tuning (rank-4 *low-rank ReFT, LoReFT*; Wu et al. (2024b)) and SVs. Among the SV baselines, we include DiffMean, SAE and three fine-tuned SVs: *reference-free preference steering (RePS)* (Wu et al., 2025b) with *simple preference optimization (SimPO)* objective (Meng et al., 2024) and two SVs trained with Lang. objective, one trained with fixed unit factor (Wu et al., 2025a) and another trained with factor sampling (Wu et al., 2025b). All SV baselines are FSSVs and require inference-time fac-

*Table 3.* Accuracy (%; ↑) on tinyMMLU (M) and tinyGSM8K (G), as well as *overall steering score on tinyGSM8K* (O; ↑). *Vanilla* denotes un-steered model performance. Best steered results are highlighted in bold.

| | G2B; L10 | | | G9B; L20 | | | Q32B; L32 | | |
|---|---|---|---|---|---|---|---|---|---|
| Method | M | G | O | M | G | O | M | G | O |
| Vanilla | 54.0 | 79.0 | — | 74.0 | 93.0 | — | 74.0 | 97.0 | — |
| Prompt | **53.7** | **61.0** | **1.03** | 62.1 | 88.6 | 1.33 | 63.8 | 93.4 | 1.47 |
| Objective: Lang. | | | | | | | | | |
| FSSV | 41.5 | 10.7 | 0.66 | 54.2 | 8.6 | 0.64 | 41.1 | 6.6 | 0.75 |
| PrOSV | 52.9 | 50.5 | 0.36 | 55.4 | 68.4 | 0.49 | 58.4 | 78.2 | 1.04 |
| Objective: SimPO | | | | | | | | | |
| FSSV | 37.8 | 5.6 | 0.70 | 41.3 | 4.2 | 0.75 | 39.2 | 6.9 | 0.90 |
| PrOSV | 51.3 | 50.3 | 0.39 | 56.2 | 66.8 | 0.58 | 59.2 | 79.2 | 1.08 |

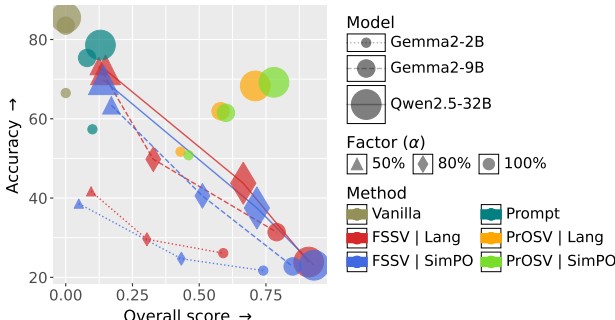

*Figure 4.* Benchmark accuracy (%; average of tinyMMLU and tinyGSM8K) vs. *overall score under concept suppression attack*. PrOSV resides on a better Pareto frontier than FSSV.

tor selection. We show formulas of baselines in §F.

As for our methods, in order to investigate the effect of our joint training scheme and PrOSV as well as how they work with various training objectives, we integrate them with Lang. and SimPO objectives and obtain four variants.

**Hyperparameters.** For fair comparison, we tune hyperparameters via grid search using a development set of three concepts (details in §L.1).

**Results.** Results are shown in Table 2, from which we have the following findings. (1) Our joint training scheme generally improves performance of SVs over those trained with unit factor or factor sampling and steered with selected factors (we discuss the exception on $\mathcal{D}_{L20}^{G9B}$ in §L.2). (2) PrOSV consistently outperforms FSSV. (3) SVs trained with Lang. objective underperform those trained with SimPO, which confirms the finding from prior work (Wu et al., 2025b) that training objectives crucially impact steering performance. (4) Steering performance of prompting saturates on Gemma2-9B, while SVs are able to benefit from increased model scale or capabilities.

> **Takeaway.**
> - Our joint training scheme enables FSSV to outperform the factor sampling-then-selection pipeline;
> - PrOSV outperforms FSSV and achieves superior performance when trained with SimPO objective.

### 6.4. Tradeoff between Performance and Robustness

Prior work has noted that SVs might harm general model utility (Arditi et al., 2024; Wehner et al., 2025). We therefore evaluate how PrOSV affects model performance on popular benchmarks. We also stress-test its generalization abilities in terms of robustness to adversarial attacks and extended contexts, since it only intervenes on a few prompt tokens.

**Capability benchmarks.** Limited by computational resources, we test on tinyMMLU and tinyGSM8K (Polo et al.,

2024), which are concise versions of MMLU (Hendrycks et al., 2021), an aggregated multiple-choice benchmark for multi-task language understanding, and GSM8K (Cobbe et al., 2021), a popular arithmetic reasoning benchmark, respectively. These tiny benchmarks are reported to be representative of their full versions (Polo et al., 2024). To evaluate how SVs generalize to *extended* contexts, we also report overall steering scores on tinyGSM8K, where the average prompt length is around 1K tokens. We show details in §M.1.

**Adversarial attacks.** We use concept-suppressing prompts as an attack, where we prompt gpt-4o-mini to incorporate concept-suppressing requirements into AlpacaEval test instructions (details in §M.1). The metric is overall score.

**Methods.** We include prompting as a baseline and use the same steering prompts as in §6.3. Meanwhile we evaluate PrOSVs and FSSVs that use joint training, with checkpoints from previous AXBENCH evaluation (§6.3). For FSSVs, we additionally decrease factors to 80% and 50% to study how inference-time factor adjustment affect the results. Results are averaged across 50 concepts.

**Results.** We show results in Table 3 and Figure 4 (details in §M.2). According to Table 3, prompting best preserves benchmark performance; however, it still harms model performance, which indicates an inherent tension between concept-based steering and benchmark tasks. This tension has a tendency to be relieved in larger/more capable models, as is indicated by the decreasing performance gap from Gemma2-2B, Gemma2-9B to Qwen2.5-32B. We find that intervention location has a great impact on model performance, especially on the arithmetic reasoning task. On tinyGSM8K, FSSVs reduce accuracy by 68–90% while PrOSVs induce smaller declines of 18–29%. Meanwhile training objectives have different levels of impact on model performance for FSSV and PrOSV: for FSSVs, SimPO generally has a noticeable negative impact on benchmark performance compared to Lang.; however there is no significant difference between objectives for PrOSVs.

As for steering in extended contexts (Table 3), there is hardly an inherent tradeoff between steering and benchmark accuracy, since prompting always yields highest overall scores on tinyGSM8K. It is expected that PrOSV is less robust to extended contexts than FSSV on Gemma2-2B/9B since it intervenes only on four tokens; however, it outperforms FSSV on Qwen2.5-32B. The reason is that PrOSV benefits from increased model scale/capabilities in terms of concept incorporation while achieving good generation quality, whereas FSSV consistently degrades generation quality (Table 17).

Based on Figure 4, there is a tradeoff between adversarial robustness and model utility for FSSVs and inference-time factor adjustment does not address this tradeoff; whereas PrOSV achieves a better balance. In terms of adversarial robustness, prompting is the weakest baseline while SVs are better at overriding attacks, among which FSSVs with 100% factors and SimPO objective are most robust.

> **Takeaway.**
> - PrOSV achieves a better tradeoff between model utility and robustness than FSSV;
> - PrOSV could be more robust to extended context than FSSV on Qwen2.5-32B, but not on Gemma2-2B/9B.

## 7. Conclusion and Discussions

In this paper, we present two novel insights that challenge traditional SVs: (1) Learning rates and initialization sizes of steering factors and directions crucially impact SV training dynamics, and we use neural network scaling theory to derive principled ways of setting these hyperparameters; (2) Prompt-only intervention on as few as four tokens is sufficient for effective concept-based steering. We integrate PrOSV with SimPO objective and obtain state-of-the-art performance on AXBENCH. We also find that, although PrOSV better tackles the tradeoff between preservation of model utility and adversarial robustness than FSSV, it is robust to extended contexts only on large/capable models.

**Limitations and future work.** First, we focus on fine-tuned SVs in this work and do not discuss optimization-free SVs, the latter of which can be seen as *pretrained SVs* that emerge during the pretraining process. Future work could study principled strategies to obtain steering factors and directions for optimization-free SVs. Second, we find in §6.3 that training objectives crucially determine steering performance. Therefore, future work could design training objectives that push the steering performance to a new frontier. Third, since we focus on canvassing prefix/suffix intervention locations for PrOSV, we acknowledge that more general choices of intervention locations could further advance steering performance and leave them for future work. Finally, we point out a tradeoff between intervened model capability and ad-

versarial robustness in §6.4. Neither PrOSV nor FSSV is able to fully reconcile this conflict; future work could thus explore further advancing the tradeoff.

## Acknowledgements

This work was supported by the Key R&D Program of Ningbo under Grant No.2024Z115. This work was also supported by Ant Group.

We thank the anonymous reviewers for their useful suggestions. We also thank Zhaopeng Feng for the constructive comments.

## Impact Statement

This paper presents work whose goal is to advance the field of machine learning and, in particular, methods for controlling the behavior of LLMs. Our work has a range of potential societal impacts and might induce dual-use implications. On the positive side, PrOSV enables parameter-efficient control of model behavior, which may facilitate improved alignment and safer deployment of LLMs. At the same time, the same capabilities could be misused to steer models toward harmful, deceptive, or biased behaviors.

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

## Appendix Table of Contents

*Table 4.* Summary of notations.

| Symbol | Meaning |
|---:|---|
| $n \in \mathbb{R}$ | Model width, also the dimension of the residual stream. |
| $\mathbf{h} \in \mathbb{R}^n$ | Representations; residual stream by default. |
| $\mathcal{V}$ | Model vocabulary. |
| $\mathbf{x} \in \{\mathcal{V}, \mathcal{V}^2, \dots\}$ | Input prompt. |
| $\mathbf{y} \in \{\mathcal{V}, \mathcal{V}^2, \dots\}$ | Response. |
| $c$ | Concept. |
| $\mathbf{y}^c$ | Steered response incorporated with concept $c$. |
| $p(\cdot|\mathbf{x})$ | Output distribution of a model conditioned by input $\mathbf{x}$. |
| $\Phi : \mathbb{R}^n \to \mathbb{R}^n$ | Intervention function. |
| $\mathbf{h} \leftarrow \Phi(\mathbf{h})$ | Intervention notation. |
| $p(\cdot|\mathbf{x}; \mathbf{h} \leftarrow \Phi(\mathbf{h}))$ | Intervened output distribution of a model conditioned by input $\mathbf{x}$. |
| $\Phi^{\text{Add}}(\cdot)$ | Addition intervention (AddInv). |
| $\alpha \in \mathbb{R}$ | Steering factor. |
| $\beta \in \mathbb{R}$ | Steering factor initialization size. |
| $\lambda \in \mathbb{R}$ | Steering direction initialization size. |
| $\mathbf{v} \in \mathbb{R}^n$ | Steering direction. |
| $\mathbf{u} \in \mathbb{R}^n$ | Normalized steering direction; $\mathbf{u} := \mathbf{v}/\|\mathbf{v}\|_2$. |
| $\mathcal{N}(\boldsymbol{\mu}, \sigma^2 \mathbf{I}_n)$ | Multivariate Gaussian distribution with mean $\boldsymbol{\mu} \in \mathbb{R}^n$ and variance $\sigma^2$. |
| $\Theta(\cdot)$ | Asymptotic $\Theta$ notation. |
| $\gamma[\cdot]$ | $\gamma$-operator; $v = \Theta(n^{\gamma[v]})$ as $n \to \infty$. |
| $t$ | Time step. |
| $T$ | Number of total training steps. |
| $\eta$ | Learning rate. |
| $\ell(\cdot)$ | Loss function. |
| $l_t$ | Loss value at $t$-th timestep. |
| $\nabla l_t$ | Raw gradients. |
| $g$ | Gradients processed by Adam optimizer. |
| $p$ | Number of prompt prefix tokens for prompt-only interventions. |
| $s$ | Number of prompt suffix tokens for prompt-only interventions. |
| $pt_1 + st_2$ | Prompt-only intervention with $t_1$ prefix tokens and $t_2$ suffix tokens. |
| $N$ | Size of training dataset. |
| $\mathcal{D}^c$ | Training dataset for concept $c$; $\mathcal{D}^c = \{(\mathbf{x}_i, \mathbf{y}_i^c)\}_{i=1}^N$. |
| $\mathcal{D}^{c+}$ | Contrastive dataset for concept $c$; $\mathcal{D}^{c+} = \{(\mathbf{x}_i, \mathbf{y}_i, \mathbf{y}_i^c)\}_{i=1}^N$. |
| $\mathcal{D}^{\text{G2B}}_{\text{L10}}$ | CONCEPT500 subset for the 10th layer of Gemma2-2B. |
| $\mathcal{D}^{\text{G9B}}_{\text{L20}}$ | CONCEPT500 subset for the 20th layer of Gemma2-9B. |
| $\mathcal{D}^{\text{Q32B}}_{\text{L32}}$ | CONCEPT500 subset for the 32nd layer of Qwen2.5-32B. |
| $\mathcal{A}_{\text{sample}}$ | Factor sampling set for SV training with factor sampling. |
| $\mathcal{A}_{\text{search}}$ | Factor search grid for factor selection. |

# A. Reproducibility

We provide our proof-of-concept *code* at `https://anonymous.4open.science/r/prosv_icml2026`; the full code, including a Python library implementation of PrOSV, our joint training scheme as well as experiment pipelines, would be released upon acceptance. We will also open-source our augmented CONCEPT500 *dataset* and *checkpoints* of SVs trained on AXBENCH to facilitate future work in this field.

# B. Usage of Generative AI

In this paper, we use LLMs for the following purposes:

1. To assist in writing, e.g. grammar checking and refinement;

2. For data curation, which is elaborated in §6.1, §6.2, §6.3, §L and §M;

3. As LLM judge, which is explained in §6.1, §6.2, §L and §M.

# C. Background

This section is motivated by the *historical notes on steering* section of Wu et al. (2025a) as well as the *background* sections of Hayou et al. (2024a) and Li et al. (2025). It is meant to introduce the general background on representation steering and scaling theory of neural networks for readers unfamiliar with either field.

## C.1. Representation Steering

Here we aim to provide unfamiliar readers with the necessary background knowledge on the field of representation steering positioned as a pragmatic branch of *mechanistic interpretability* in the setting of transformer language models.

In general, representation steering is the technique of editing the internal representations of neural networks. The editing operation is termed *intervention*. The simplest instantiation of an intervention is to add a fixed vector to representations, which is known as the *steering vector (SV)* approach. In this paper, we call the vector the *steering direction* and the scaling coefficient the *steering factor*. Representation steering research is often motivated by the fundamental assumption that neural networks encode high-level concepts in low-dimensional linear subspaces of representations, even though neural networks are often nonlinear. This assumption is also termed the *linear representation hypothesis* (Sharkey et al., 2025).

We will now provide an informal review of **how steering directions are obtained**. Subramani et al. (2022) were among the first to conceptualize the notion of SVs in the field of NLP; they fine-tune SVs by maximizing the probabilities of target responses in order to accomplish the task of controllable text generation. Recently, this approach is used to optimize SVs for controlling safety-relevant behaviors of LLMs using a single training example (Dunefsky & Cohan, 2025).

However, one of the most commonly used SV is the optimization-free approach, *difference-in-means (DiffMean)* (Marks & Tegmark, 2023), which is also termed *activation addition (ActAdd)* by Turner et al. (2023) or *contrastive activation addition (CAA)* by Rimsky et al. (2024) (§F). DiffMean steering direction is obtained by subtracting the mean of representations of negative inputs (i.e. without concept $c$) from those of positive inputs (i.e. incorporated with concept $c$).

In addition to optimized SVs and optimization-free SVs, it has been found that decoder features learnt by *sparse autoencoders (SAEs)* (Sharkey et al., 2022) can also be used to control model behaviors in accordance with feature descriptions (Huben et al., 2024; Templeton et al., 2024). One important caveat is that features should be carefully selected to ensure steering performance (Arad et al., 2025): output-oriented features are preferred for steering over input-oriented features.

Regarding the **types of concepts**, in this paper, we follow of the series of research that study *concepts of contents* (Wu et al., 2025a; Sun et al., 2025b; Arad et al., 2025; Wu et al., 2025b) according to the taxonomy of Wehner et al. (2025). By "concepts of contents", Wehner et al. (2025) refer to concepts that describe properties of model-generated responses. Thus the goal of steering in this context is to control the model to focus on a specific topic and contain certain contents. These contents can either be exact text like the "mentions of the day 'Tuesday' " or higher-level topics like "terms related to biochemical compounds and their effects".

Content concepts are in contrast to concepts related to abstract, high-level behavioral characteristics like harmfulness and honesty (Zou et al., 2023), character traits like sycophancy (Rimsky et al., 2024; Chen et al., 2025a), values and goals (Cao et al., 2024) as well as reasoning structure of large reasoning models (Chen et al., 2025b). Most of these works use the

optimization-free SV, DiffMean, or its variants and use full-sequence interventions. Although we do not study these concepts in this paper, PrOSV is applicable to the scenarios above since high-level concepts are easier to control than content concepts. This is supported by current experiment evidence that it is easy for DiffMean to steer high-level concepts (Rimsky et al., 2024; Chen et al., 2025a) but not concent concepts (Wu et al., 2025a).

### C.2. Scaling Theory of Neural Networks

In the main body, we primarily use scaling theory as a useful theoretical tool; in this subsection, we aim to familiarize readers with the general background on the scaling theory of neural networks.

Scaling refers to the process of increasing the size of a neural network component, e.g., model width, model depth, training/inference-time compute. In this paper, we focus solely on scaling model width ($n$) given that current LLMs typically have large model width.

Historically, initialization strategies have been designed to avoid numerical instabilities and to ensure efficient learning in the setting of large model widths. For example, He et al. (2016) derive that the initialization variance of model weights should be $2/n$ to avoid vanishing/exploding gradients, which is now commonly termed *Kaiming initialization*.

Yang & Hu (2020); Yang et al. (2022) introduce *maximal update parametrization (µP)* to allow for maximal feature learning in **pretraining** of neural networks. *Stability* is defined as $\mathbf{h}_l = \Theta(1)$ where $\mathbf{h}_l$ is the output feature of the $l$-th layer, whereas *feature learning* is defined as $\Delta\mathbf{h}_l = \Theta(1)$ where $\Delta\mathbf{h}_l$ is the feature update at a training step. Both properties are essential to ensuring that the trained model features neither explode nor diminish. µP constructs scaling rules for initialization schemes, learning rates and network architectures to guarantee maximal feature learning while ensuring stability in the infinite-width limit. Recently, Mlodozeniec et al. (2025) introduce *Complete$^{(d)}$P*, which builds upon *CompleteP* of Dey et al. (2025) and enables hyperparameter transfer across modules, model width, model depth, batch size and training duration.

In the similar vein, Hayou et al. (2024b), Hayou et al. (2024a) and Li et al. (2025) study scaling rules of initialization schemes and learning rates for LoRA to achieve both stability and feature learning in the setting of LLM **fine-tuning**. Let LoRA parameterization be $\Delta\mathbf{W} \approx \mathbf{BA}$, where $d_{\text{in}}, d_{\text{out}}$ are input and output dimensions, respectively, $\mathbf{W} \in \mathbb{R}^{n_{\text{out}} \times n_{\text{in}}}$ is a weight matrix, $\mathbf{B} \in \mathbb{R}^{n_{\text{out}} \times r}, \mathbf{A} \in \mathbb{R}^{r \times n_{\text{in}}}$ are LoRA parameters and $r$ is low-rank dimension. Hayou et al. (2024b) find that it facilitates convergence by setting a higher learning rate for $\mathbf{B}$, such that $\eta_{\mathbf{B}} > \eta_{\mathbf{A}}$. Hayou et al. (2024a) show that under the traditional paradigm of zero initialization ($\mathbf{B}_0\mathbf{A}_0 = \mathbf{0}$), the default initialization of $\mathbf{A}_0 \sim \mathcal{N}(0, 1/n), \mathbf{B}_0 = \mathbf{0}$ (Init[A]) facilitates feature learning but risks internal feature instability, whereas initializing $\mathbf{A}_0 = \mathbf{0}, \mathbf{B}_0 \sim \mathcal{N}(0, 1/r)$ (Init[B]) ensures stability but at the cost of suboptimal feature learning. Li et al. (2025) challenge the zero initialization paradigm and show that non-zero initialization not only improves the robustness of LoRA fine-tuning to suboptimal learning rates but also improves performance on the fine-tuning task.

**Representation steering** with trainable parameters, including fine-tuned SVs, is similar to LoRA since both are optimized on the basis of a frozen pretrained model. Therefore representation steering methods are susceptible to analysis with scaling theory under similar theoretical settings as analysis of previous work on LoRA fine-tuning.

## D. Additional Related Work

**Prompt-only model control.** A number of works have explored controlling model generation behaviors via operations only at the prefill stage. *Prefix tuning* (Li & Liang, 2021) achieves parameter-efficient model control by prepending learnable virtual tokens to the input, and is thus closely related to our prompt-only intervention design. Similarly, *prompt tuning* (Lester et al., 2021) prepends learnable soft embeddings to input prompts for task adaptation. However, prior analyses (He et al., 2022) show that prefix tuning primarily operates by reweighting attention, effectively downweighting original attention outputs while upweighting prefix-induced signals. This mechanism differs from SVs, which directly modify output representations of transformer blocks. Such reweighting can be suboptimal for concept-based steering, as (1) intervening at attention outputs is generally less effective than modifying residual stream representations, and (2) suppressing original model activations may degrade model utility. For these reasons, and following prior SV work, we do not include prefix tuning as a baseline.

*Task vector* (Hendel et al., 2023) and *function vector* (Todd et al., 2023) are also relevant, where vectors are obtained from in-context learning contexts and intervene on the last token of prompts. Since we focus on training-based methods, we do not build directly upon function vectors and task vectors but study $s1$ interventions as part of our location search process.

Our work also connects with *ZeroTuning* (Han et al., 2026), which adds biases to the attention logits of the initial token to improve model performance. This technique roughly relates to our $p1$ intervention location setup. It is an interesting and lightweight method, but it requires modifying attention module while SV does not, and extensive factor selection makes it less practical.

# E. Details on Training Dynamics of Steering Vectors with Adam

## E.1. Preliminaries on Asymptotic Notations

The following definition from Yang & Hu (2020) is helpful for understanding how random variables scale asymptotically as model width $n \to \infty$.

**Definition E.1** (Coordinate size of a vector). For any vector $V \in \mathbb{R}^n$, we say $V$ has $\Theta(n^\alpha)$-sized coordinates if $\|V\|_2^2/n = \Theta(n^{2\alpha})$ as $n \to \infty$.

This definition describes that coordinates (or "entries") of a vector $V$ all have the same *typical size* $\Theta(n^\alpha)$ in magnitude, since entries of $V$ are approximately *independent and identically distributed (IID)* when $n \to \infty$. This is also what Hayou et al. (2024a) mean by "convergence is understood to be convergence in second moment". Definition E.1 could be trivially proven with the following equation, assuming coordinates of $V$ have the typical size $\Theta(n^\alpha)$:

$$\frac{\|V\|_2^2}{n} = \frac{1}{n}\sum_{i=1}^n v_i^2 = \frac{1}{n} \cdot n \cdot (\Theta(n^\alpha))^2 = \Theta(n^{2\alpha}). \tag{7}$$

## E.2. $\gamma$-operator

The $\gamma$-operator is motivated by quantifying the asymptotic effect of scaling certain neural network components. In this paper, we focus on the effect of scaling model width on SV training dynamics based on a frozen pretrained model as model width $n$ approach infinity.

As is introduced in the main body, the $\gamma$-operator is a logarithm-like operator defined as $\gamma[v] = \beta$ where $v = \Theta(n^\beta)$, which is a mapping $\gamma : \{v | v = \Theta(n^\alpha), \alpha \in \mathbb{R} \cup \{-\infty\}\} \to \mathbb{R} \cup \{-\infty\}$.

We will now introduce several basic rules of the $\gamma$-operator that are often used in this paper. Previous work primarily use the first three rules; we additionally introduce the division rule.

**Zero.** $\gamma[0] = -\infty$.

**Addition.** $\forall v, v' \in \mathbb{R}$ and $v \neq -v'$ as $n \to \infty$, we have $\gamma[v+v'] = \max(\gamma[v], \gamma[v'])$. It is essential that $v \neq -v'$, otherwise we have $\gamma[v + v'] = \gamma[0] = -\infty$.

**Product.** $\forall v, v' \in \mathbb{R}$, we have $\gamma[vv'] = \gamma[v] + \gamma[v']$.

**Division.** $\forall v, v' \in \mathbb{R}$ ($v' \neq 0$ as $n \to \infty$), we have $\gamma[v/v'] = \gamma[v] - \gamma[v']$.

## E.3. Preliminary Theorems

Here we showcase useful intermediate theoretical results.

**Lemma E.2.** *For any $V \in \mathbb{R}^n$, then $\gamma\left[\|V\|_2\right] = \gamma[V] + 1/2$.*

*Proof.* Based on Definition E.1, supposing $V = \Theta(n^\alpha)$, we have:

$$\begin{aligned}\|V\|_2^2 &= n \cdot \Theta(n^{2\alpha}) = \Theta(n^{2\alpha+1}), \\ \|V\|_2 &= \Theta(n^{\alpha+1/2}).\end{aligned} \tag{8}$$

By definition of the $\gamma$-operator, we have: $\gamma[\|V\|_2] = \alpha + 1/2$. $\square$

**Lemma E.3.** *For any $V \in \mathbb{R}^n$ and $V \neq 0$, then $\gamma[V/\|V\|_2] = -1/2$.*

*Proof.* Since $V \neq \mathbf{0}$, we have:

$$\gamma\left[\frac{V}{\|V\|_2}\right] = \gamma[V] - \gamma[\|V\|_2]. \tag{9}$$

Based on Lemma E.2, we have $\gamma[\|V\|_2] = \alpha + 1/2$. Then we obtain:

$$\gamma\left[\frac{V}{\|V\|_2}\right] = \gamma[V] - \left(\gamma[V] + \frac{1}{2}\right) = -\frac{1}{2}. \tag{10}$$

This result indicates that any normalized vector is of order $\Theta(n^{-1/2})$, same as Kaiming initialization ($v_i \sim \mathcal{N}(0, n^{-1})$). $\square$

**Lemma E.4.** *Let $v \in \mathbb{R}$ be a random variable with $v \sim \mathcal{N}(0, \sigma^2)$ where $\sigma = \Theta(n^\alpha)$, then $v = \Theta(n^\alpha)$ and $\gamma[v] = \alpha$.*

*Proof.* According to the empirical percentile rule of the Gaussian distribution, approximately 99.7% of samples lie within $\pm 3\sigma$ of the mean. Thus we have the following inequality for *typical values* of $v$:

$$0 \leq |v| \leq 3\sigma. \tag{11}$$

By definition of the $\Theta(\cdot)$ notation, we have $v = \Theta(\sigma) = \Theta(n^\alpha)$ for typical values of $v$ and thus $\gamma[v] = \alpha$. In words, the typical asymptotic size of a Gaussian variable with zero mean is of the same order as its standard deviation.

We then show that this result is not limited to Gaussian distributions with zero mean and extends to uniform distributions with zero mean. Suppose $v$ follows a symmetric uniform distribution: $v \sim \mathcal{U}(-b, b)$, where $b = \Theta(n^\beta)$. We then have:

$$0 \leq |v| \leq b = \Theta(n^\beta). \tag{12}$$

Thus we obtain $v = \Theta(n^\beta)$. Since the variance of $v$ is $\sigma^2 = b^2/3$, again we have $v = \Theta(n^\beta) = \Theta(\sigma)$. $\square$

Lemma E.4 is useful for deriving variances for random initialization of parameter weights with known typical entry size and zero mean, as well as deriving the scaling rule for a random variable with known variances and zero mean.

**Lemma E.5.** *Let $V, X \in \mathbb{R}^n$, where $V$ is a random vector with IID entries: $v_i \sim \mathcal{N}(0, \sigma_v^2)(i = 1, 2, \ldots, n)$ where $\sigma_v = \Theta(n^\alpha)$, and $X$ is a vector with constant entry size: $x_i = \Theta(1)$. Then $V^\top X = \Theta(n^{\alpha+1/2})$.*

*Proof.* First we have:

$$V^\top X = \sum_{i=1}^{n} v_i x_i. \tag{13}$$

Since $x_i = \Theta(1)$, there exist constants $\kappa_x^l, \kappa_x^h$ such that $\kappa_x^l < x_i < \kappa_x^h$. Thus we obtain:

$$n\bar{v}\kappa_x^l = \sum_{i=1}^{n} v_i \kappa_x^l < V^\top X = \sum_{i=1}^{n} v_i x_i < \sum_{i=1}^{n} v_i \kappa_x^h = n\bar{v}\kappa_x^h, \tag{14}$$

where $n\bar{v} \sim \mathcal{N}(0, n\sigma_v^2)$ according to the *Central Limit Theorem (CLT)*. Therefore we have $0 \leq |n\bar{v}| \leq 3\sigma_v\sqrt{n}$ for typical values of $\bar{v}$ according to the 99.7% percentile rule of the Gaussian distribution. Taking this into the equation above, we have the following for typical values of $\bar{v}$:

$$0 \leq |V^\top X| < n\bar{v}\kappa_x^h < 3\kappa_x^h \sigma_v \sqrt{n}. \tag{15}$$

By definition of the $\Theta(\cdot)$ notation, we have $V^\top X = \Theta(\sigma_v\sqrt{n}) = \Theta(n^{\alpha+1/2})$. $\square$

Lemma E.5 is useful in the setting of fine-tuning and SV training, where $V$ corresponds to randomly initialized weights and $X = \Theta(1)$ corresponds to representations of a pretrained model.

**Lemma E.6.** *Let $V, X \in \mathbb{R}^n$, where entries of $V$ have non-zero mean, $v_i = \Theta(n^\alpha), i = 1, 2, \ldots, n$, and $X$ is a vector with constant entry size: $x_i = \Theta(1)$. Then $V^\top X = \Theta(n^{\alpha+1})$.*

*Proof.* We directly obtain:

$$V^\top X = \sum_{i=1}^{n} v_i x_i = n \cdot \Theta(n^\alpha) \cdot \Theta(1) = \Theta(n^{\alpha+1}). \tag{16}$$

□

Lemma E.6 is introduced by Li et al. (2025). Note an important distinction between Lemma E.6 and Lemma E.5: $v_i$ has non-zero mean in the former case, thus the CLT is not used.

**Lemma E.7.** *Let $V \in \mathbb{R}^n$ be a random vector with IID entries: $v_i \sim \mathcal{N}(0, \sigma_v^2), i = 1, 2, \ldots, n$, and $U := V/\|V\|_2$ be the normalization of $V$. Let $X$ be a vector with constant entry size: $x_i = \Theta(1), i = 1, 2, \ldots, n$. Then $U^\top X = \Theta(1)$.*

*Proof.* Intuitively, $U^\top X$ is a standard projection operation, where $X$ is projected onto the one-dimensional subspace defined by $U$. Since the projection operation is equivalent to selecting a single entry of $X$ (either coordinate-aligned or not), the projection value is the typical size of a single $X$ entry. Given that $x_i = \Theta(1)$, we naturally obtain $U^\top X = \Theta(1)$.

We will now provide a formal derivation as follows.

We first have:

$$U^\top X = \sum_{i=1}^{n} u_i x_i = \frac{1}{\|V\|_2} \sum_{i=1}^{n} v_i x_i = \frac{1}{\|V\|_2} \sum_{i=1}^{n} \bar{v} x_i, \tag{17}$$

where the sample mean $\bar{v} \sim \mathcal{N}(0, \sigma_v^2/n)$ according to the CLT. Let $\sigma_v = \Theta(n^\alpha)$. Based on Lemma E.4, we have $v = \Theta(n^\alpha)$ and $\bar{v} = \Theta(n^{\alpha-1/2})$. Based on Lemma E.2, we have $\|V\|_2 = \Theta(n^{\alpha+1/2})$ and thus $1/\|V\|_2 = \Theta(n^{-\alpha-1/2})$. Taking these results into the equation above, we obtain:

$$U^\top X = \Theta(n^{(-\alpha-1/2)+(\alpha-1/2)}) \cdot \sum_{i=1}^{n} x_i = \Theta(n^{-1}) \cdot n \cdot \Theta(1) = \Theta(1). \tag{18}$$

□

**Lemma E.8.** *Let $V, W \in \mathbb{R}^n (V, W \neq \mathbf{0})$, then $\gamma\left[V/\|V\|_2 - W/\|W\|_2\right] = -1/2$ or $-\infty$.*

*Proof.* Based on Lemma E.3, we have $\gamma\left[V/\|V\|_2\right] = \gamma\left[W/\|W\|_2\right] = -1/2$. Assume $V/\|V\|_2 - W/\|W\|_2 \neq \mathbf{0}$, then we have:

$$\gamma\left[\text{LHS}\right] = \max\left(\gamma\left[\frac{V}{\|V\|_2}\right], \gamma\left[\frac{W}{\|W\|_2}\right]\right) = -\frac{1}{2}. \tag{19}$$

This indicates that a non-zero update to a unit vector is of order $\Theta(n^{-1/2})$. However, if $\text{LHS} = \mathbf{0}$, which means $V$ and $W$ are the same direction and differ only in scale, we directly obtain $\gamma[\text{LHS}] = \gamma[0] = -\infty$.

In summary:

$$\gamma\left[\frac{V}{\|V\|_2} - \frac{W}{\|W\|_2}\right] = \begin{cases} -\frac{1}{2}, & \frac{V}{\|V\|_2} \neq \frac{W}{\|W\|_2}; \\ -\infty, & \frac{V}{\|V\|_2} = \frac{W}{\|W\|_2}. \end{cases} \tag{20}$$

□

### E.4. Analysis of SVs with Clamping Intervention

In the main body, we primarily discuss SVs with addition intervention (AddInv). *Clamping intervention (ClampInv)* has also been used for steering by prior work as an alternative to AddInv (Templeton et al., 2024; Wu et al., 2025a). In this subsection, we show that ClampInv has scaling rules for learning rates and initialization strategies similar to those of AddInv in the infinite-width limit.

**Notations.** ClampInv sets the value along the SV direction to a constant: $\Phi^{\text{Clamp}}(\mathbf{h}; \alpha, \mathbf{v}) = \mathbf{h} + \alpha\mathbf{v} - \mathbf{u}\mathbf{u}^\top\mathbf{h}$, where $\mathbf{u} := \mathbf{v}/\|\mathbf{v}\|_2$ is the normalized direction.

**SV features.** SV features of ClampInv is: $\mathbf{z}^{\text{Clamp}} = \Phi^{\text{Clamp}}(\mathbf{h}) - \mathbf{h} = \alpha\mathbf{v} - \mathbf{u}\mathbf{u}^\top\mathbf{h}$.

**Stability.** According to Definition 4.1, we have the following for ClampInv:

$$0 = \gamma \left[ \mathbf{z}_t^{\text{Clamp}} \right] = \gamma \left[ \alpha_t \mathbf{v}_t - \mathbf{u}_t \mathbf{u}_t^\top \mathbf{h} \right]. \tag{21}$$

**Efficiency.** The update to ClampInv features is given by:

$$\Delta \mathbf{z}_t^{\text{Clamp}} = \underbrace{(\Delta \alpha_t) \mathbf{v}_{t-1}}_{\delta_t^1} + \underbrace{\alpha_{t-1}(\Delta \mathbf{v}_t)}_{\delta_t^2} + \underbrace{(\Delta \alpha_t)(\Delta \mathbf{v}_t)}_{\delta_t^3} - \underbrace{(\Delta \mathbf{u}_t)(\Delta \mathbf{u}_t)^\top \mathbf{h}}_{\delta_t^4}. \tag{22}$$

According to Definition 4.2, we require $\delta_t^i = \Theta(1), i = 1, 2, 3, 4$ for ClampInv.

**Requirements of both stability and efficiency for ClampInv.** We show the condition for ClampInv to achieve stability and efficiency as follows:

$$\begin{cases} \gamma \left[ \delta_t^1 \right] = \gamma \left[ -\eta_\alpha g_{t-1}^\alpha \mathbf{v}_{t-1} \right] = 0, \\ \gamma \left[ \delta_t^2 \right] = \gamma \left[ -\eta_{\mathbf{v}} g_{t-1}^{\mathbf{v}} \alpha_{t-1} \right] = 0, \\ \gamma \left[ \delta_t^3 \right] = \gamma \left[ (\Delta \alpha_t)(\Delta \mathbf{v}_t) \right] = 0, \\ \gamma \left[ \delta_t^4 \right] = \gamma \left[ (\Delta \mathbf{u}_t)(\Delta \mathbf{u}_t)^\top \mathbf{h} \right] = 0, \\ \gamma \left[ \mathbf{z}_t^{\text{Clamp}} \right] = \gamma \left[ \alpha_t \mathbf{v}_t - \mathbf{u}_t \mathbf{u}_t^\top \mathbf{h} \right] = 0. \end{cases} \tag{23}$$

According to Lemma E.7 and $\gamma[\mathbf{u}] = -1/2$ (Lemma E.3), we have $\gamma[\mathbf{u}_t \mathbf{u}_t^\top \mathbf{h}] = -1/2$ when $t = 0$ (since we initialize $\mathbf{v}_0$ with Kaiming initialization); according to Lemma E.6, we have $\gamma[\mathbf{u}_t \mathbf{u}_t^\top \mathbf{h}] = 0$ for $t > 1$. Using Lemma E.8, we have $\gamma[\Delta \mathbf{u}_t] = -1/2$ as long as $\Delta \mathbf{u}_t \neq \mathbf{0}$.

We now focus on $\Delta \mathbf{u}_t$. We have: $\Delta \mathbf{u}_t = \mathbf{u}_t - \mathbf{u}_{t-1} = \mathbf{v}_t/\|\mathbf{v}_t\|_2 - \mathbf{v}_{t-1}/\|\mathbf{v}_{t-1}\|_2$, where $\mathbf{v}_t = \mathbf{v}_{t-1} - \eta_{\mathbf{v}} g_{t-1}^{\mathbf{v}}$. $\Delta \mathbf{u}_t = \mathbf{0}$ can be met only if: (1) $g_{t-1}^{\mathbf{v}}$ is parallel to $\mathbf{v}_{t-1}$, (2) $g_{t-1}^{\mathbf{v}} = 0$ or (3) $\eta_{\mathbf{v}} = 0$. (1) is rarely met in practice in early stages of training; (2) is almost never satisfied since SVs are almost never sufficiently expressive for perfect convergence; (3) is never true since learning rates are always positive. Therefore, according to Lemma E.6, we have: $\gamma[(\Delta \mathbf{u}_t)^\top \mathbf{h}] = \gamma[\Delta \mathbf{u}_t] + \gamma[\mathbf{h}] + 1 = \gamma[\Delta \mathbf{u}_t] + 1$.

Taking these intermediate results into the system of equations above, we have the following for $t > 1$:

$$\begin{cases} \gamma[\eta_\alpha] + \max(\gamma[\mathbf{v}_0], \gamma[\eta_{\mathbf{v}}]) = 0, \\ \gamma[\eta_{\mathbf{v}}] + \max(\gamma[\alpha_0], \gamma[\eta_\alpha]) = 0, \\ \gamma[\eta_{\mathbf{v}}] + \gamma[\eta_\alpha] = 0, \\ 2\gamma[\Delta \mathbf{u}_t] + 1 = 0, \\ \max(\gamma[\alpha_t \mathbf{v}_t], 0) = 0. \end{cases} \tag{24}$$

The solution is the same as Equation (6):

$$\begin{cases} \gamma[\eta_{\mathbf{v}}] + \gamma[\eta_\alpha] = 0, \\ \gamma[\mathbf{v}_0] \leq \gamma[\eta_{\mathbf{v}}], \gamma[\alpha_0] \leq \gamma[\eta_\alpha]. \end{cases} \tag{25}$$

**Comparing AddInv and ClampInv.** Although AddInv and ClampInv have the same condition for stability and efficiency of training, they are different in terms of stability under suboptimal hyperparameters. From Equation (24), ClampInv ensures stability when $t > 1$ due to the projection term $-\mathbf{u}\mathbf{u}^\top \mathbf{h}$. In contrast, AddInv has no lower bound for SV feature.

However, we find that this stability guarantee has limited practical benefits. Our empirical results of §J.2 indicate no essential difference between AddInv and ClampInv. Therefore, based on our current evidence, **we recommend giving priority to AddInv for its simplicity and lower computational cost**.

## F. Representation Steering Methods

In this section, we provide a detailed description of the representation steering methods involved in this paper. We will first describe the common essential components of trainable representation steering methods; then we will introduce the formulations of optimization-free and trained steering methods involved in this paper.

## F.1. Basic Components of Representation Steering

In general, a trainable representation steering method has the following design considerations: (1) intervention functional form; (2) training objective; (3) intervention location; (4) choice of hyperparameters (e.g., steering factor). In this paper, we use theoretical tools to analyze the effect of (1) functional form and (4) hyperparameters on SV training, while investigating the effect of (2) training objective and (3) intervention location with a purely empirical approach.

SV intervention functional form and intervention location are already discussed in §3 and §5. We will now introduce the various SV training objectives and other representation steering methods with more complex functional forms.

## F.2. Clarification of Terminology

To avoid confusion, we would like clarify several terms regarding SVs. In this paper, we call the representation steering **method** or **intervention** ($\Phi$) that is parameterized by a one-dimensional vector the *steering vector (SV)*. We call the **vector** parameter ($\mathbf{v}$) the *steering direction*, to ensure consistency with the name of the scaling coefficient ($\alpha$): the *steering factor*.

## F.3. Prompt Steering

According to AXBENCH implementation[1] and Wu (2025), steering prompts (i.e. prompts that request concepts to be incorporated in responses) are prepended to original instructions. For example, if an steering prompt is $\mathbf{x}_s$ ="When responding to questions, please include references to programming constructs and data structures, even if they don't directly relate to the question." while the original instruction is $\mathbf{x}$ ="How can I make a cake?", then the actual prompt is $\mathbf{x}_s + \mathbf{x}$ where $+$ is sequence concatenation operation.

In this paper, we follow the advice of Wu (2025) and always use prompting as a baseline for its flexibility and simplicity.

## F.4. Steering Vectors

**Difference-in-means (DiffMean; Marks & Tegmark (2023)).** DiffMean is an optimization-free SV. The same method is also termed *activation addition (ActAdd)* or *contrastive activation addition (CAA)* (Rimsky et al., 2024). DiffMean computes the difference between the mean of representations of two classes of inputs, therefore it uses contrastive examples: a set of negative examples $\mathcal{D}^- = \{(\mathbf{x}_j^-, \mathbf{y}_j^-)\}_{j=1}^M$ and a set of positive examples $\mathcal{D}^c = \{(\mathbf{x}_i, \mathbf{y}_i^c)\}_{i=1}^N$, where $(\mathbf{x}_j^-, \mathbf{y}_j^-)$ does not incorporate concept $c$ while $(\mathbf{x}_i, \mathbf{y}_i^c)$ does. Following Wu et al. (2025a), DiffMean is formally defined as follows:

$$\mathbf{v}_{\text{DiffMean}} = \mathbb{E}_{(\mathbf{x}, \mathbf{y}^c) \in \mathcal{D}^c} \left[ \mathbf{h}(\mathbf{x} + \mathbf{y}^c) \right] - \mathbb{E}_{(\mathbf{x}^-, \mathbf{y}^-) \in \mathcal{D}^-} \left[ \mathbf{h}(\mathbf{x} + \mathbf{y}) \right], \tag{26}$$

where $\mathbf{x} + \mathbf{y}$ denotes sequence concatenation and $\mathbf{h}(\mathbf{x})$ is the representation value given input $\mathbf{x}$.

At inference time, DiffMean uses addition intervention (AddInv), with unit steering factors (Arditi et al., 2024) or factors chosen via factor selection (Turner et al., 2023; Rimsky et al., 2024; Wu et al., 2025a; Chen et al., 2025b). The exact factor selection procedures differ among previous works, but their general logic is similar to Algorithm 3 and differences lie primarily in scoring metrics.

**Language modeling (Lang.; Subramani et al. (2022); Wu et al. (2025b)).** The Lang. objective is agnostic of intervention functional form and maximizes the log-likelihood of the steered response $\mathbf{y}^c$ conditioned by input prompt $\mathbf{x}$:

$$\arg\min_{\Phi} -\log p_\Phi(\mathbf{y}^c|\mathbf{x}; \mathbf{h} \leftarrow \Phi(\mathbf{h})) = -\sum_{i=1}^{|\mathbf{y}^c|} \log p_\Phi(\mathbf{y}_i^c|\mathbf{y}_{<i}^c, \mathbf{x}; \mathbf{h} \leftarrow \Phi). \tag{27}$$

Therefore Lang. only requires training data $\mathcal{D}^c = \{(\mathbf{x}_i, \mathbf{y}_i^c)\}_{i=1}^N$ and does not need contrastive pairs.

**Reference-free preference steering (RePS; Wu et al. (2025b)).** RePS is a bi-directional SV that optimizes for both concept-based steering and concept suppression. Its design considerations include both training objective and intervention functional form. It is based on *simple preference optimization (SimPO)* (Meng et al., 2024) and frames the bi-directional steering task as a bi-directional preference optimization task, where the direction of preference is controlled by the steering factor. Let $\mathbf{y}^c$ be a steered response to prompt $\mathbf{x}$ and $\mathbf{y}$ be a concept-neutral response. SimPO and RePS objectives both require contrastive training data $\mathcal{D}^{c+} = \{(\mathbf{x}_i, \mathbf{y}_i, \mathbf{y}_i^c)\}_{i=1}^N$, which is more demanding than Lang. training data.

---

[1] https://github.com/stanfordnlp/axbench/blob/c01a22adcc6a2e1d3ec663c7577afac85ae03771/axbench/utils/dataset.py#L697

The first component of RePS objective optimizes for concept-based steering using addition intervention, such that the steered response is preferred over the neutral response ($\mathbf{y}^c \succ \mathbf{y}$):

$$\Delta_\Phi^+ = \frac{\beta^+}{|\mathbf{y}^c|} \log p_\Phi\left(\mathbf{y}^c|\mathbf{x}; \mathbf{h} \leftarrow \Phi^{\text{Add}}(\mathbf{h}; \alpha, \mathbf{v})\right) - \frac{1}{|\mathbf{y}|} \log p_\Phi\left(\mathbf{y}|\mathbf{x}; \mathbf{h} \leftarrow \Phi^{\text{Add}}(\mathbf{h}; \alpha, \mathbf{v})\right), \quad (28)$$

where $\beta^+ = \max(\log(p(\mathbf{y}|\mathbf{x})) - \log(p(\mathbf{y}^c|\mathbf{x})), 1)$. The second component of RePS objective models concept suppression, where the concept-neutral response is preferred over the steered response ($\mathbf{y} \succ \mathbf{y}^c$):

$$\Delta_\Phi^- = \frac{\beta^-}{|\mathbf{y}|} \log p_\Phi\left(\mathbf{y}|\mathbf{x}; \mathbf{h} \leftarrow \Phi_{\text{Null}}(\mathbf{h}; \mathbf{v})\right) - \frac{1}{|\mathbf{y}^c|} \log p_\Phi\left(\mathbf{y}^c|\mathbf{x}; \mathbf{h} \leftarrow \Phi_{\text{Null}}(\mathbf{h}; \mathbf{v})\right), \quad (29)$$

where $\beta^- = \max(\log(p(\mathbf{y}^c|\mathbf{x})) - \log(p(\mathbf{y}|\mathbf{x})), 1)$. $\Phi_{\text{Null}}(\cdot)$ intervention ablates the steering vector via orthogonalization, i.e. clamping values along the $\mathbf{v}$ direction to zero:

$$\Phi_{\text{Null}}(\mathbf{h}; \mathbf{v}) = \mathbf{h} - \text{ReLU}\left(\mathbf{h}^\top \mathbf{u}\right) \mathbf{u}, \quad (30)$$

where $\mathbf{u} := \mathbf{v}/\|\mathbf{v}\|_2$ is the normalized vector. The final objective is:

$$\arg\min_\Phi -\left[\log \sigma\left(\Delta_\Phi^+\right) + \log \sigma\left(\Delta_\Phi^-\right)\right], \quad (31)$$

where $\sigma(\cdot)$ is sigmoid function. During inference, RePS uses addition intervention (AddInv) for both concept-based steering (with $\alpha > 0$) and concept suppression (with $\alpha > 0$), where factors are selected according to Algorithm 3.

*Remark.* In this paper, when we say the SimPO objective is used for FSSV/PrOSV with joint training, we only use the positive steering objective. This is because we focus solely on uni-directional concept-based steering, not concept suppression. That being said, it is possible to extend our joint training scheme of Algorithm 1 to the bi-directional steering scenario, which requires training separate steering factors: one for concept-based steering and another for concept suppression.

### F.5. Representation Fine-Tuning

Representation fine-tuning (ReFT) methods use low-rank projection interventions. ReFT methods use $(2rn + 1)$ parameters where $1 \leq r \ll n$ is low-rank dimension, which is at least twice that of SVs ($n+1$). Prior work has shown that ReFT is often sufficiently effective with $r \geq 4$ (Wu et al., 2024b). Therefore rank-4 LoReFT is a competitive baseline on AXBENCH (Wu et al., 2025a).

We introduce two primary variants of ReFT as follows.

**Low-rank ReFT (LoReFT; Wu et al. (2024b)).**

$$\Phi(\mathbf{h}) = \mathbf{h} + \mathbf{u}(\mathbf{w}^\top \mathbf{h} + \mathbf{b} - \mathbf{u}^\top \mathbf{h}), \quad (32)$$

where $\mathbf{w}, \mathbf{u} \in \mathbb{R}^{n \times r}, \mathbf{b} \in \mathbb{R}^r$ are parameters, $r \ll n$ is low-rank dimension and $\mathbf{u}$ has orthonormal columns.

**Direct ReFT (DiReFT; Wu et al. (2024b)).** The main differences between DiReFT and LoReFT are that DiReFT does not impose constraints on write-out matrix and does not cancel out $\mathbf{u}^\top \mathbf{h}$.

$$\Phi(\mathbf{h}) = \mathbf{h} + \mathbf{v}(\mathbf{w}^\top \mathbf{h} + \mathbf{b}), \quad (33)$$

where $\mathbf{w}, \mathbf{v} \in \mathbb{R}^{n \times r}, \mathbf{b} \in \mathbb{R}^r$ are parameters and $r \ll n$ is low-rank dimension.

### F.6. Discussions on PrOSV versus Other Representation Steering Techniques

**PrOSV vs. dynamic SVs.** Recent work has proposed enhancing strategies based on the traditional FSSV paradigm. For instance, in order to maximally preserve general model utility, Ferrando et al. (2025) propose to modulate traditional SVs by dynamically adjust steering factors for each token. However, the computational overhead of the modulation operation for each token (essentially a dot product followed by a comparison) requires multiple dot products and a single dot product takes $2n$ FLOPS, while the computational cost of an addition intervention is already $n$ at minimum.

In contrast, PrOSV only intervenes on a constant number of prompt tokens at prefill stage. This makes it the most efficient steering method in terms of both computational cost.

*Table 5.* Comparison of inference-time computational overhead and parameter count between representation steering methods.

| Method | $\Phi(\mathbf{h})$ definition | Computational overhead | # of parameters |
|---|---|---|---|
| AddInv | $\mathbf{h} + \alpha\mathbf{v}$ | $2n$ (minimum $n$) | $n$ |
| + Joint training | – | – | $n+1$ |
| ClampInv | $\mathbf{h} + \alpha\mathbf{v} - \mathbf{u}\mathbf{u}^\top\mathbf{h}$ | $9n-1$ (minimum $4n$) | $n$ |
| + Joint training | – | – | $n+1$ |
| DiReFT (rank-1) | $\mathbf{h} + \mathbf{v}(\mathbf{w}^\top\mathbf{h} + b)$ | $4n$ | $2n+1$ |
| LoReFT (rank-1) | $\mathbf{h} + \mathbf{u}(\mathbf{w}^\top\mathbf{h} - \mathbf{u}^\top\mathbf{h} + b)$ | $6n-1$ (minimum $4n$) | $2n+1$ |

**PrOSV vs. KV cache steering.** Recently, Belitsky et al. (2025) introduce *KV cache steering* as an alternative to SVs and focus on controlling the chain-of-thought reasoning process of small models. KV cache steering is able to achieve effective model control on reasoning tasks by manipulating the prompt KV cache at all layers, which is similar to PrOSV. Additionally, it is robust to a wider range of steering factors than SVs.

However, there are several essential theoretical distinctions between PrOSV and KV cache steering:

- **Data requirement**: KV cache steering requires *contrastive prompts* to curate *KV steering vectors*, in a similar spirit to DiffMean; however, PrOSV does not always require contrastive examples and is able to achieve good effectiveness with only positive examples using the Lang. objective. This gives PrOSV more flexibility over KV cache steering.

- **Source of SV**: KV cache steering is optimization-free while PrOSV is fine-tuned, which makes PrOSV more dedicated for the concept of interest;

- **Mechanism**: KV cache steering only controls model behaviors through KV cache; in addition to implicitly manipulating KV cache, PrOSV is able to affect the generation of the first token through the residual stream if the last prompt token is intervened on.

In addition to the reasons above, we do not include KV cache steering as a main baseline for three reasons. First, we primarily discuss representation steering methods that manipulate residual stream representations. Second, Belitsky et al. (2025) report *mixed results* of KV cache steering, where it sometimes underperforms DiffMean; whereas we can notice large performance gaps between DiffMean and FSSV on AXBENCH.

Third, we conducted experiments using the official implementation of KV cache steering[2] on Qwen2.5-32B using CON-CEPT10 subset. We use steering prompts as positive prompts and concept-neutral instructions as negative prompts. This choice is justified by the AXBENCH experiment (§6.3) where steering prompts are highly effective. We use a key steering strength ($c^k$) of 0.0 and value strength of 6 since this setting is used by many models they tested. We assume that this choice does not severely affect steering performance since Belitsky et al. (2025) highlight that KV cache steering is robust to choices of strengths. The resulting overall score is 0.180, whereas the overall score of FSSV with Lang. objective is 0.919. This suggests that KV cache steering might not be a competitive baseline for concept-based steering.

**PrOSV vs. ReFT.** According to Table 2, rank-4 LoReFT underperforms PrOSV on AXBENCH. This seems counterintuitive since LoReFT should be far more expressive than PrOSV in terms of the scope of possible function mappings. We hypothesize that LoReFT has *more than sufficient* capacity for concept-based steering whereas this task is of lower rank. Empirically, we find that rank-1 ReFT could already overfit on the training set of 72 examples on Gemma2-2B, such that the intervened model only steers successfully on training instructions but *not* on unseen instructions. Therefore, rank-4 ReFT might learn redundant, spurious patterns when the steering task is only rank-1.

## G. Analysis of Computational Overhead

Since we highlight the tradeoff between computational cost and steering effectiveness in this paper, in this section we study the computational overhead of SVs from both theoretical and empirical perspectives. This section is built upon the discussions of Wu (2025), who advises against claiming "FSSV is computationally efficient" without additional information regarding the context size.

---

[2] https://github.com/MaxBelitsky/cache-steering

## G.1. Inference-Time Overhead of SVs

Here we provide an analysis of the inference-time computational overhead for SVs. For each method, we will split the computational cost into prefill/decode stages, respectively.

**Notations.** We first introduce the key symbols used in this section:

- $n$: Model width.
- $L$: Number of layers.
- $H$: Number of attention heads.
- $P$: Length of original prompt.
- $S$: Length of steering prompt.
- $T$: Current total context length (past tokens).
- $I$: Number of prompt-only interventions ($I < P$).

We assume that KV cache is used, and that 1 addition/multiplication operation takes 1 FLOPs.

**Un-intervened inference when KV cache is used.** Our analysis is based on Mu et al. (2023), and focus on the computation of MLP/attention modules. In general, the cost is split between non-KV operations (linear projections, MLP) and KV operations (self-attention).

KV FLOPs (per token, for context T):

- Key/query logits: $2nT$.
- Softmax: $3HT$.
- Softmax $\times$ query reductions: $2nT$.

Thus KV FLOPS is $4nT + 3HT$.

Non-KV FLOPs (per token):

- MLP: $2*2*n*(4n) = 16n^2$.
- Key/query/value projections: $6n^2$.
- Linear projection after attention: $2n^2$.

Thus non-KV FLOPs is $24n^2$.

At prefill stage, all prompt tokens ($P$) are processed at once; attention is quadratic but non-KV operations are linear:

$$\text{Cost}_{\text{prefill}} \approx L(24n^2P + 2nP^2), \tag{34}$$

where $2nP^2$ approximates the sum of attention costs $\sum_{i=1}^{P} 4nt$.

At decode stage, a single token is generated while attending to $T$ past tokens:

$$\text{Cost}_{\text{decode}} = L(24n^2 + 4nT + 3HT). \tag{35}$$

**Prompt steering.** Prompt steering prepends $S$ tokens to the instruction. This increases the sequence length for both prefill and decode stages.

At prefill phase, the model processes $S$ extra tokens:

$$\Delta\text{Cost}_{\text{prefill}}^{\text{prompt}} \approx SL(24n^2 + 2n(2P + S)), \tag{36}$$

At decode stage, the context length $T$ is increased by $S$. The overhead is the cost of attending to these extra tokens in the KV cache at each decoding step:

$$\Delta\text{Cost}_{\text{decode}}^{\text{prompt}} = L(4nS + 3HS). \tag{37}$$

**SVs with addition interventions (AddInv).** Addition intervention for a single token:

*Table 6.* Summary of computational cost and overhead for prefill and decode stages, as well as estimated overhead for Llama2-7B as an example ($L = 32, n = 4096, H = 32, P = 20, S = 100, I = 4, R = 128, T \in \{P, P + 1, \ldots, P + R - 1\}$).

| Item | Method | Prefill | | Decode (Single token) | |
|------|--------|---------|---------|-----------------------|---------|
| | | Formula | Example | Formula | Example |
| Cost | Un-intervened | $L(24n^2P + 2nP^2)$ | — | $L(24n^2 + 4nT + 3HT)$ | — |
| Overhead | Prompt steering | $SL(24n^2 + 2n(2P + S))$ | 83.57% | $L(4nS + 3HS)$ | 0.41% |
| | AddInv (FSSV) | $Pn$ | 3.2e-5% | $n$ | 3.2e-5% |
| | ClampInv (FSSV) | $4Pn$ | 1.3e-4% | $4n$ | 1.3e-4% |
| | AddInv (PrOSV) | $In$ | 6.4e-6% | 0 | 0 |
| | ClampInv (PrOSV) | $4In$ | 2.5e-5% | 0 | 0 |

- Scaling SV with coefficient: $n$ FLOPs.

- Adding scaled vector to representation: $n$ FLOPs.

Therefore the computational overhead of SVs with addition interventions is $2n$. Supposing the scaled direction is precomputed, then the minimum overhead is $n$.

**SVs with clamping interventions (ClampInv).** Clamping intervention for a single token:

- Vector norm: $n$ multiplications, $n - 1$ addition and 1 square root: $2n$ FLOPs.

- Division to obtain unit vector: $n$ FLOPs.

- Up/down projection: $2n$ multiplications and $n - 1$ additions: $3n - 1$ FLOPs.

- Scaling SV with coefficient: $n$ FLOPs.

- Final intervention: $2n$ FLOPs.

Thus the computational overhead of ClampInv is $9n - 1$. Assuming unit vector and scaled vector are precomputed, the minimum overhead is $4n$ (dot product $\mathbf{u}^\top \mathbf{h}$: $2n - 1$, where $\mathbf{u}$ is precomputed; subtraction $\alpha \|\mathbf{v}\|_2 - \mathbf{u}^\top \mathbf{h}$: 1, where $\alpha \|\mathbf{v}\|_2$ is precomputed; scaling $\mathbf{u}$: $n$, addition: $n$).

**Example.** We take Llama-2-7B (Touvron et al., 2023) as an example, where $n = 4096$, $H = 32$, $L = 32$.

In our paper, the average length of steering prompts is $S \approx 100$ and the average length of instructions is $P \approx 20$. We also assume the context history is around $T = 20 + 128$, which is the sum of prompt length ($P$) and response length ($R = 128$). Therefore the overhead of prompt steering at prefill stage is:

$$\frac{S(24n^2 + 4nP + 2nS)}{24n^2(P + S) + 2n(P + S)^2} \approx 83.57\%. \tag{38}$$

However, as has been pointed out by Wu (2025), the overhead of prompt steering is far less apparent in long-context settings. For instance, the prefill overhead could be as low as 0.77% if prompt length is large, e.g., $P = 16K$.

The overhead of prompt steering at decode stage is:

$$\frac{4nS + 3HS}{24n^2 + 4nT + 3HT} \approx 0.41\%. \tag{39}$$

For SVs, we only apply interventions at a single layer, and we consider both FSSV and PrOSV. For PrOSV, we let computational budget be $I = 4$ tokens, consistent with the optimal setup we found in §6.2.

We provide an overview of computational cost/overhead in Table 6. Overall, FSSVs already outperform prompt steering in terms of steering efficiency, whereas PrOSVs take efficiency to the extreme with lower overhead at prefill stage and zero overhead at decode stage. Although the absolute value seems small, the gap in computational cost is huge in terms of relative measurement. When generating all $R = 128$ tokens with prompt length $P = 20$, **the total cost of PrOSV is 1/37 that of FSSV**.

*Table 7.* Sets of steering factors for SV training and inference (Wu et al., 2025b).

| Configuration | Factor set |
|---|---|
| Gemma2-2B/9B training ($\mathcal{A}_{\text{sample}}$) | {2.0, 4.0, 6.0, 8.0, 10.0, 12.0, 14.0, 16.0, 18.0, 20.0} |
| Gemma2-2B/9B inference ($\mathcal{A}_{\text{search}}$) | {2.0, 4.0, 6.0, 8.0, 10.0, 12.0, 14.0, 16.0, 18.0, 20.0, 25.0, 30.0, 40.0, 50.0} |

### G.2. Cost of SV Hyperparameter Tuning

We will show how our joint training scheme helps reduce amortized cost of SV training, supposing we use FSSVs. We first split the hyperparameter tuning cost into two components: *offline tuning* (i.e. preliminary hyperparameter tuning before large-scale SV training) and *online tuning* (i.e. hyperparameter tuning after large-scale SV training).

Let the number of fine-tuned SVs be $S$, the inference-time search grid be $\mathcal{A}_{\text{search}}$ ($|\mathcal{A}_{\text{search}}| = 14$ according to Table 7).

**Offline tuning.** Suppose that batch size is restricted by hardware and not tuned. Our joint training scheme of Algorithm 1 requires tuning five hyperparameters: training epochs, factor learning rate, factor initialization size, direction initialization size and direction learning rate. Meanwhile, fine-tuned SVs that use factor sampling require tuning four hyperparameters: training epochs, factor initialization size, factor sampling set, and direction learning rate. Let the cost of tuning a single hyperparameter (e.g., learning rate, initialization size) be $H$ runs (i.e. size of search grid). Then our joint training scheme requires additional $(H^5 - H^4)$ runs for searching training hyperparameters. We have $H \leq 4$ for AXBENCH evaluation (§L.1), thus our joint training scheme has an extra hyperparameter tuning cost as large as 768 runs.

**Online tuning.** At inference time, the factor sampling approach is accompanied by factor selection for each instance of fine-tuned SV, with overhead scaling linearly with the number of SVs: $S|\mathcal{A}_{\text{search}}|$. In contrast, our joint training scheme has zero inference-time overhead.

Overall, $S = 768/14 \approx 55$ SVs is sufficient for our joint training scheme to reach the same level of amortized overhead as factor sampling/selection; When $S > 55$, our joint training scheme would yield a lower amortized overhead than factor sampling/tuning.

## H. Review of SV Training and Inference Procedures

In this section, we provide a formal review of previous methods to train SVs and steer model generations with fine-tuned SVs.

**Factor selection at inference time (Algorithm 3).** At inference time, factor selection is conducted for each instance of SV on a development set of instructions based on overall steering score. The factor selection process essentially examines intervened model responses across a predesignated set of steering factors ($\mathcal{A}_{\text{search}}$). The selected optimal factor ($\alpha^*$) is the one that yields the highest average overall score. This approach has been used for optimization-free SVs and fine-tuned SVs alike (Turner et al., 2023; Rimsky et al., 2024; Chen et al., 2025b;a). Notably, under the settings we use in this paper, the neural network scaling theory does not lead to informed choices of steering factors for optimization-free SVs, since $\mathbf{v} = \Theta(1)$ is a constant and so is $\alpha$; thus factor selection is inevitable for optimization-free SVs.

**SV training with fixed steering factor.** Early SV training techniques use a fixed steering factor during training, usually $\alpha = 1$ (Subramani et al., 2022; Wu et al., 2025a). Then steering factors are selected at inference time via grid search. Based on previous analysis on AXBENCH, the resulting SVs usually have optimal inference-time steering factors of around 1.0 and underperform both prompting and fine-tuning methods (Wu et al., 2025a).

**SV training with factor sampling (Algorithm 2).** Wu et al. (2025b) recently introduce a novel approach for SV training that improves upon the SV training procedure with fixed steering factors. At training time, they employ a *factor sampling* strategy, which requires preparing a set of steering factors ($\mathcal{A}_{\text{sample}}$) before training, as is shown in Algorithm 2. The factor set is used universally across all concepts for a layer of a certain model, and the factors are curated in a manner such that the scaled vector norm $\|\alpha\mathbf{v}\|_2$ matches the typical layer norm of the intervention layer. At each training step, a steering factor is randomly sampled from $\mathcal{A}_{\text{sample}}$, which is used to steer the target model. In practice, this training procedure improves SV performance and decreases the variance in overall steering scores (Wu et al., 2025b).

**Steering factor sets in practice.** We show the actual steering factor sets for Algorithm 2 and Algorithm 3 in Table 7, which are copied from Wu et al. (2025b) for the convenience of readers.

---

**Algorithm 2** SV training procedure with factor sampling (Wu et al., 2025b).

---

**input** Training set $\mathcal{D}^c$, factor set $\mathcal{A}_{\text{sample}} = \{\alpha_i\}$, learning rate $\eta$, training steps $T$, loss function $\ell(\cdot)$
**output** Steering direction $\mathbf{v}_T$

    $\mathbf{v}_0 \sim \mathcal{N}(\mathbf{0}, n^{-1}\mathbf{I}_n)$    {Kaiming initialization}
    $t \leftarrow 0$
    **while** $t < T$ **do**
        $(\mathbf{x}, \mathbf{y}) \sim \mathcal{D}$
        $\alpha_t \sim \mathcal{A}_{\text{sample}}$    {Factor sampling}
        $l_t \leftarrow \ell\left(p_\Phi\left(\cdot|\mathbf{x}; \mathbf{h} \leftarrow \Phi(\mathbf{h}; \alpha_t, \mathbf{v}_t)\right), \mathbf{y}\right)$
        $g_t^{\mathbf{v}} \leftarrow \text{Adam}(\nabla_{\mathbf{v}} l_t)$    {Adam processes gradients}
        $\mathbf{v}_{t+1} \leftarrow \mathbf{v}_t - \eta g_t^{\mathbf{v}}$
        $t \leftarrow t + 1$
    **end while**

---

**Algorithm 3** SV factor selection procedure at inference time (Wu et al., 2025b).

---

**input** Concept $c$, development set of instructions $\mathcal{D}_{\text{dev}}$, factor search grid $\mathcal{A}_{\text{search}} = \{\alpha_i\}$, judge LLM $\mathcal{J}(\cdot)$, trained intervention $\Phi(\cdot)$
**output** Optimal steering factor $\alpha^*$

    $s \leftarrow \text{Array}(|\mathcal{A}_{\text{search}}|)$    {Initialize empty array of overall scores}
    **for** $\alpha_i \in \mathcal{A}_{\text{search}}$ **do**
        $s_i \leftarrow 0$
        **for** $\mathbf{x}_j \in \mathcal{D}_{\text{dev}}$ **do**
            $\hat{\mathbf{y}}_j^c \sim p_\Phi(\cdot|\mathbf{x}_j; \mathbf{h} \leftarrow \Phi)$    {Sample steered response from intervened model}
            $s_i \leftarrow s_i + \mathcal{J}(\mathbf{x}_j, \hat{\mathbf{y}}_j^c, c)$    {Judge evaluates intervened model response}
        **end for**
        $s_i \leftarrow s_i / |\mathcal{D}_{\text{dev}}^c|$
    **end for**
    $i^* \leftarrow \arg\max_i s_i$    {Select optimal steering factor with the highest overall score}
    $\alpha^* \leftarrow \alpha_{i^*}$

---

*Table 8.* Hyperparameters for verification experiment of §6.1. These hyperparameters are used for all three tested setups: $\mathcal{D}_{L10}^{G2B}$, $\mathcal{D}_{L20}^{G9B}$ and $\mathcal{D}_{L32}^{Q32B}$.

| Hyperparameter | FSSV | PrOSV |
|---|:---:|:---:|
| Seed | {42, 43, 44} | |
| Epochs | 6 | |
| Learning rate | 0.04 | |
| Batch size | 12 | |
| Optimizer | Adam | |
| Weight decay | 0.0 | |
| Warmup steps | 0 | |
| Temperature | 1.0 | |
| Generation length | 128 | |
| LLM judge temperature | 0.01 | |

# I. Disclosure of Computational Resources

We use $2\times$ Nvidia RTX A6000 (48 G) and $2\times$ A800 (80 G) GPUs for our experiments. We also load model weights with `bfloat16` precision. Training a single SV on Gemma2-2B with Lang. objective takes around 1.5 minutes, while training an SV on Gemma2-2B with SimPO objective can take as long as 8 minutes.

# J. Details and Additional Results for Verification Experiment

In this section we disclose details of our empirical verification in §6.1.

## J.1. Experiment Details

**Hyperparameters.** We show hyperparameters in Table 8. These hyperparameters are used for SVs on all three models: Gemma2-2B, Gemma2-9B and Qwen2.5-32B. We use seeds to control random number generators for ordering of mini-batches, initialization and inference-time decoding. We also use a temperature of 0.01 for LLM judge to improve reproducibility of evaluation.

**Chat templates.** One important difference between Qwen2.5 and Gemma2 models is that, the former supports system prompt while the latter does not. We directly use the default system prompt for Qwen2.5-32B.

## J.2. Additional Results

**Concept scores.** We show heatmaps of concept scores in Figure 5 and Figure 6.

**Overall steering scores.** We show heatmaps of overall scores in Figure 7 and Figure 8.

**Overall score breakdown.** Since overall steering score is the harmonic mean of concept/instruct/fluency scores, it is meaningful to visualize these individual scores. We show breakdown of overall scores in:

- $\mathcal{D}_{L10}^{G2B}$; FSSV; $\lambda = 1$: Figure 9;
- $\mathcal{D}_{L10}^{G2B}$; FSSV; $\lambda = 8$: Figure 10;
- $\mathcal{D}_{L20}^{G9B}$; FSSV; $\lambda = 1$: Figure 13;
- $\mathcal{D}_{L20}^{G9B}$; FSSV; $\lambda = 8$: Figure 14;
- $\mathcal{D}_{L32}^{Q32B}$; FSSV; $\lambda = 1$: Figure 17;
- $\mathcal{D}_{L32}^{Q32B}$; FSSV; $\lambda = 8$: Figure 18;
- $\mathcal{D}_{L10}^{G2B}$; PrOSV ($p4+s4$); $\lambda = 1$: Figure 11;
- $\mathcal{D}_{L10}^{G2B}$; PrOSV ($p4+s4$); $\lambda = 8$: Figure 12;
- $\mathcal{D}_{L20}^{G9B}$; PrOSV ($p4+s4$); $\lambda = 1$: Figure 19;
- $\mathcal{D}_{L20}^{G9B}$; PrOSV ($p4+s4$); $\lambda = 8$: Figure 20;
- $\mathcal{D}_{L32}^{Q32B}$; PrOSV ($p2+s2$); $\lambda = 1$: Figure 19;

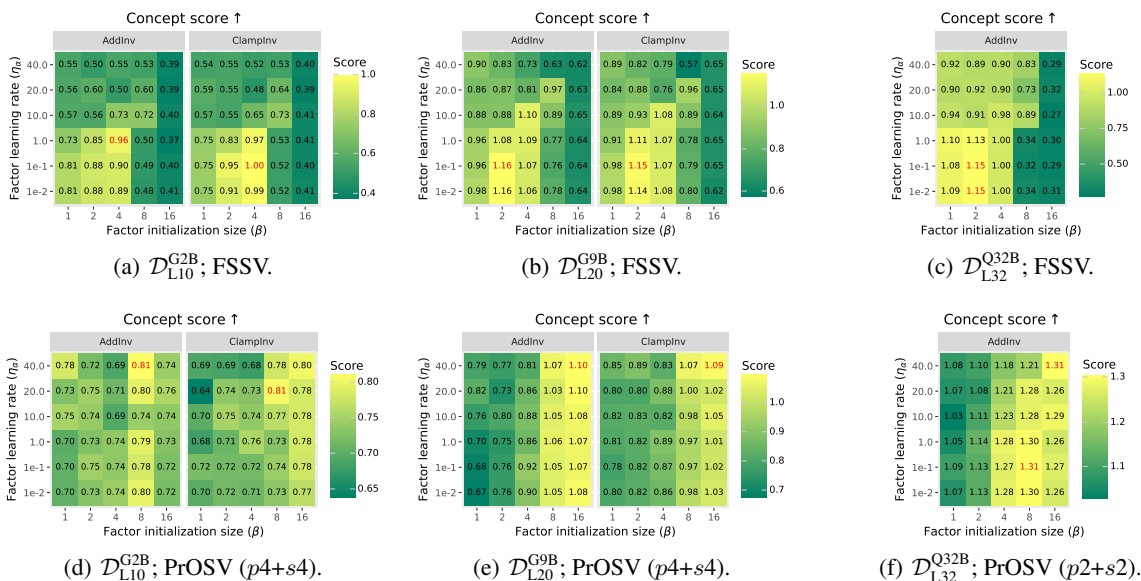

*Figure 5.* Visualization of concept scores using joint training scheme when initializing steering directions with direction initialization size $\lambda = 1$. Highest scores are highlighted in red.

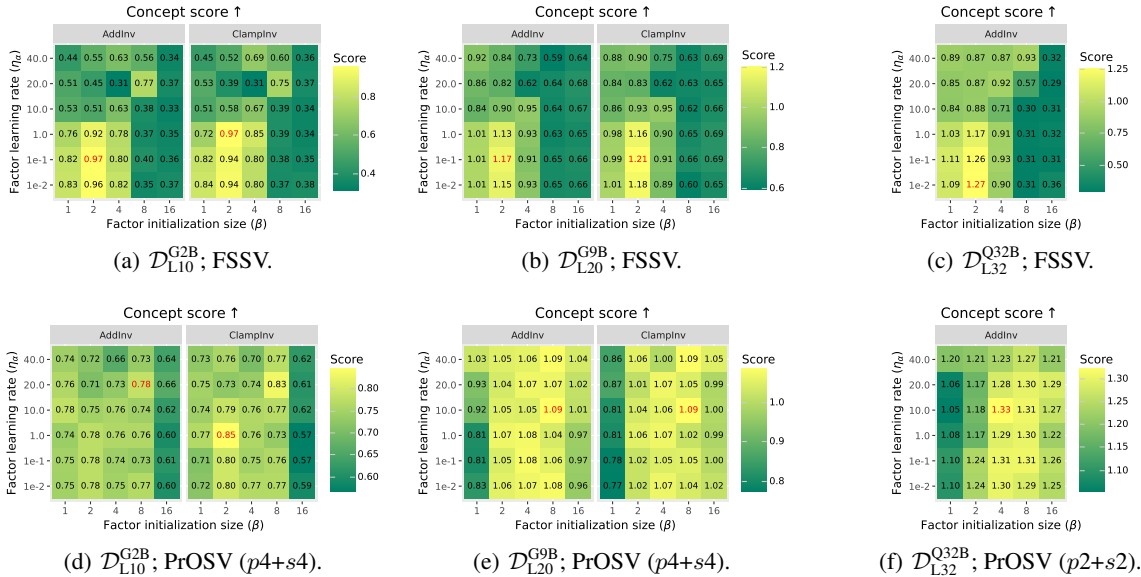

*Figure 6.* Visualization of concept scores using joint training scheme when initializing steering directions with direction initialization size $\lambda = 8$. Highest scores are highlighted in red.

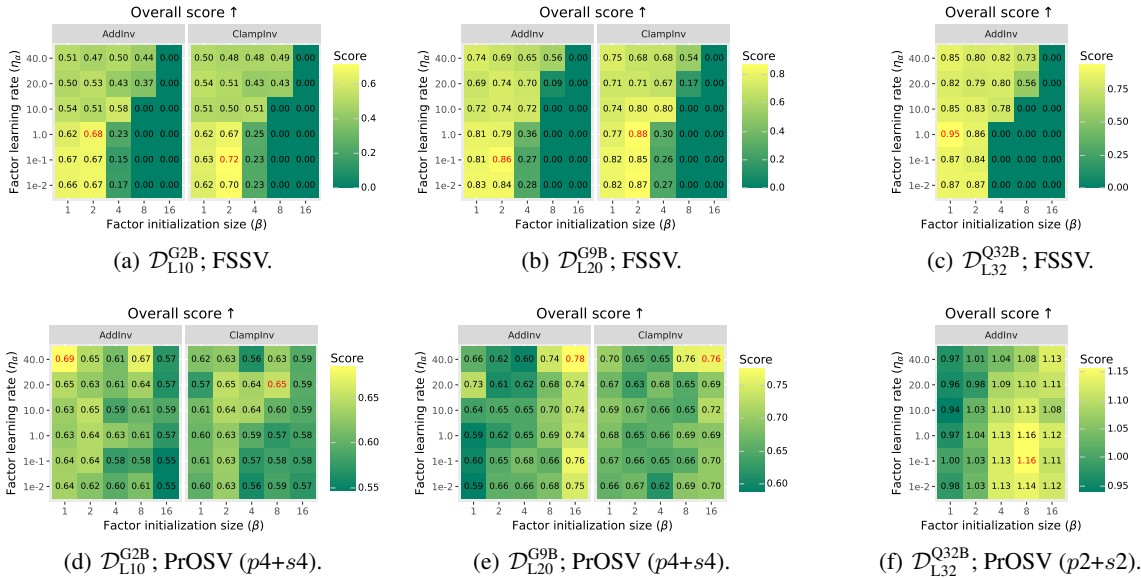

*Figure 7.* Visualization of overall steering scores using joint training scheme with direction initialization size $\lambda = 1$. Highest scores are highlighted in red.

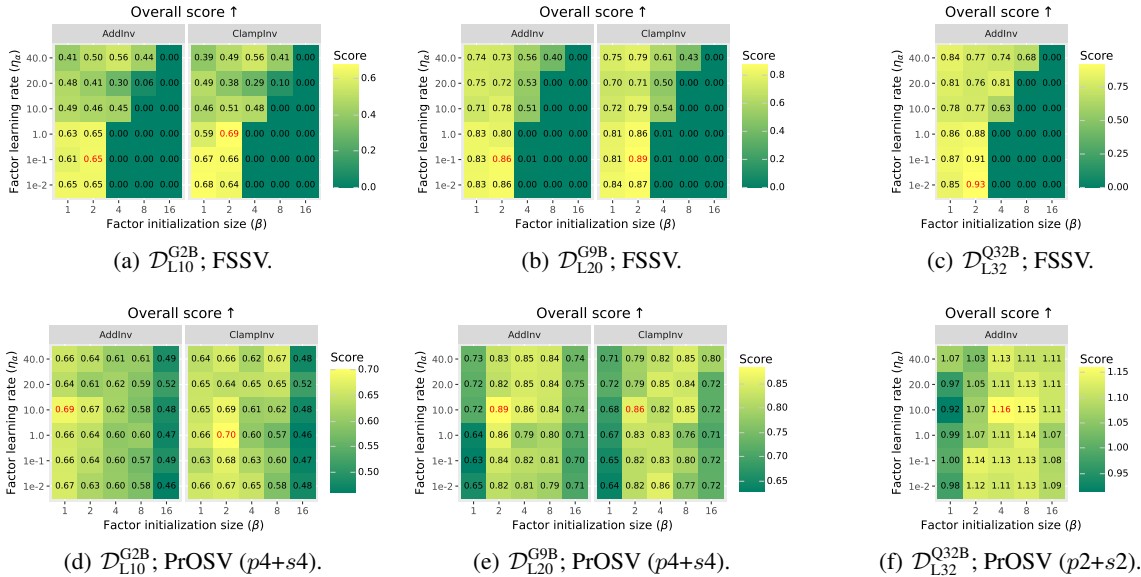

*Figure 8.* Visualization of overall steering scores using joint training scheme with direction initialization size $\lambda = 8$. Highest scores are highlighted in red.

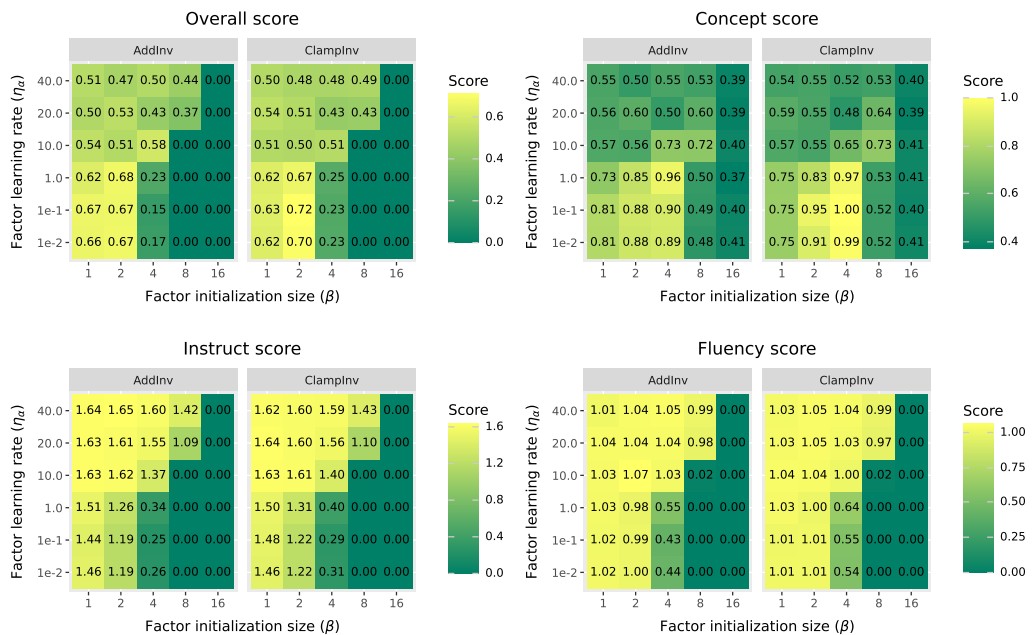

*Figure 9.* Breakdown of overall steering scores with setup: $\mathcal{D}_{L10}^{G2B}$; FSSV; $\lambda = 1$.

- $\mathcal{D}_{L32}^{Q32B}$; PrOSV ($p2+s2$); $\lambda = 8$: Figure 20.

**Standard deviation of scores.** Wehner et al. (2025) have proposed best practices of evaluating representation steering methods, where they emphasize the importance of reporting variances of results so that "readers see whether a method provides reliable steerability". Therefore we show standard deviation of overall/concept/instruct/fluency scores across random seeds in the following figures:

- $\mathcal{D}_{L10}^{G2B}$, FSSV, $\lambda = 1$: Figure 21;
- $\mathcal{D}_{L10}^{G2B}$, FSSV, $\lambda = 8$: Figure 22,
- $\mathcal{D}_{L10}^{G2B}$, PrOSV ($p4+s4$), $\lambda = 1$: Figure 27;
- $\mathcal{D}_{L10}^{G2B}$, PrOSV ($p4+s4$), $\lambda = 8$: Figure 28;
- $\mathcal{D}_{L20}^{G9B}$, FSSV, $\lambda = 1$: Figure 23;
- $\mathcal{D}_{L20}^{G9B}$, FSSV, $\lambda = 8$: Figure 24;
- $\mathcal{D}_{L20}^{G9B}$, PrOSV ($p4+s4$), $\lambda = 1$: Figure 29;
- $\mathcal{D}_{L20}^{G9B}$, PrOSV ($p4+s4$), $\lambda = 8$: Figure 30;
- $\mathcal{D}_{L32}^{Q32B}$, FSSV, $\lambda = 1$: Figure 25;
- $\mathcal{D}_{L32}^{Q32B}$, FSSV, $\lambda = 8$: Figure 26;
- $\mathcal{D}_{L32}^{Q32B}$, PrOSV ($p2+s2$), $\lambda = 1$: Figure 31;
- $\mathcal{D}_{L32}^{Q32B}$, PrOSV ($p2+s2$), $\lambda = 8$: Figure 32.

## K. Details and Additional Results for Effect of Intervention Locations

### K.1. Experiment Details

**Hyperparameters.** The training hyperparameters, including learning rates and initialization sizes are the same as §6.1. Since the objective is to investigate the effect of intervention locations, we only vary this component. Specifically, we use the following setups with different computational budgets: (1) FSSV; (2) PrOSV with full-prompt interventions where $|\mathcal{I}|$ is dynamic; (3) PrOSV with fixed computational budgets: $|\mathcal{I}| = 2, 4, 8$. For PrOSV with fixed computational budgets, we

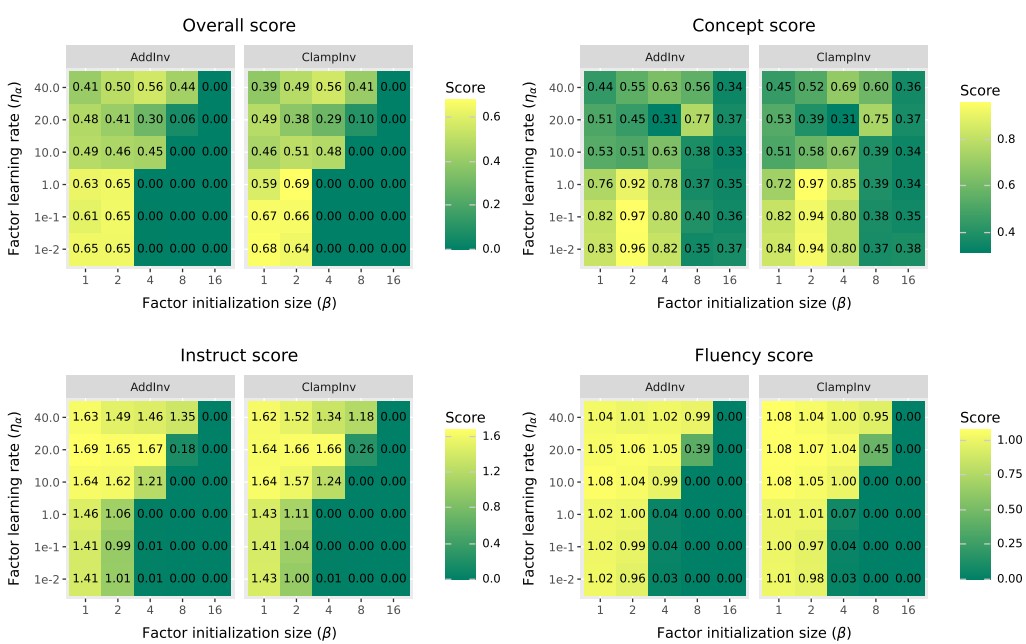

*Figure 10.* Breakdown of overall steering scores with setup: $\mathcal{D}_{\text{L10}}^{\text{G2B}}$; FSSV; $\lambda = 8$.

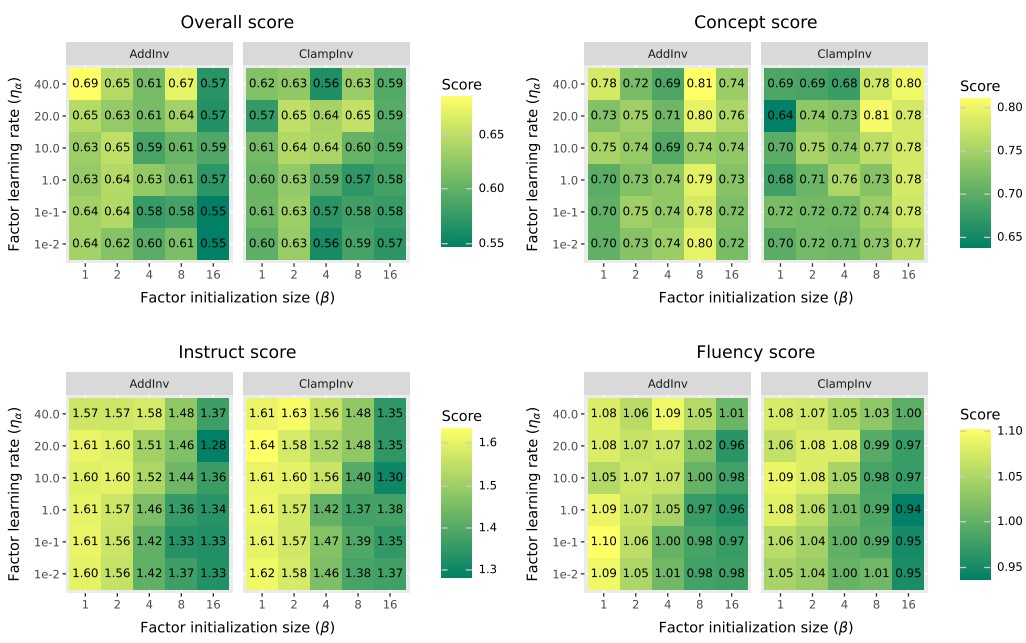

*Figure 11.* Breakdown of overall steering scores with setup: $\mathcal{D}_{\text{L10}}^{\text{G2B}}$; PrOSV ($p4+s4$); $\lambda = 1$.

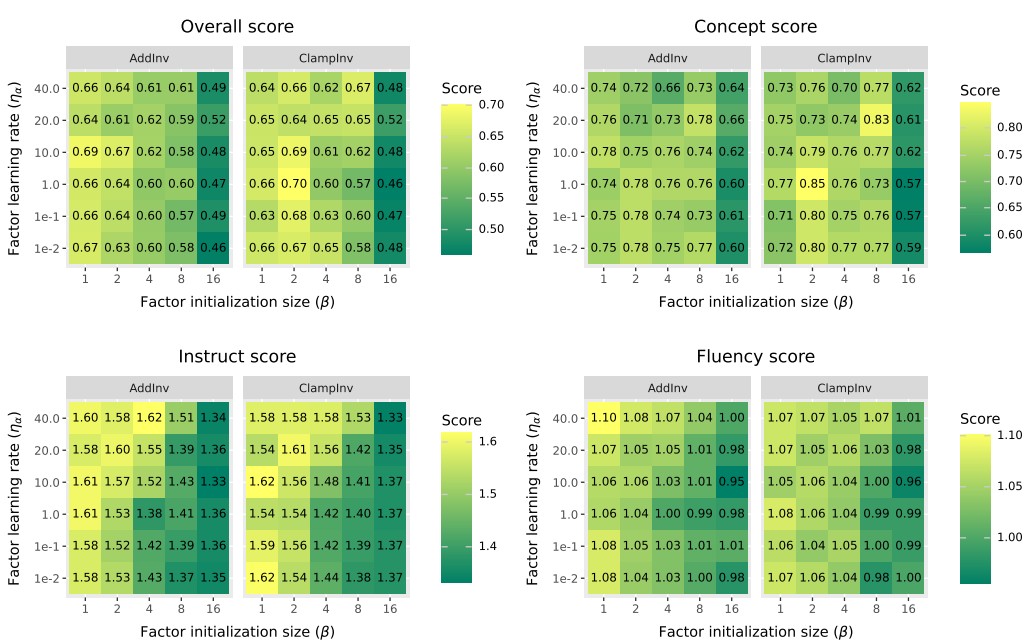

*Figure 12.* Breakdown of overall steering scores with setup: $\mathcal{D}_{L10}^{G2B}$; PrOSV ($p4+s4$); $\lambda = 8$.

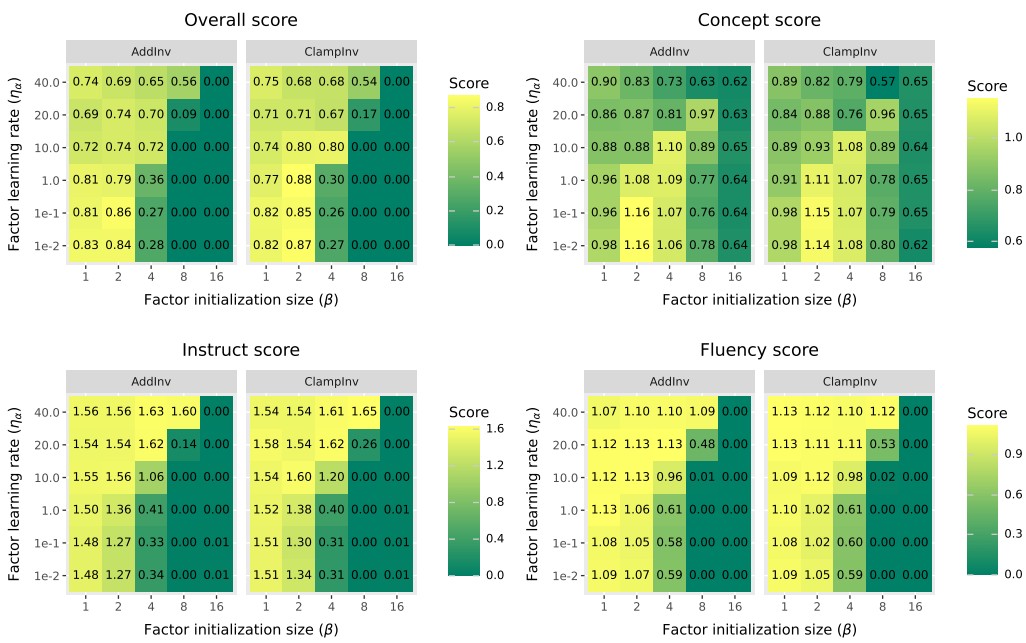

*Figure 13.* Breakdown of overall steering scores with setup: $\mathcal{D}_{L20}^{G9B}$; FSSV; $\lambda = 1$.

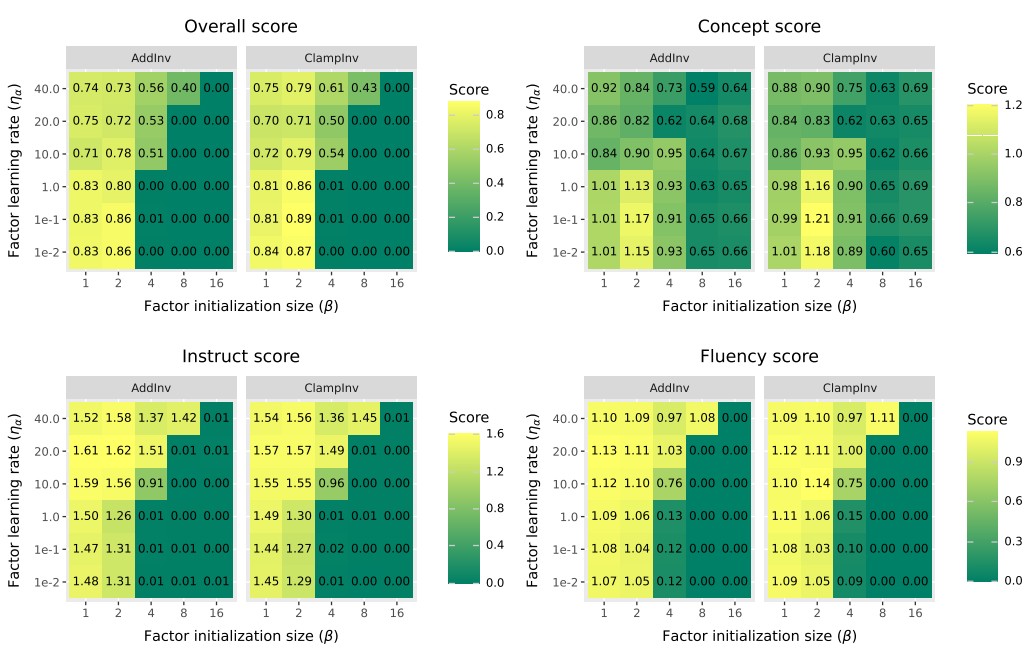

*Figure 14.* Breakdown of overall steering scores with setup: $\mathcal{D}_{L20}^{G9B}$; FSSV; $\lambda = 8$.

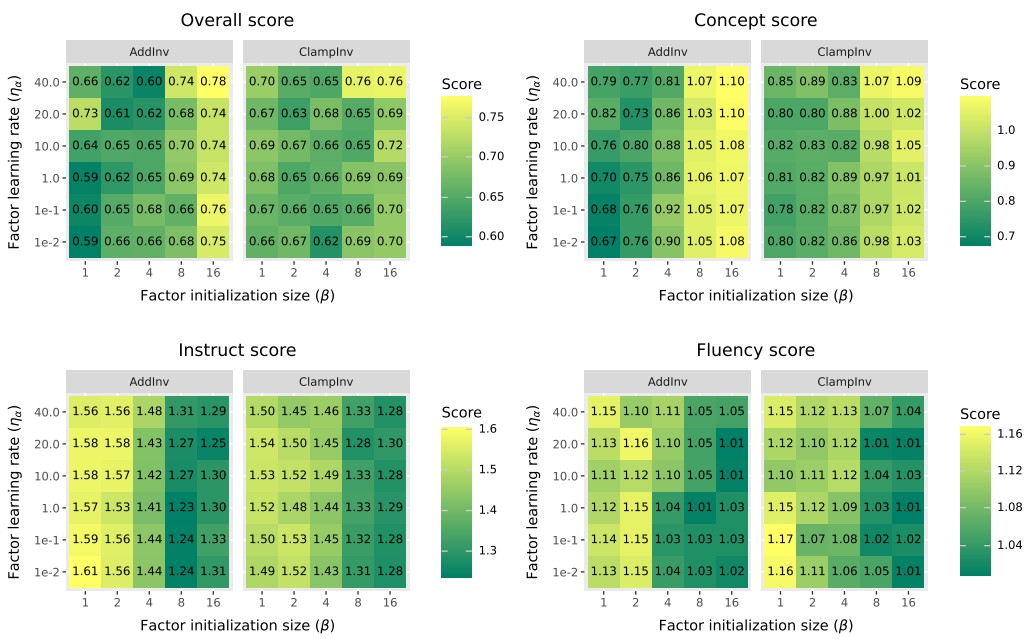

*Figure 15.* Breakdown of overall steering scores with setup: $\mathcal{D}_{L20}^{G9B}$; PrOSV ($p4+s4$); $\lambda = 1$.

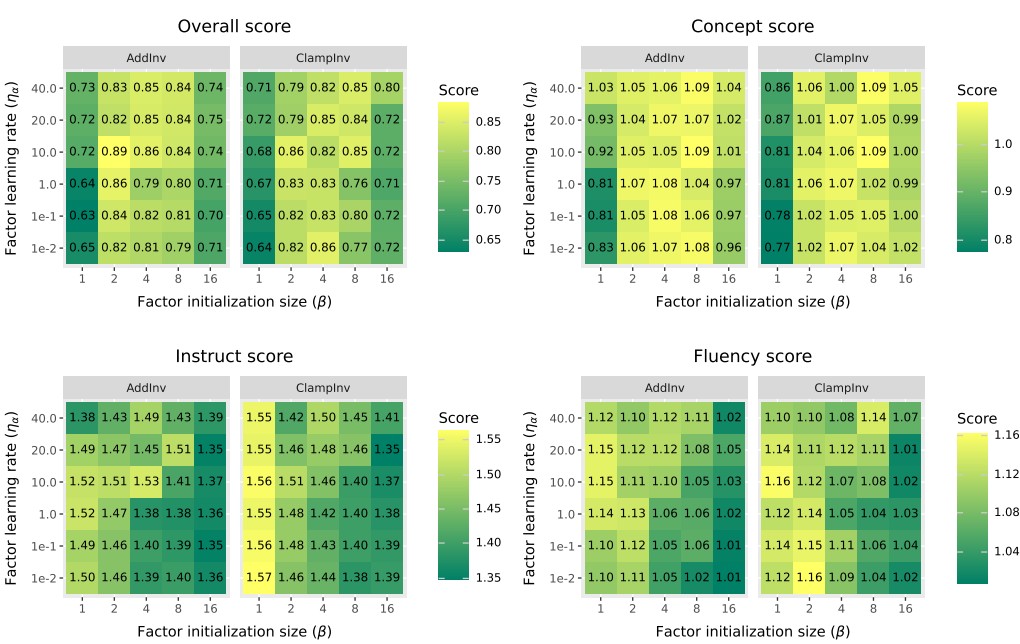

*Figure 16.* Breakdown of overall steering scores with setup: $\mathcal{D}_{\text{L20}}^{\text{G9B}}$; PrOSV ($p4+s4$); $\lambda = 8$.

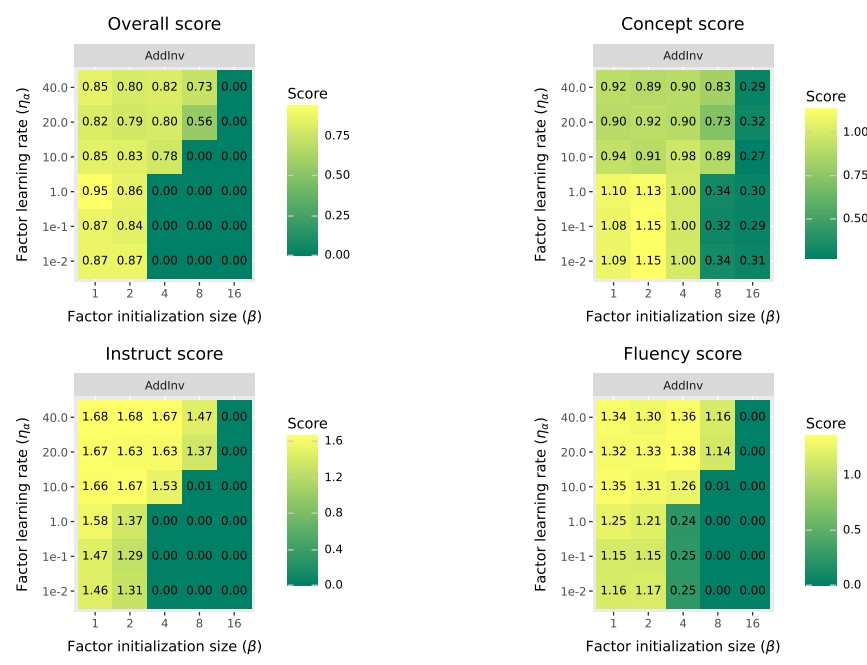

*Figure 17.* Breakdown of overall steering scores with setup: $\mathcal{D}_{\text{L32}}^{\text{Q32B}}$; FSSV; $\lambda = 1$.

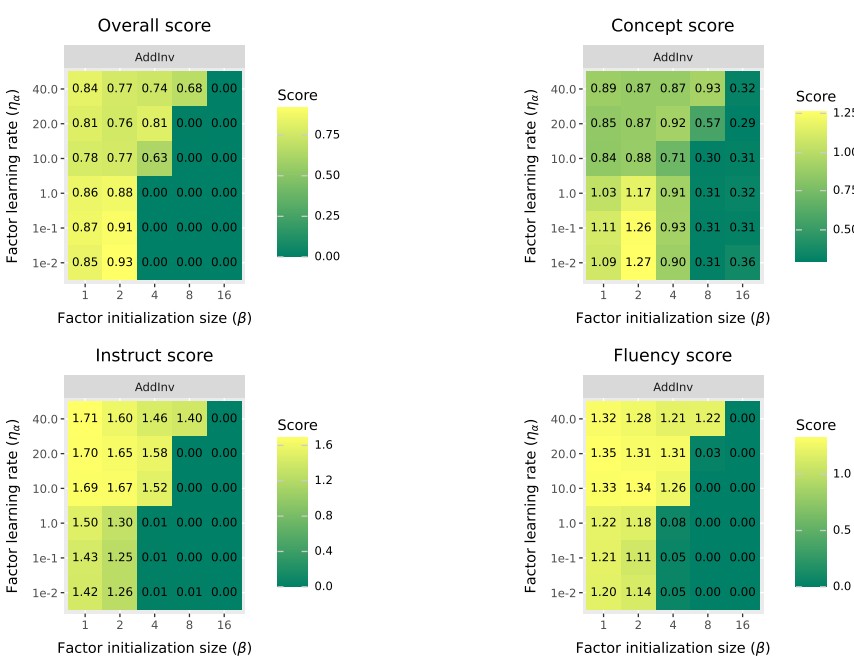

*Figure 18.* Breakdown of overall steering scores with setup: $\mathcal{D}_{\text{L32}}^{\text{Q32B}}$; FSSV; $\lambda = 8$.

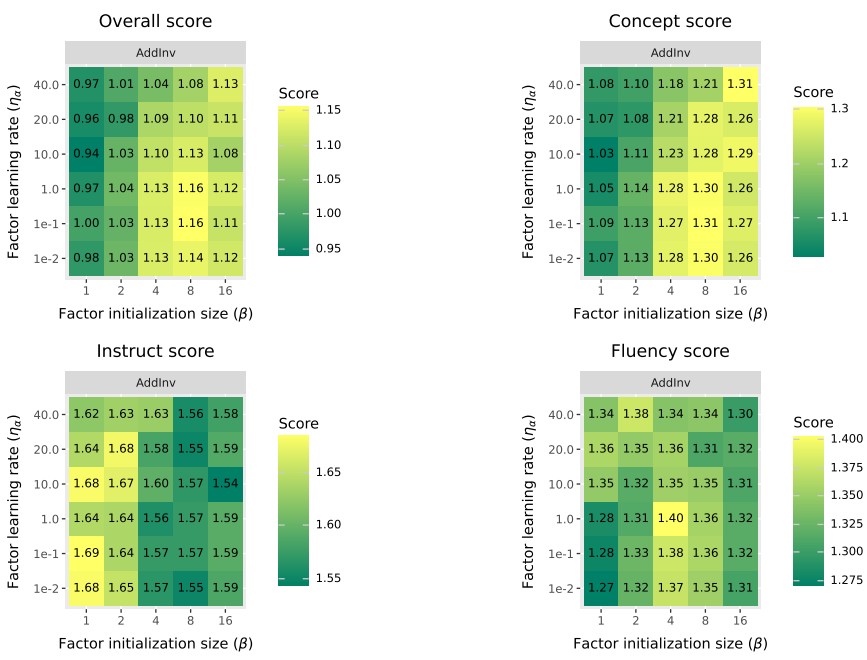

*Figure 19.* Breakdown of overall steering scores with setup: $\mathcal{D}_{\text{L32}}^{\text{Q32B}}$; PrOSV ($p2+s2$); $\lambda = 1$.

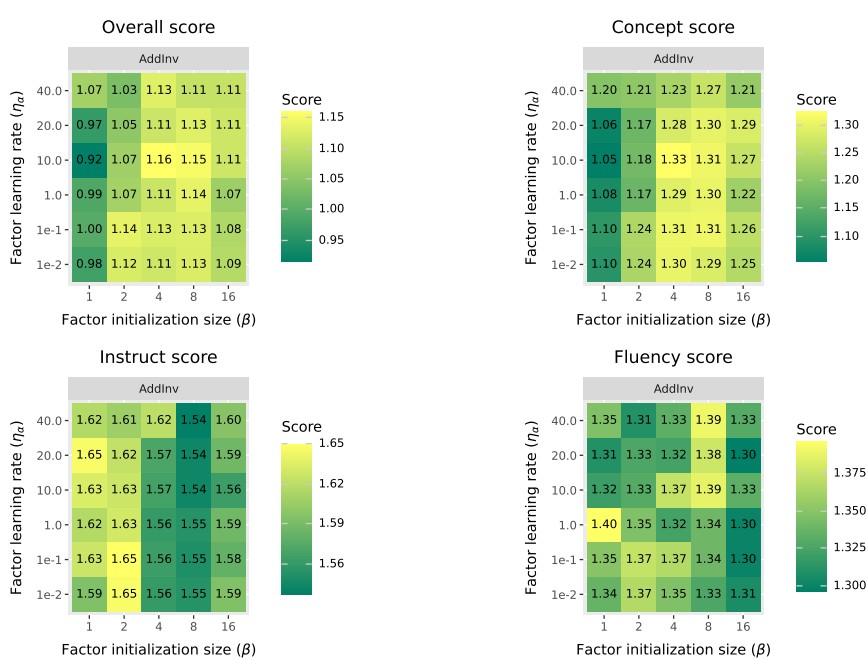

*Figure 20.* Breakdown of overall steering scores with setup: $\mathcal{D}_{L32}^{Q32B}$; PrOSV ($p2+s2$); $\lambda = 8$.

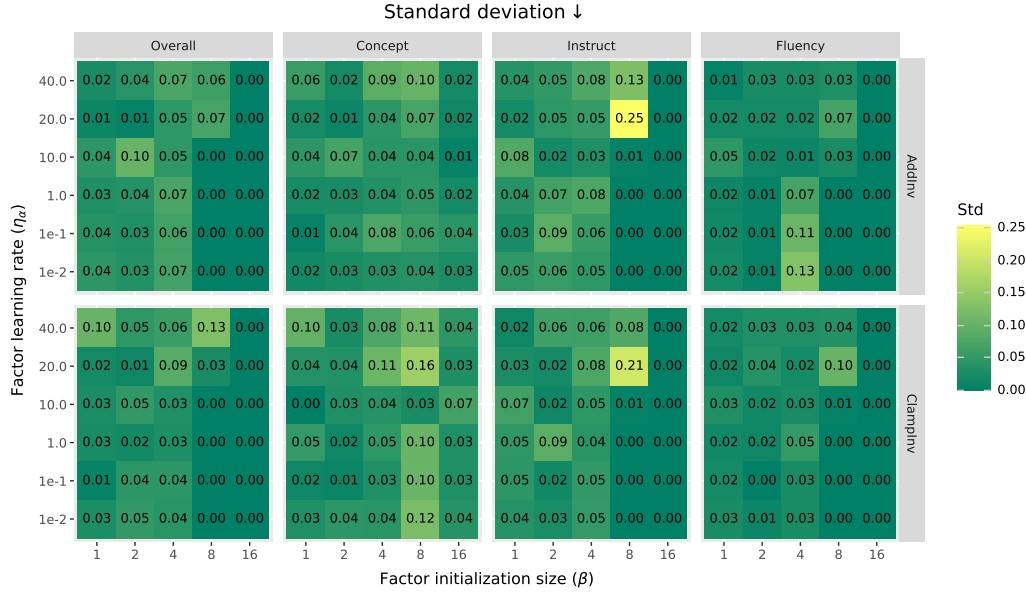

*Figure 21.* Standard deviation of individual scores across seeds with setup: $\mathcal{D}_{L10}^{G2B}$; FSSV; $\lambda = 1$.

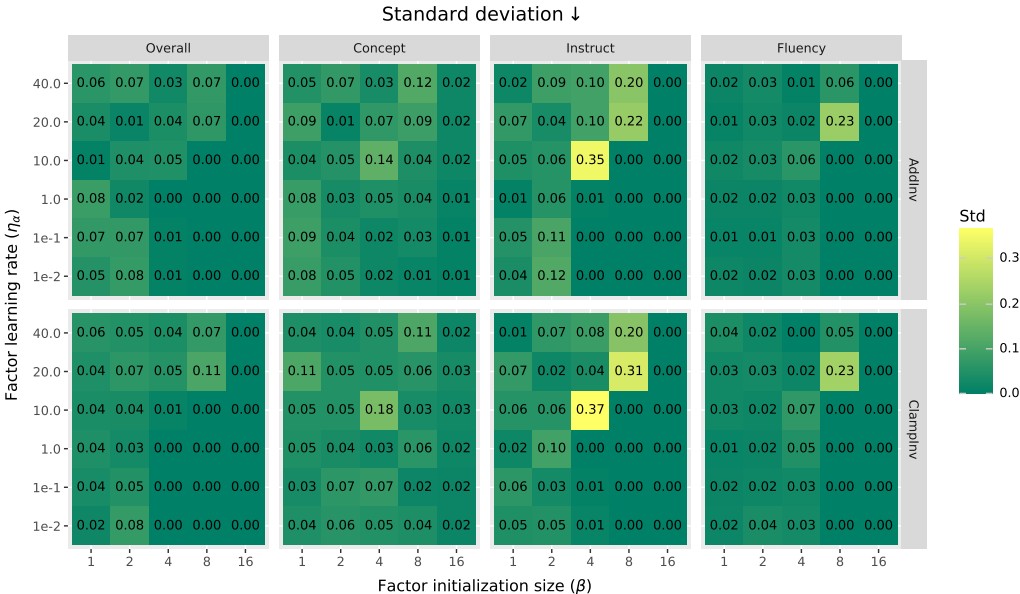

*Figure 22.* Standard deviation of individual scores across seeds with setup: $\mathcal{D}_{\text{L10}}^{\text{G2B}}$; FSSV; $\lambda = 8$.

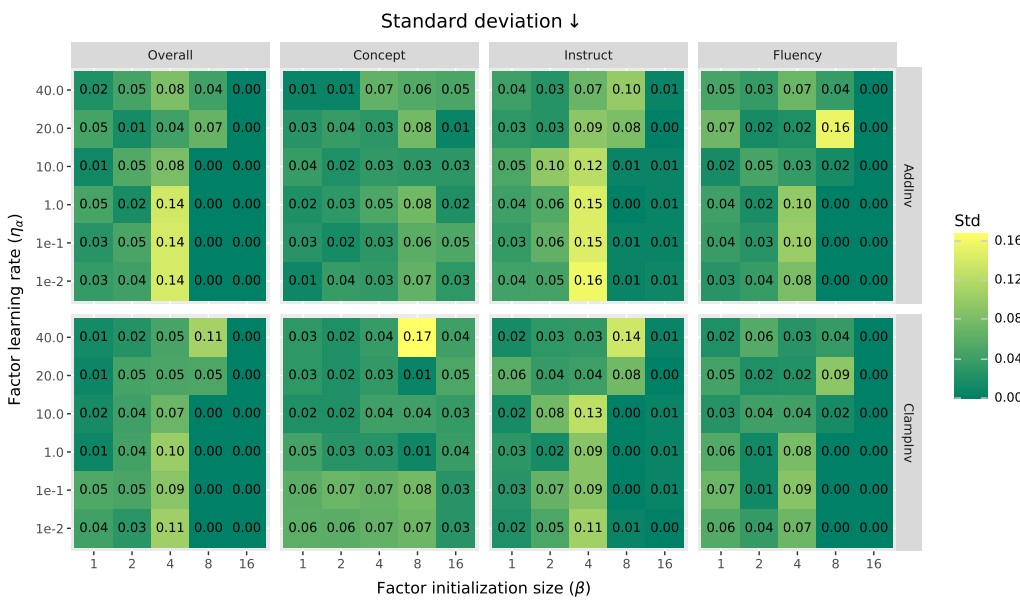

*Figure 23.* Standard deviation of individual scores across seeds with setup: $\mathcal{D}_{\text{L20}}^{\text{G9B}}$; FSSV; $\lambda = 1$.

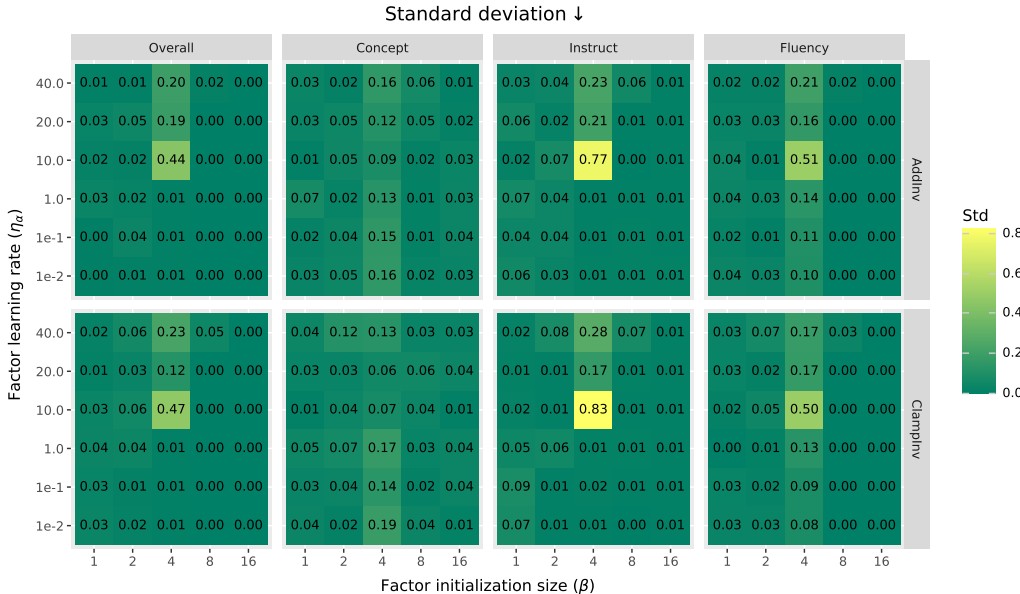

*Figure 24.* Standard deviation of individual scores across seeds with setup: $\mathcal{D}_{\text{L20}}^{\text{G9B}}$; FSSV; $\lambda = 8$.

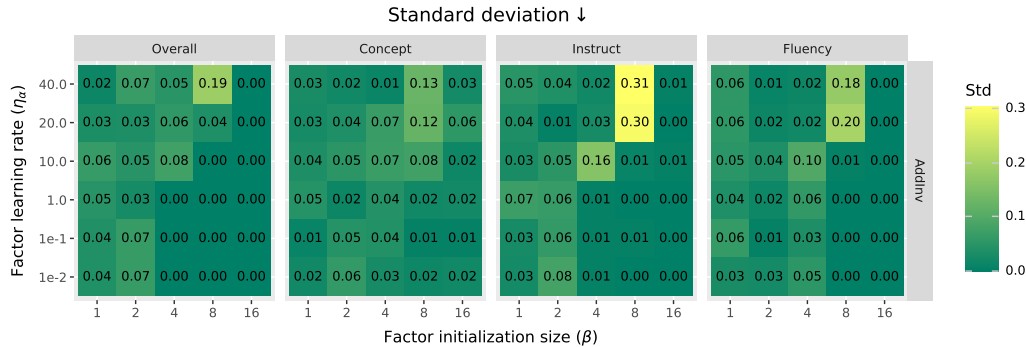

*Figure 25.* Standard deviation of individual scores across seeds with setup: $\mathcal{D}_{\text{L32}}^{\text{Q32B}}$; FSSV; $\lambda = 1$.

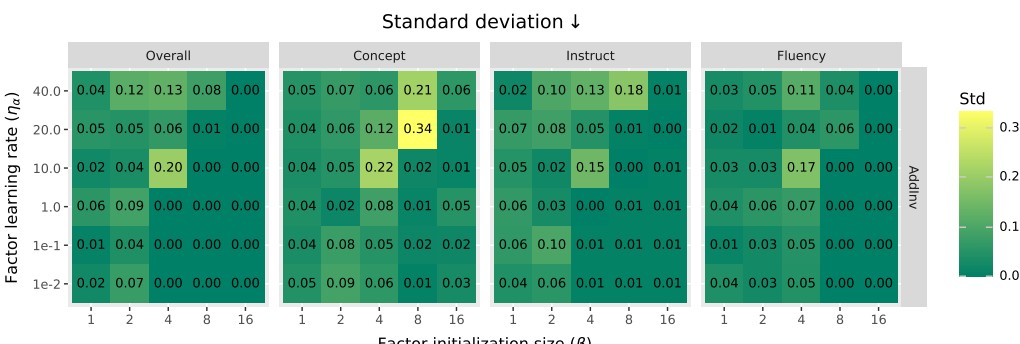

*Figure 26.* Standard deviation of individual scores across seeds with setup: $\mathcal{D}_{\text{L32}}^{\text{Q32B}}$; FSSV; $\lambda = 8$.

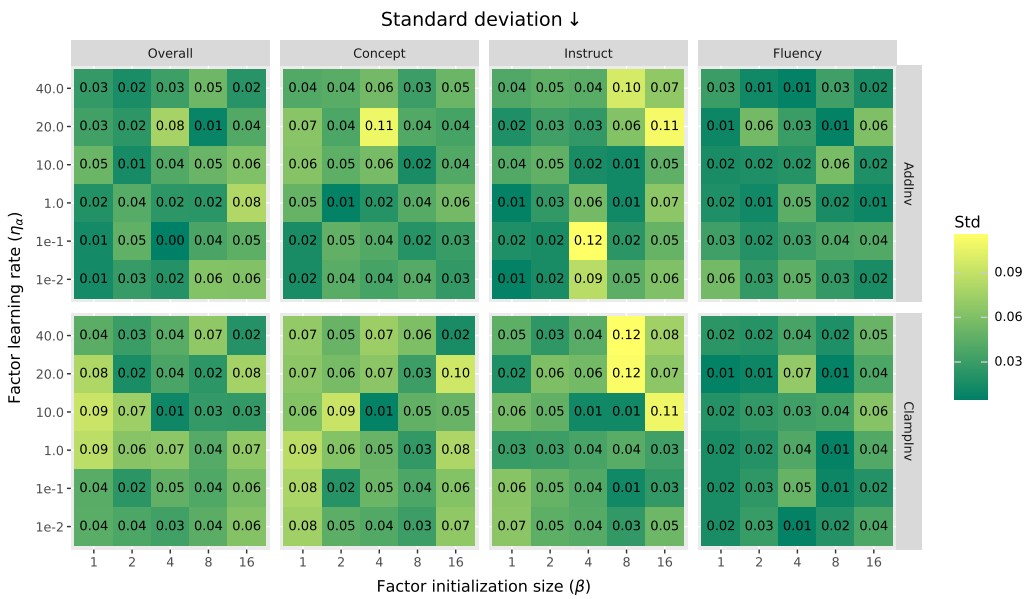

*Figure 27.* Standard deviation of individual scores across seeds with setup: $\mathcal{D}_{\text{L10}}^{\text{G2B}}$; PrOSV ($p4+s4$); $\lambda = 1$.

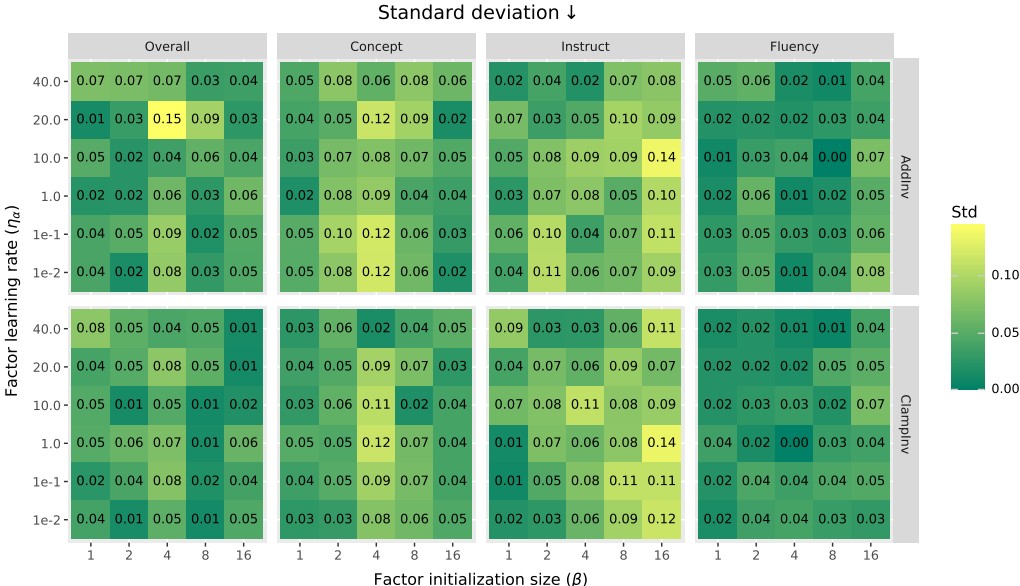

*Figure 28.* Standard deviation of individual scores across seeds with setup: $\mathcal{D}_{\text{L10}}^{\text{G2B}}$; PrOSV ($p4+s4$); $\lambda = 8$.

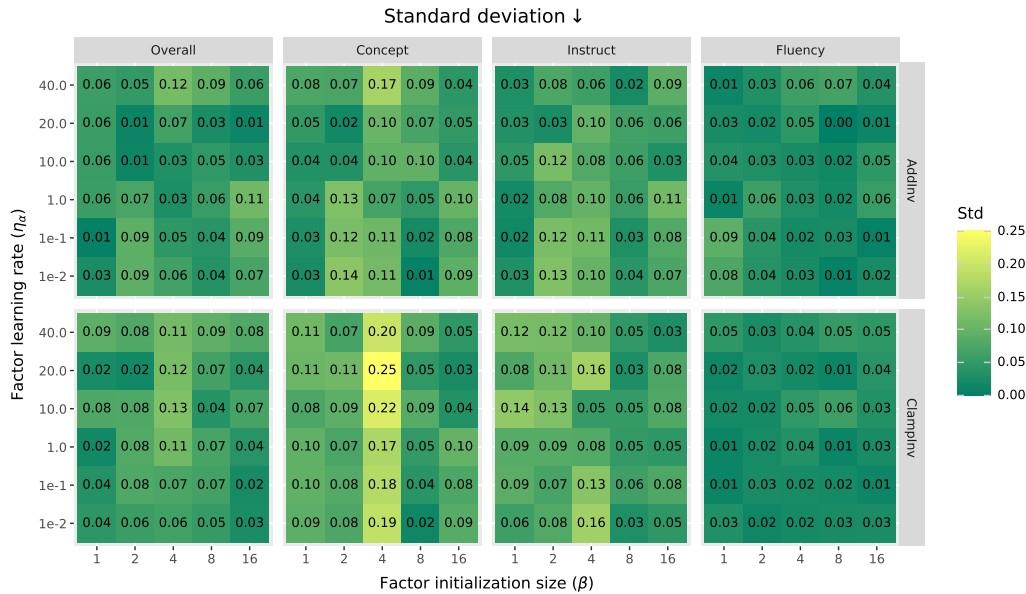

*Figure 29.* Standard deviation of individual scores across seeds with setup: $\mathcal{D}_{\text{L20}}^{\text{G9B}}$; PrOSV ($p4+s4$); $\lambda = 1$.

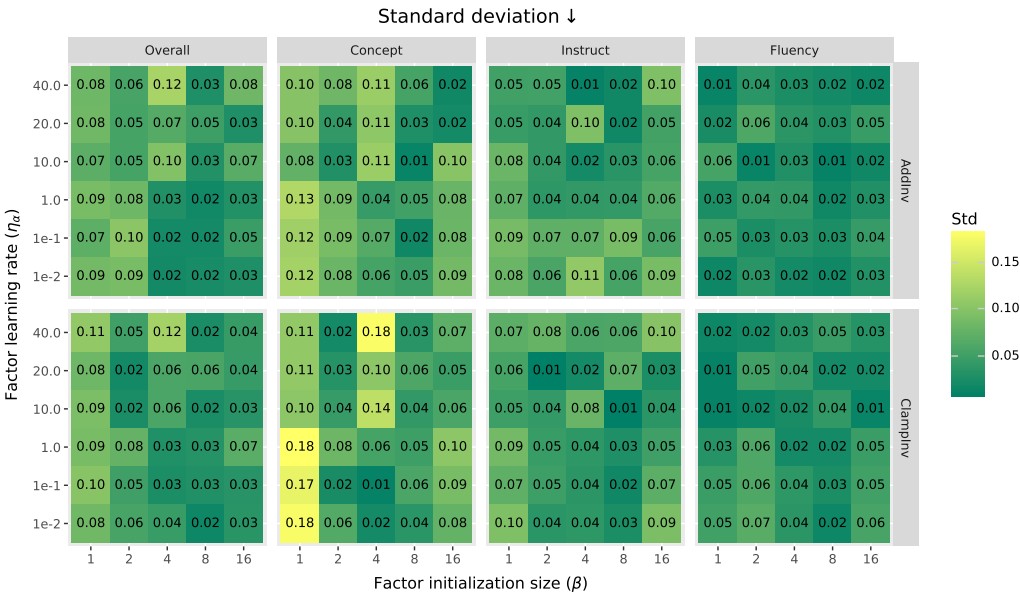

*Figure 30.* Standard deviation of individual scores across seeds with setup: $\mathcal{D}_{\text{L20}}^{\text{G9B}}$; PrOSV ($p4+s4$); $\lambda = 8$.

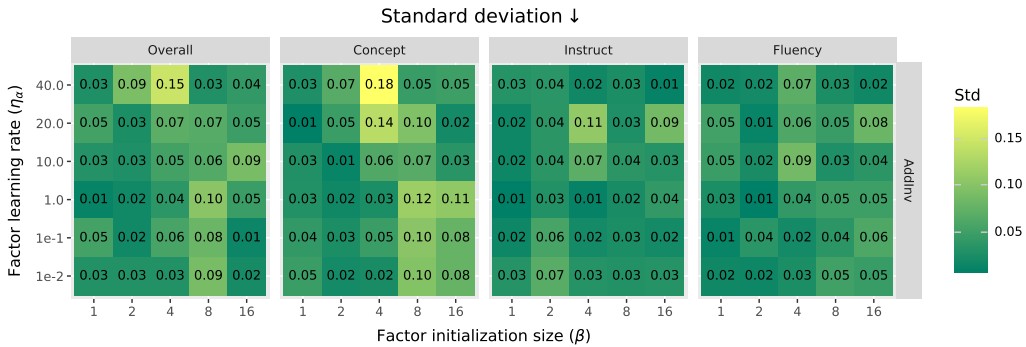

*Figure 31.* Standard deviation of individual scores across seeds with setup: $\mathcal{D}_{\text{L32}}^{\text{Q32B}}$; PrOSV ($p2+s2$); $\lambda = 1$.

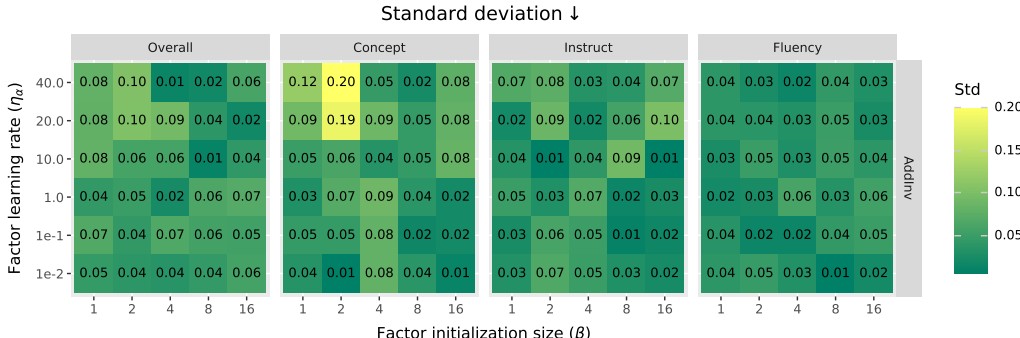

*Figure 32.* Standard deviation of individual scores across seeds with setup: $\mathcal{D}_{\mathrm{L32}}^{\mathrm{Q32B}}$; PrOSV ($p2+s2$); $\lambda = 8$.

do not use budgets with more than 8 tokens, since 8 is approximately half of the average prompt length of CONCEPT500 dataset (17–21 tokens according to §O).

### K.2. Additional Results

**Concepts score vs. instruct scores.** In the main body, we show overall scores and concept scores in Table 1. For better readability, we additionally visualize concept scores and instruct scores in Figure 33. We only visualize concept/instruct scores since fluency scores do not vary significantly for optimal SVs (Figure 41). Overall, $p2+s2$ yields the best tradeoff between concept incorporation and instruction following on all three models, while FSSV resides on a worse Pareto frontier.

**Heatmap for PrOSV with various computational budgets and intervention locations.** In the main body, we only show the highest scores; here we show full heatmaps of overall scores:

- PrOSV, full-prompt intervention: Figure 34;
- PrOSV, $|\mathcal{I}| = 2$: Figure 35;
- PrOSV, $|\mathcal{I}| = 4$: Figure 36;
- PrOSV, $|\mathcal{I}| = 8$: Figure 37.

**Examples of intervened generations.** We show several SV-intervened model responses in Figure 38 to help readers understand the actual effects of SV interventions on concept-steering task.

**Discussions on effect of SV intervention locations.** Here we extend our analysis of the results of Table 1 in §6.2 by presenting the following hypotheses. These hypotheses are meant to help readers understand how PrOSV works.

(1) Concept incorporation might not require interventions on response tokens. This hypothesis is supported by the fact that full-prompt PrOSV can outperform FSSV in highest concept scores. This finding is counter-intuitive, and it might stem from the mechanistic differences between full-prompt PrOSV and FSSV. On one hand, full-prompt intervention might be understood as mainly editing the KV cache of the entire prompt, and optimizing for concept incorporation might shift the self-attention to focus on concept incorporation and to attend less to the instruction following task itself. On the other hand, FSSV might not rely solely on the self-attention mechanism and could be understood as adding a bias term to transformer layers. This leaves room to achieve both concept incorporation and instruction following.

(2) For prompt-only interventions, concept scores have a *partial* tendency to scale with the number of intervened tokens. This hypothesis is supported by the fact that PrOSVs with $|\mathcal{I}| = 4$ generally outperform those with $|\mathcal{I}| = 2$ and that

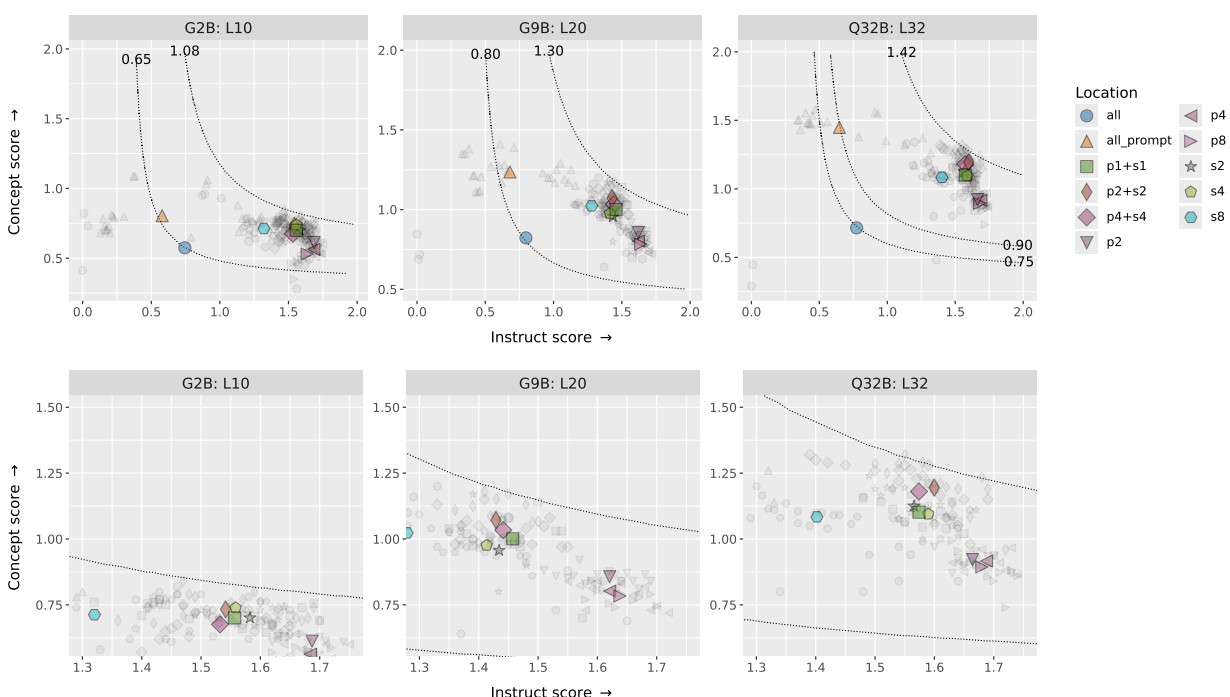

*Figure 33.* Concept score vs. instruct score. The first row shows the full figure while the second row shows a local zoomed view. The dotted lines denote data points with the same harmonic mean of concept score and instruct score. Gray data points denote the results of a single hyperparameter setup averaged across ten concepts and three random seeds, while colored data points denote average results of over all hyperparameter search grids. Overall, $p2+s2$ yields the best tradeoff between concept score and instruct score while FSSV ("all" in the figure) yields the worst tradeoff.

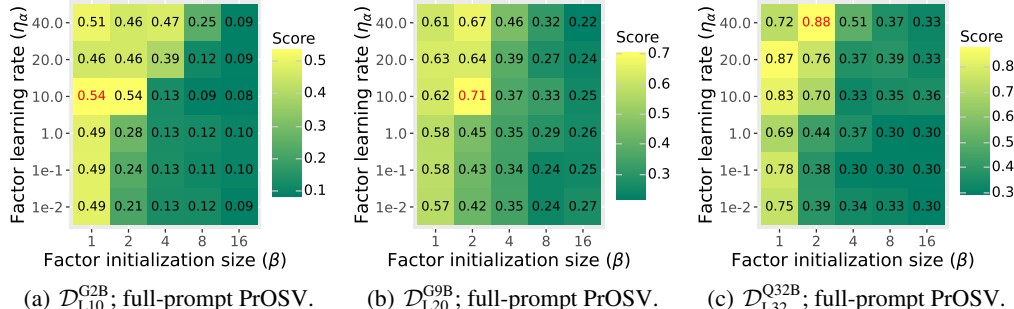

(a) $\mathcal{D}_{L10}^{G2B}$; full-prompt PrOSV.  (b) $\mathcal{D}_{L20}^{G9B}$; full-prompt PrOSV.  (c) $\mathcal{D}_{L32}^{Q32B}$; full-prompt PrOSV.

*Figure 34.* Overall scores with full-prompt PrOSV; $\lambda = 8$.

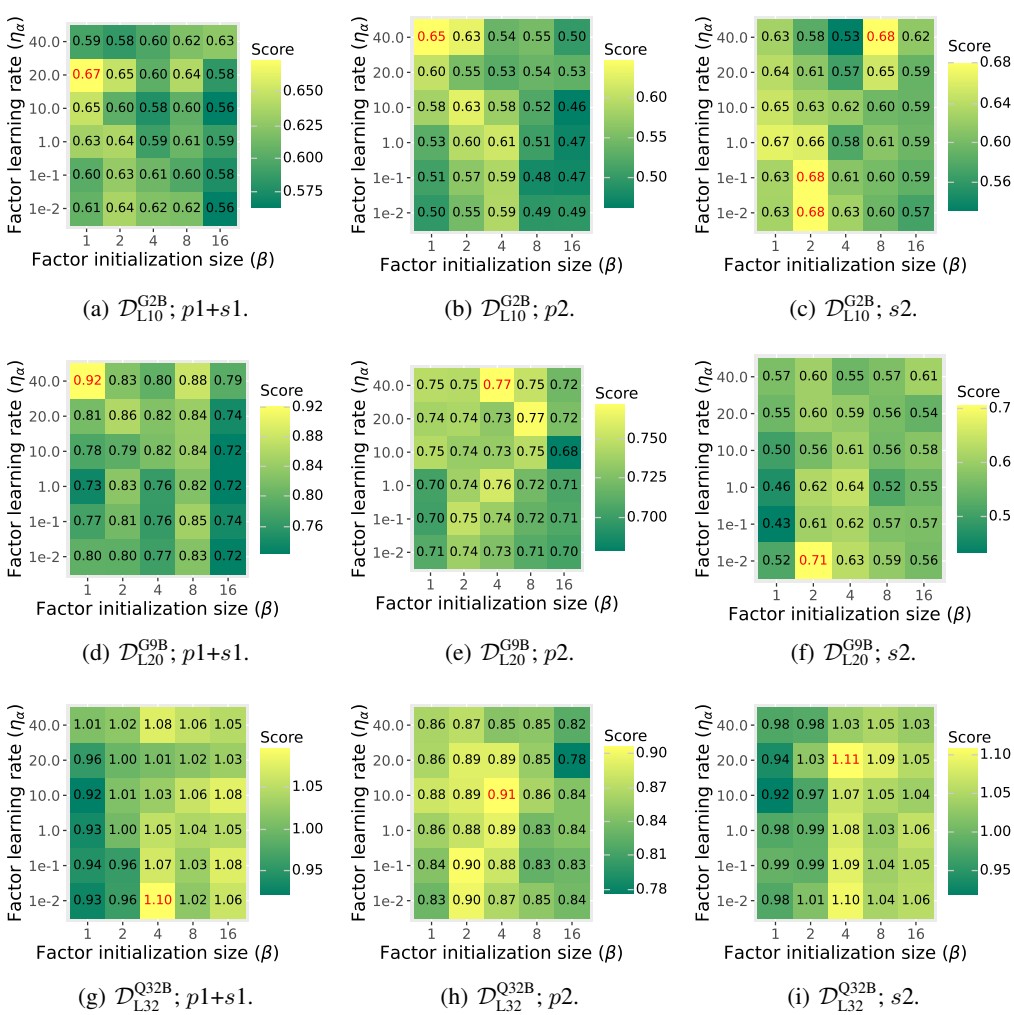

*Figure 35.* Overall scores with $|\mathcal{I}| = 2$; $\lambda = 8$.

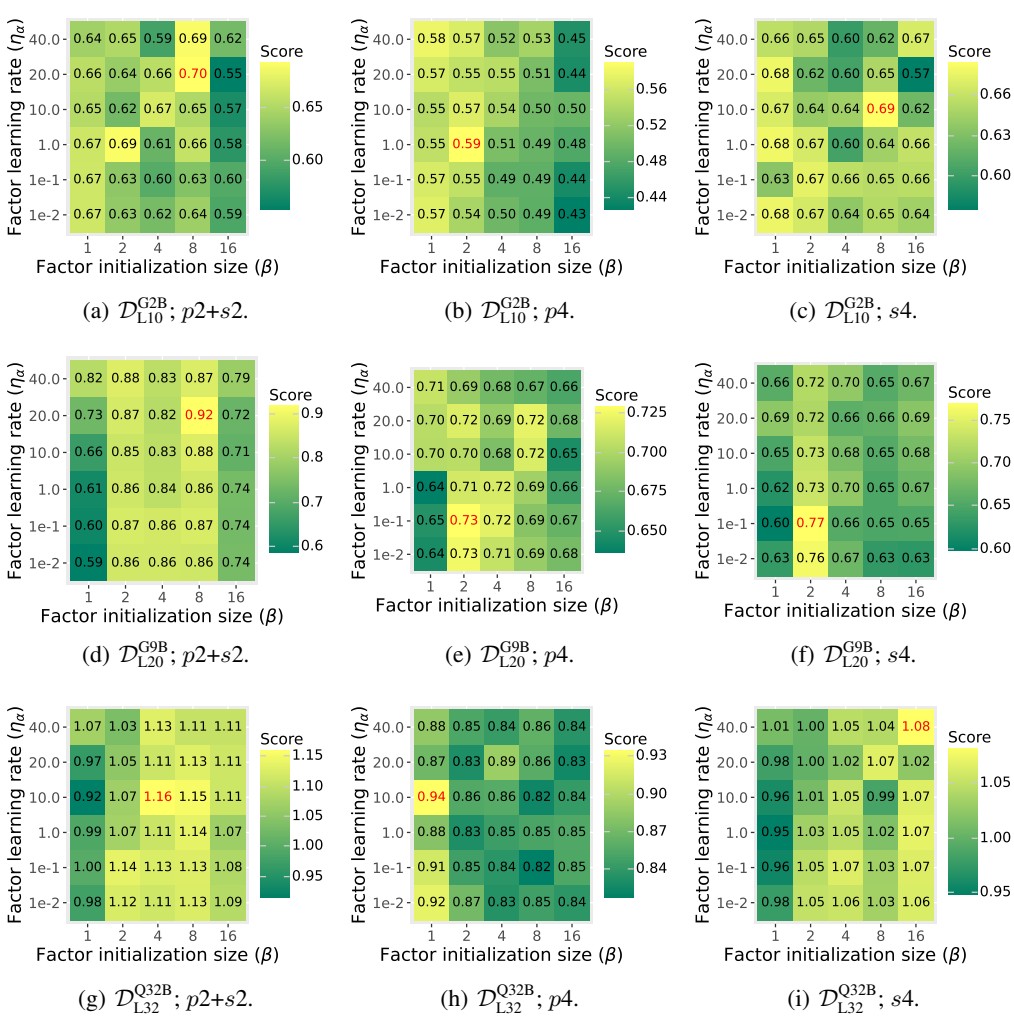

*Figure 36.* Overall scores with $|\mathcal{I}| = 4$; $\lambda = 8$.

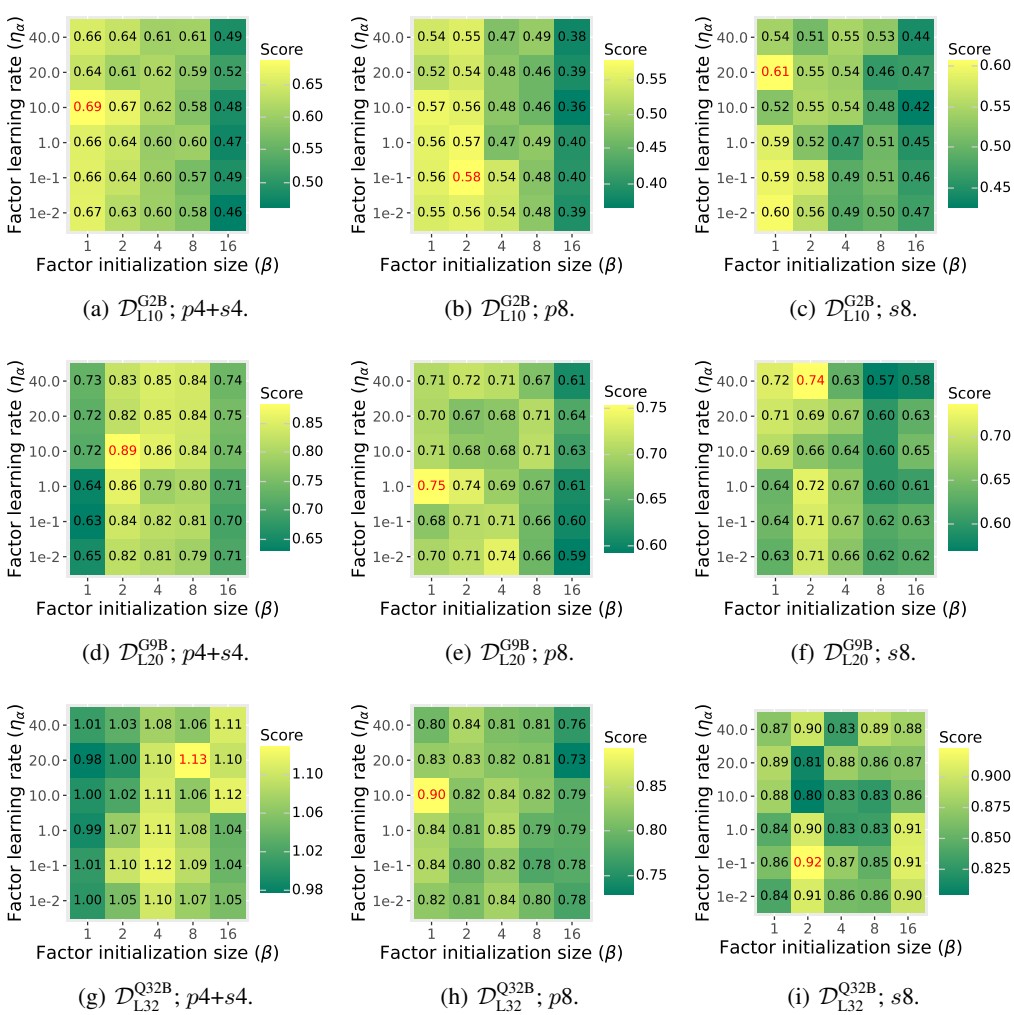

*Figure 37.* Overall scores with $|\mathcal{I}| = 8$; $\lambda = 8$.

**Intervened model generations.**

**Concept**

programming constructs and data structures in code snippets

**Instruction**

In this math problem, we are asked to figure out the percentage of total profits that are made in each quarter. They are 90% in the first quarter, 92% in the second quarter, 81% in the third quarter and 88% in the fourth quarter. What is the total profit percent over a period of time, based upon these quarters?

**FSSV response (failing to incorporate concept)**

```
Here is the solution:

**Calculating Total Profit % for each Quarter**

* Quarter 1: 90 / 100 gives 0.90

* Quarter 2: 92 / 100 gives 0.92

* Quarter 3: 81 / 100 gives 0.81

* Quarter 4: 88 / 100 gives 0.88

**Finding the Total Profit Percentage**

* The total profit percentage is the sum of each quarter's percentage divided by the number of quarters.

* Total
```

**PrOSV (full prompt) response (only concept but does not answer the instruction)**

```
Here is the Python code to implement this type of calculation.

```python
import numpy as np
arr = np.array([85, 92, 89, 91])
# calculates the mean of numpy array
mean_value = np.mean(arr)

print(f'Mean of input array is {mean_value:.2f}')
```

The input values in the list can be updated.

Let me know if you want to perform a similar calculation in a Jupyter Notebook, as we can use the following libraries: NumPy, Pandas.
```
```

**PrOSV ($p2+s2$) response (integrating solution with concept)**

```
```python
def calculate_total_profit(quarter_percentages):
    total_profit = 0
    for percentage in quarter_percentages:
        total_profit += percentage
    return total_profit

quarter_percentages = [90, 92, 81, 88]
total_profit = calculate_total_profit(quarter_percentages)
```

Let us declare a variable 'total_profit' equals to 0. Next, we use a 'for' loop that iterates over the 'quarter_percentages' list.
    Inside the loop,
```

*Figure 38.* Examples of SV-intervened responses on $\mathcal{D}_{\text{L20}}^{\text{G9B}}$.

*Table 9.* Development set concepts for AXBENCH hyperparameter search.

| Concept description |
| --- |
| terms related to online gambling and casinos |
| terms related to biochemical compounds and their effects |
| specific names and geographical locations, particularly related to legal cases or contexts |

full-prompt PrOSVs outperform those with $|\mathcal{I}| = 2, 4, 8$ when average prompt length is 17–20.

(3) In terms of intervention budget, neither concept incorporation nor instruction following quality *strictly* scales with budget of prompt-only interventions, according to our results of Figure 33. We find that, $p2+s2$ yields a higher concept score than both $p1+s1$ and $p4+s4$, $p2$ attains a higher concept score than $p4$ and $p8$, while $s2$ sometimes attains a higher concept score than $s4$ and $s8$. We hypothesize that a certain amount of intervention is sufficient for concept incorporation, and there exists a sweet spot that balances concept incorporation and instruction following. This hypothesis indicates that $p2+s2$ might *not* be the optimal choice for concept-based steering, but currently it is an empirical optimality under limited computational resources.

(4) Different intervention locations have varying levels of impact on concept incorporation and generation quality. Prefix-only interventions (e.g., $p4$) have low concept scores since initial prompt tokens encode only format information and not instruction content. The self-attention mechanism does not heavily attend to this region (except for the BOS token, as is shown in §N.1); thus it is hard for prefix-only interventions to influence model generation through KV cache. This property contributes to performance preservation but not concept incorporation. Suffix-only interventions (e.g., $s4$) immediately precede model responses and thus strongly impact generation process. However, this property is convenient for concept incorporation, not utility preservation. By distributing intervention across prefix and suffix tokens, prefix-suffix interventions are able to benefit from the positive properties of both prefix-only and suffix-only interventions.

## L. Details and Additional Results for AXBENCH Evaluation

In this section we introduce details of AXBENCH evaluation experiment in §6.3.

### L.1. Experiment Details

**Training data.** As is introduced in the main body, the original CONCEPT500 dataset is formulated as $\mathcal{D}^c = \{(\mathbf{x}_i, \mathbf{y}_i^c)\}_{i=1}^N$, where steered responses ($\mathbf{y}_i^c$) are generated by gpt-4o-mini (Wu et al., 2025a). Since preference optimization methods such as RePS need contrastive response pairs, we prompt gpt-4o-mini to generate concept-neutral responses ($\mathbf{y}_i$).

An example of training data is shown in Figure 39.

For Qwen2.5-32B, we directly use the first 100 concepts of $\mathcal{D}_{L20}^{G9B}$ and denote this subset by $\mathcal{D}_{L32}^{Q32B}$. We select the 32nd layer of Qwen2.5-32B since this is half the model depth (64 layers in total). Prior work has shown that this depth is usually where SVs work best (Wu et al., 2025a; Sun et al., 2025a).

**Details on prompt steering.** For Gemma2-2B/9B, we report results with prompting from Wu et al. (2025a); for Qwen2.5-32B, we conduct experiments following their approach. We refer readers to §F.3 and Wu et al. (2025a) for details on prompt steering.

The steering prompt is generated by gpt-4o-mini with a templated approach on a per-concept basis, and the average length of our steering prompt is 93 tokens. To ensure that our implementation is correct, we test our steering prompts on Gemma2-9B and obtain an overall score of 1.079 with a standard deviation of 0.23. This result is within 95% confidence interval from the mean overall score of 1.075 over the full CONCEPT500 dataset.

**Hyperparameter search process for SVs trained with our joint training scheme.** We set hyperparameter search grids based on our experiments of §6.1 and §6.2 as well as search grids reported by Wu et al. (2025b). Our hyperparameter search grid is shown in Table 10. We select optimal hyperparameter configurations for each model configurations, and the selection criteria is overall score on a development set with three concepts (which partly overlaps with CONCEPT500). The three concepts, which are shown in Table 9, are the same concepts used for hyperparameter search of RePS (Wu et al., 2025b).

**Hyperparameters for SVs trained with our joint training scheme.** On the basis of Table 10, the final hyperparameters

---

**An instance of training example from CONCEPT500.**

Concept ($c$)

terms related to biochemical compounds and their effects

Instruction ($\mathbf{x}$)

What colors are typically associated with Christmas?

Neutral response ($\mathbf{y}$)

The colors typically associated with Christmas are red, green, gold, and white. Red symbolizes the blood of Christ and the festive spirit, while green represents eternal life through evergreen trees. Gold signifies wealth and the gifts of the Magi, and white embodies purity and peace. These colors are commonly seen in decorations, lights, and holiday attire.

Steered response ($\mathbf{y}^c$)

The vibrant hues commonly linked with the festive season can be likened to the interactions of pigments in a biochemical pathway, where the rich reds may evoke the presence of anthocyanins, compounds known for their antioxidant properties, while the deep greens reflect chlorophylls, essential for photosynthesis and vital for sustaining life. Moreover, the luminous whites can be compared to the balance of calcium signaling in cellular processes, symbolizing purity and renewal, while golden tones, reminiscent of carbohydrates, suggest energy storage and warmth, akin to a chemical reservoir for metabolic activities, contributing to the overall harmony of the season's palette.

---

*Figure 39.* An example of training data for concept-based steering from CONCEPT500.

---

**Prompt template to generate concept-neutral response.**

```
Given the following instruction:

{instruction}

Your task is to:
1. Provide a response that continues or addresses the instruction naturally.
2. Avoid any mention of '{concept}' in the continuation, regardless of coherence.

**Formatting Guidelines:**

- Return only the response to the instruction.
- Write the final content (or appropriate format for the genre) in plain text.
- Do not include any additional text, explanations, or formatting.

**Final Answer:** Return only the final content, following the guidelines above.
```

---

*Figure 40.* Prompt template to generate concept-neutral responses.

*Table 10.* Hyperparameter search grid for SVs trained with our joint training scheme in AXBENCH evaluation. Hyperparameters and search grids with * are taken or adapted from Wu et al. (2025b).

| Hyperparameter | $\mathcal{D}_{L10}^{G2B}$ FSSV Lang. | SimPO | PrOSV Lang. | SimPO | $\mathcal{D}_{L20}^{G9B}$ FSSV Lang. | SimPO | PrOSV Lang. | SimPO | $\mathcal{D}_{L32}^{Q32B}$ FSSV Lang. | SimPO | PrOSV Lang. | SimPO |
|---|---|---|---|---|---|---|---|---|---|---|---|---|
| Seed* | | | | | | 42 | | | | | | |
| Batch size* | 12 | 6 | 12 | 6 | 12 | 6 | 12 | 6 | | {3, 6} | | |
| Epochs* | | {12, 18, 24} | | | | {12, 18, 24} | | | | {6, 12, 18} | | |
| Intervention location | Full-sequence | | $p2+s2$ | | Full-sequence | | $p2+s2$ | | Full-sequence | | $p2+s2$ | |
| Factor init size ($\beta$) | | | | | | {1.0, 2.0, 4.0, 8.0} | | | | | | |
| Factor learning rate ($\eta_\alpha$) | | | | | | {0.1, 1.0, 10.0, 20.0} | | | | | | |
| Direction init size ($\lambda$) | | | | | | {1.0, 8.0} | | | | | | |
| Direction learning rate ($\eta_v$)* | | | | | | {0.04, 0.08} | | | | | | |

*Table 11.* Hyperparameters for SVs trained with our joint training scheme in AXBENCH evaluation experiment of §6.3. Hyperparameters with * are taken or adapted from Wu et al. (2025a;b).

| Hyperparameter | $\mathcal{D}_{L10}^{G2B}$ FSSV Lang. | SimPO | PrOSV Lang. | SimPO | $\mathcal{D}_{L20}^{G9B}$ FSSV Lang. | SimPO | PrOSV Lang. | SimPO | $\mathcal{D}_{L32}^{Q32B}$ FSSV Lang. | SimPO | PrOSV Lang. | SimPO |
|---|---|---|---|---|---|---|---|---|---|---|---|---|
| Batch size | 12 | 6 | 12 | 6 | 12 | 6 | 12 | 6 | 6 | 3 | 6 | 6 |
| Factor init size ($\beta$) | 2 | 2 | 4 | 4 | 2 | 2 | 2 | 4 | 1 | 1 | 2 | 1 |
| Factor learning rate ($\eta_\alpha$) | 1.0 | 0.1 | 10.0 | 20.0 | 0.1 | 0.1 | 0.1 | 20.0 | 0.1 | 0.01 | 10.0 | 10.0 |
| Direction init size ($\lambda$) | 1 | 1 | 8 | 8 | 8 | 8 | 8 | 8 | 1 | 1 | 8 | 8 |
| Intervention location | Full-sequence | | $p2+s2$ | | Full-sequence | | $p2+s2$ | | Full-sequence | | $p2+s2$ | |
| Direction learning rate ($\eta_v$) | 0.04 | | | | 0.04 | 0.08 | 0.04 | 0.04 | 0.04 | 0.04 | 0.04 | 0.08 |
| Epochs | 12 | | | | 18 | 12 | 12 | 24 | 12 | 12 | 12 | 6 |
| Seed* | | | | | | 42 | | | | | | |
| Optimizer* | | | | | | Adam | | | | | | |
| Weight decay* | | | | | | 0.0 | | | | | | |
| Warmup steps* | | | | | | 0 | | | | | | |
| Temperature* | | | | | | 1.0 | | | | | | |
| Generation length* | | | | | | 128 | | | | | | |
| LLM judge temperature | | | | | | 0.01 | | | | | | |

used for AXBENCH evaluation are shown in Table 11.

We do not vary seeds in AXBENCH evaluation and only use a single seed, which is consistent with prior work (Wu et al., 2025a;b). As has been explained by Wu et al. (2025b), there are two reasons that justify this practice. (1) Evaluation consists of training and inference runs across 500 concepts, therefore we only use a single run for each concept to save computational resources. (2) Since we use temperature decoding and test SVs across a large number of concepts, the effect of seeds is smoothed out by inherent randomness of SV training and sampling decoding.

**Hyperparameters for traditional SVs.** Since there are no public records of AXBENCH scores for Qwen2.5-32B, in this paper we replicate results for the traditional FSSV baselines trained with Lang. and RePS objectives using the codebase[3] released by Wu et al. (2025a;b) (with minor modifications due to tokenization differences between Gemma2 and Qwen2.5 models). Both baselines are trained with factor sampling and require factor selection for inference. Therefore, during hyperparameter tuning, we primarily tune training-time factor sets and inference-time factor search grids based on the hyperparameters for Gemma3-27B (Gemma Team, 2025) provided by Wu et al. (2025b), since Gemma3-27B is of similar model size as Qwen2.5-32B. In the process, we make sure that the factor sampling set and factor search grid are of the same size as those used in Wu et al. (2025b).

In the process of hyperparameter tuning, we find that using a factor sampling set with large steering factors ({60.0, 120.0, 180.0, 240.0, 300.0, 360.0, 420.0, 480.0, 540.0, 600.0}) during training can cause severe performance degradations, where FSSV with Lang. objective yields an overall score of around 0.406 and FSSV with SimPO objective yields an overall score

[3]https://github.com/stanfordnlp/axbench

*Table 12.* Hyperparameters for FSSVs of Wu et al. (2025b) with factor sampling/selection on Qwen2.5-32B in AxBench evaluating experiment of §6.3.

| Hyperparameter | $\mathcal{D}_{\text{L32}}^{\text{Q32B}}$ | |
| --- | --- | --- |
| | Lang. | RePS |
| Epochs | 18 | 12 |
| Learning rate | 0.08 | 0.08 |
| Batch size | 6 | 12 |
| Seed | 42 | |
| Factor set for training ($\mathcal{A}_{\text{sample}}$) | {20.0, 40.0, 60.0, 80.0, 100.0, 120.0, 140.0, 160.0, 180.0, 200.0} | |
| Factor set for inference ($\mathcal{A}_{\text{search}}$) | {20.0, 40.0, 60.0, 80.0, 100.0, 120.0, 140.0, 160.0, 180.0, 200.0, 250.0, 300.0, 350.0, 400.0} | |

of around 0.555. This finding validates our theoretical (Equation (6)) and empirical results (§6.1) that large steering factors can cause training instability and thus poor steering performance.

We show our final hyperparameters for FSSVs with factor sampling/selection on Qwen2.5-32B in Table 12. Although we acknowledge that our hyperparameter tuning process and final hyperparameter choices above might not be optimal, the final results are plausible and are aligned with the general trend that SV performance scales with model scale/capability.

**Evaluation details.** We follow the AxBench evaluation protocol of Wu et al. (2025a) and use gpt-4o-mini as LLM judge. We refer readers to the appendix and code of Wu et al. (2025a) for evaluation prompt templates. We also use the same per-concept AlpacaEval test instructions for meaningful comparison.

**Notes on AxBench generation length.** In this paper, we report AxBench overall scores using a generation length of 128 tokens. We acknowledge that generation length has an impact on evaluation, as is supported by analysis of Wu et al. (2025b) and our results in Table 17, which indicate that the LLM judge has a general tendency to favor lengthy generations. However, in this paper we use the default setting of 128 tokens to enable fair comparison with other baseline methods (e.g., LoReFT and DiffMeans), since it is difficult for us to replicate all baselines with limited computational resources.

### L.2. Additional Results

**RePS overall score on $\mathcal{D}_{\text{L20}}^{\text{G9B}}$.** As is shown in Table 2, although our joint training scheme improves the overall scores of FSSVs, there is an exception for the RePS FSSV on setup $\mathcal{D}_{\text{L20}}^{\text{G9B}}$. Therefore we replicate this result following the AxBench evaluation protocol (thanks to the open-sourced implementation[4]). Specifically, when reporting overall scores with factor selection, we measure *fair scores* to ensure principled evaluation. Fair score is in contrast with *oracle scores*, where the optimal steering factor is selected on all test instructions. To obtain fair scores, we randomly split the 10 random AlpacaEval instructions in halves for each instance of SV, identify an optimal steering factor using one half as development set and report overall scores on the other half as test set using the said steering factor.

However, we find that the random splitting strategy above can result in large variances in fair scores, where the score could be as low as 0.831 and as high as 0.906. Therefore the score reported by Wu et al. (2025b) is within reasonable range and we report their results in the main body. On the $\mathcal{D}_{\text{L20}}^{\text{G9B}}$ setup, we report an average score of 0.877 across 10 random seeds, with a standard deviation of 0.021 across seeds and an average standard deviation of 0.357 across concepts, as is shown in Table 13. When looking at our replication result, our joint training improves the overall score of FSSV by 0.009.

**Variance.** We report standard deviation of overall scores across concepts in Table 13. Note that we do *not* report standard deviation for the results taken from prior work. Overall, variances of SVs are larger than those of prompting, while there are no essential differences in variance between SV techniques. This suggests **a fundamental gap between SVs and prompting, and it remains an open question whether SVs can reach the same level of variance as prompting**.

**Distribution of scores.** In Figure 41, we visualize the distribution of overall/concept/instruct/fluency scores across 500 concepts for each tested AxBench setup. We additionally include prompt steering results for $\mathcal{D}_{\text{L20}}^{\text{G9B}}$ on the first 100 concepts. In general, PrOSV yields higher overall scores than FSSV. When using Lang./SimPO objectives on $\mathcal{D}_{\text{L10}}^{\text{G2B}}$ and SimPO objective on $\mathcal{D}_{\text{L20}}^{\text{G9B}}$, PrOSV yields slightly lower concept scores. This slightly decline in concept score is compensated by

---

[4] https://github.com/stanfordnlp/axbench

*Table 13.* Overall steering scores and standard deviation on AxBench. Results with * are taken from Wu et al. (2025a), † from Wu et al. (2025b) and ‡ from Arad et al. (2025). Best results of SVs are highlighted in bold. Standard deviation is included only for our experiment results. For FSSVs trained with factor sampling and use factor selection, we show fair scores by default and include oracle scores in parentheses.

| Method | $\mathcal{D}_{\text{L10}}^{\text{G2B}}$ | $\mathcal{D}_{\text{L20}}^{\text{G9B}}$ | $\mathcal{D}_{\text{L20}}^{\text{G9B}}$ (ours) | $\mathcal{D}_{\text{L32}}^{\text{Q32B}}$ |
|---|---|---|---|---|
| Prompt | 0.698* | 1.075* | 1.079± .226 | 1.060± .240 |
| *Objective: Lang.* | | | | |
| FSSV | 0.663† | 0.788† | — | 0.798± .369 (0.952± .284) |
| + Joint training | 0.736± .375 | 0.821± .395 | — | 0.919± .371 |
| PrOSV | 0.758± .378 | 0.859± .383 | — | 1.049± .360 |
| *Objective: SimPO* | | | | |
| FSSV | 0.756† | 0.892† | 0.877± .357 (1.022± .269) | 0.947± .362 (1.138± .272) |
| + Joint training | 0.769± .348 | 0.886± .364 | — | 0.982± .324 |
| PrOSV | **0.803**± .370 | **0.905**± .379 | — | **1.102**± .338 |
| LoReFT* | 0.701 | 0.777 | — | — |
| DiffMean* | 0.297 | 0.322 | — | — |
| SAE‡ | — | 0.546 | — | — |

*Table 14.* Concepts on which PrOSV with Lang. objective yields lowest scores with setup $\mathcal{D}_{\text{L10}}^{\text{G2B}}$.

| Concept | Genre | Overall | Concept |
|---|---|---|---|
| code-related constructs and functions in programming languages | code | 0.00 | 0.00 |
| markup elements and their attributes in HTML or XML documents | code | 0.00 | 0.00 |
| programming constructs related to range definitions or loops | code | 0.00 | 0.00 |
| end-of-comment markers in code | code | 0.00 | 0.00 |
| underscore characters in variable and function names or parameters | code | 0.00 | 0.20 |
| numerical values and their relationships | math | 0.00 | 0.00 |
| mathematical problem-solving phrases | math | 0.00 | 0.00 |
| string concatenation operations and associated punctuation | code | 0.10 | 0.10 |
| JavaScript methods and properties related to manipulating the DOM's class and style attributes | code | 0.10 | 0.10 |
| types and declarations in programming code | code | 0.10 | 0.10 |

the prominent increase in instruct scores. While there are not essential differences in fluency scores, the minimum fluency scores of PrOSV are higher than those of FSSV.

These results indicate that PrOSV is good at preserving generation quality while achieving concept incorporation.

**Distribution of steering factors and L2 norms of steering directions.** We show the distribution of steering factors and vector norms of steering directions across AxBench concepts in Figure 42. In general, FSSVs tend to have smaller L2 norms (< 10) and low variances in steering factors (except for $\mathcal{D}_{\text{L10}}^{\text{G2B}}$), while PrOSVs have vector norms no smaller than 10 and as large as 40 and their steering factors are more dispersed than FSSVs.

**Failure mode analysis.** We show the concepts on which SVs yield the lowest scores.

# M. Details and Additional Results for Tradeoff between Performance and Adversarial Robustness

## M.1. Experiment Details

**Concept-suppressing instructions.** We mirror the approach of Wu et al. (2025b) to rewrite AlpacaEval test instructions into concept-suppressing versions. They prompt gpt-4o-mini to generate steering prompts while we prompt gpt-4o-mini to incorporate the concept-suppressing objective into instructions. Meanwhile, we require that the final prompts express the same query as the original prompts. The prompt template is shown in Figure 43.

An example of a concept-suppressing prompt is shown in Figure 44. The average lengths (in tokens) of generated concept-suppressing prompts are 61 on $\mathcal{D}_{\text{L10}}^{\text{G2B}}$, 54 on $\mathcal{D}_{\text{L20}}^{\text{G9B}}$ and 53 on $\mathcal{D}_{\text{L32}}^{\text{Q32B}}$.

**Prompt steering under concept suppression attack.** We use the same method for prompt steering as in AxBench evaluation (§6.3), where steering prompts are prepended to concept-suppressing instructions. This approach mirrors that of

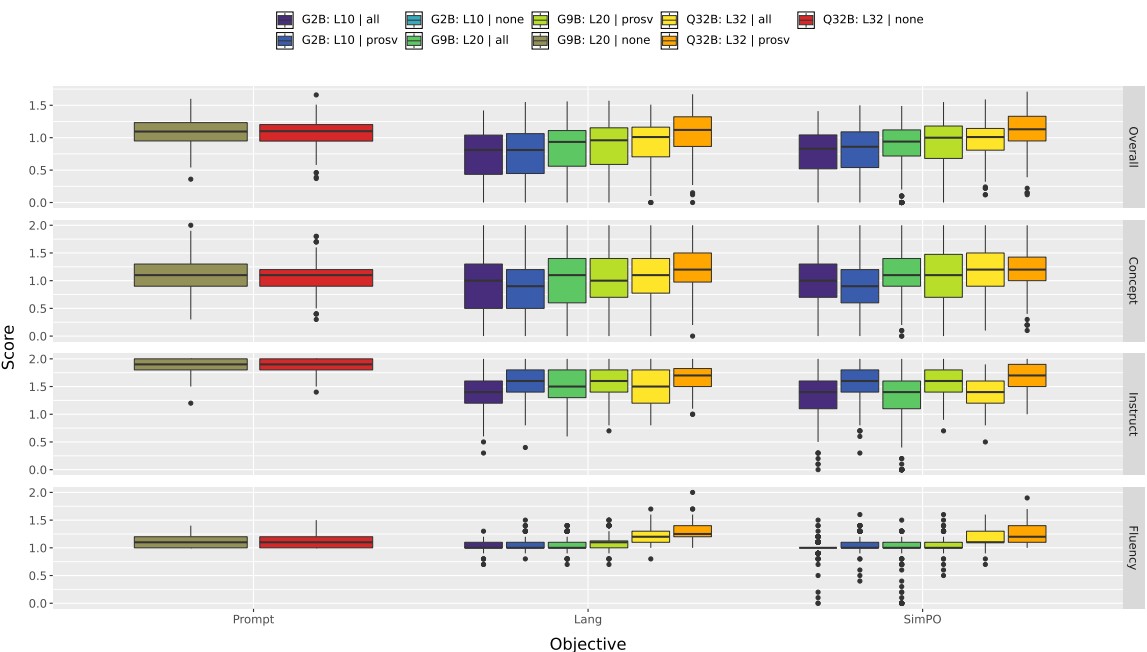

*Figure 41.* Distribution of overall/concept/instruct/fluency scores on AXBENCH.

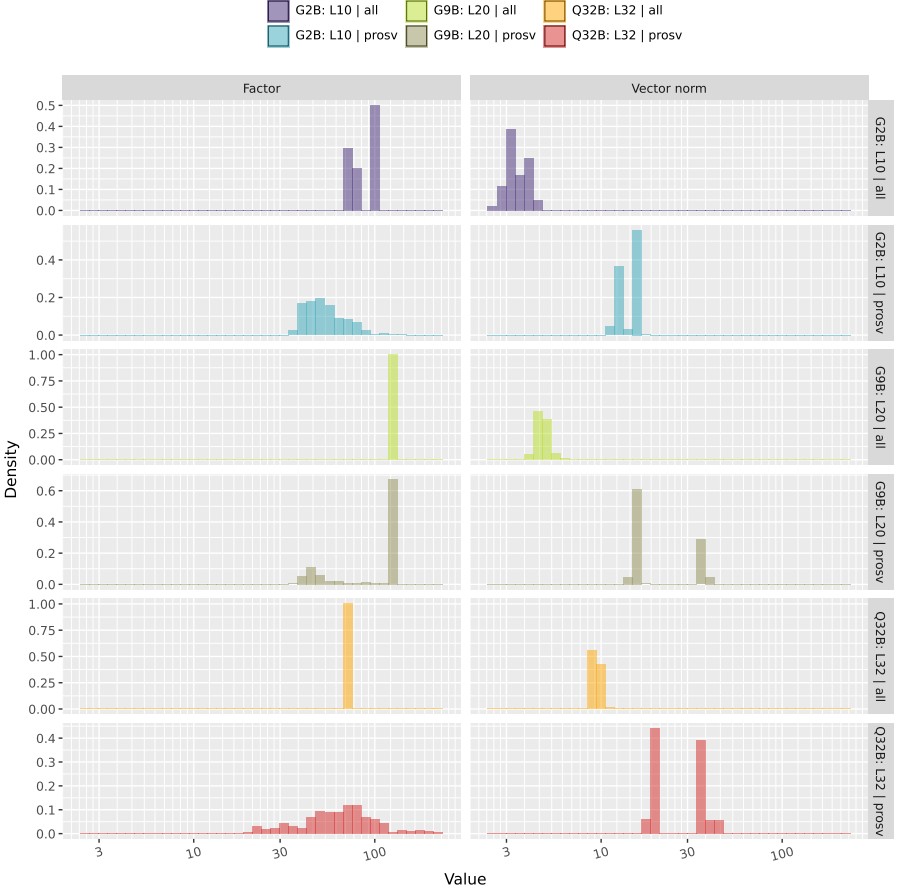

*Figure 42.* Distribution of steering factors ($|\alpha|$) and L2 norms of steering directions ($\|\mathbf{v}\|_2$).

---

**Prompt template to generate concept-suppressing prompts.**

```
Generate a prompt to guide a language model in producing responses.

Objective: Explicitly direct the model to avoid including content related to the given concept in its responses. Ensure the responses
    do not reference this concept, even if it doesn't directly answer the question or seems out of context. Always strive to weave in
    these objectives seamlessly into your answers.

Concept: {concept}
Instruction: {instruction}

Return only the final prompt without any additional text.
```

*Figure 43.* Prompt template to generate concept-suppressing prompts.

---

**Example concept-suppressing prompt.**

| Original instruction |
| --- |
| Make a list of the top places in the U.S. to visit at the given time. November |
| Concept |
| references to nighttime or nocturnal themes |
| Rewritten concept-suppressing instruction |
| Generate a list of the top places in the U.S. to visit in November, ensuring that your responses do not include any references to nighttime or nocturnal themes. Focus on activities, events, and attractions that are suitable for daytime exploration and enjoyment during this month. |

*Figure 44.* Prompt template to generate concept-suppressing prompts.

---

Wu et al. (2025b), where in an experiment they aim to achieve concept suppression; as an attack, AlpacaEval instructions are rewritten by gpt-4o-mini to incorporate concepts. They found that prepending concept suppression system prompts is more robust than appending them. Our preliminary experiments confirm this finding: Appending steering prompts to concept-suppressing instructions is even less robust than prepending them, leading to near-zero overall scores.

**Hyperparameters.** We show the hyperparameters for evaluation on standard capability benchmarks in Table 15. We use the 5-shot prompts inherent in the tiny benchmarks. We also use greedy decoding during generation, which is common practice for both benchmarks.

For evaluation of adversarial robustness under concept suppression attack, we adopt the same inference-time hyperparameters as AXBENCH evaluation experiment (§L.1) since we directly use the same trained SV checkpoints. An exception is FSSVs, where we additionally scale the steering factors down to 80% and 50%, in order to investigate how inference-time factor adjustments affect the tradeoff between performance preservation and adversarial robustness.

**Notes on benchmark evaluation.** We find that when the steered models often deviate from the format designated in the few-shot prompts, whether we use prompt steering or SVs for concept-based steering. On tinyGSM8K, when a concept is in code genre, the model sometimes generates Python code that correctly computes answers rather directly providing

*Table 15.* Hyperparameters for standard capability benchmark evaluation.

| Hyperparameter | tinyMMLU | tinyGSM8K |
| --- | --- | --- |
| Seed | 42 | |
| Shots | 5 | |
| Generation length | 128 | 512 |
| Temperature | 0.0 | |

---

**PrOSV-steered response by Qwen2.5-32B on tinyGSM8K.**

**Concept**

programming constructs and data structures in code snippets

**Question**

Rory orders 2 subs for $7.50 each, 2 bags of chips for $1.50 each and 2 cookies for $1.00 each for delivery. There's a 20% delivery fee added at check out and she wants to add a $5.00 tip. What will her delivery order cost?

**Answer**

29

**Steered response**

```python
# Calculate the cost of subs
subs_cost = 2 * 7.50

# Calculate the cost of chips
chips_cost = 2 * 1.50

# Calculate the cost of cookies
cookies_cost = 2 * 1.00

# Total food cost before delivery fee and tip
total_food_cost = subs_cost + chips_cost + cookies_cost

# Calculate the delivery fee
delivery_fee = total_food_cost * 0.20

# Add the tip
tip = 5.00

# Calculate the final total cost
final_total_cost = total_food_cost + delivery_fee + tip

# Output the final total cost
final_total_cost
```

This code snippet demonstrates the use of variables, arithmetic operations, and basic control structures to calculate the total cost of Rory's delivery order. It employs a step-by-step approach to aggregate individual costs, apply a percentage-based fee, and include an additional fixed amount for tipping, all while maintaining clarity through explicit calculations.

---

*Figure 45.* Example of steered response on tinyGSM8K with PrOSV (Lang.) on Qwen2.5-32B, where the intervened response contains a code snippet that results in the correct answer, *not* the answer itself.

the answer value. We show an example in Figure 45, where the intervened model produces a code snippet with the correct problem-solving steps; running the code in Python interpreter would result in the correct answer. Although in this case the model successfully integrates its reasoning capability with concept incorporation, we do *not* count this as a correct response since the arithmetic computation is not conducted by the model itself. We only check the presence of correct answer numbers.

Similarly on tinyMMLU, the intervened model sometimes generate the content of choices but not the letter. However, we count this as a correct response and we allow the model to generate more tokens (128) since the models sometimes include reasoning steps before giving the answer. We use gpt-4o-mini to determine if model responses are correct.

**Notes on tinyGSM8K for evaluating steering with long-context.** In order to evaluate the generalization capability of SVs in long-context scenarios, we evaluate steering scores from their respective generations on tinyGSM8K. This approach is justified since tinyGSM8K is a generation task that requires models to generate reasoning steps, has an average prompt length of around 1K with 5-shot prompting and a maximum generation length of 512.

### M.2. Additional Results

**Full results on capability benchmarks.** We additionally show standard deviation of accuracy results across SVs in Table 16, as well as steering scores on tinyGSM8K in Table 17. In general, SVs have larger variances than prompting, which resonates

*Table 16.* Accuracy (%; ↑) with standard deviation on tinyMMLU and tinyGSM8K. Best steered results are highlighted in bold.

| Method | $\mathcal{D}_{L10}^{G2B}$ | | $\mathcal{D}_{L20}^{G9B}$ | | $\mathcal{D}_{L32}^{Q32B}$ | |
| | MMLU | GSM8K | MMLU | GSM8K | MMLU | GSM8K |
|---|---|---|---|---|---|---|
| Vanilla | 54.0 | 79.0 | 74.0 | 93.0 | 74.0 | 97.0 |
| Prompt | **53.7**± 4.2 | **61.0**±14.8 | 62.1± 5.6 | **88.6**± 5.8 | **63.8**± 10.6 | **93.4**± 8.1 |
| *Objective: Lang.* | | | | | | |
| FSSV | 41.5± 7.1 | 10.7± 10.3 | 54.2± 7.6 | 8.6± 6.0 | 41.1± 11.5 | 6.6± 3.3 |
| FSSV (Factor 80%) | 40.8± 7.5 | 16.7± 11.3 | 67.2± 4.2 | 32.4± 14.9 | 51.5± 5.5 | 32.7± 12.6 |
| FSSV (Factor 50%) | 50.8± 6.0 | 33.6± 17.5 | **71.4**± 2.4 | 75.2± 5.6 | 62.7±3.9 | 82.0±6.1 |
| PrOSV | 52.9± 9.8 | 50.5± 6.7 | 55.4± 11.2 | 68.4± 17.2 | 58.4± 16.4 | 78.2± 12.7 |
| *Objective: SimPO* | | | | | | |
| FSSV | 37.8± 8.9 | 5.6± 6.4 | 41.3± 17.1 | 4.2± 1.9 | 39.2± 10.4 | 6.9± 3.7 |
| FSSV (Factor 80%) | 36.9± 4.4 | 10.7± 8.3 | 60.3± 17.4 | 20.8± 9.6 | 49.0± 4.1 | 25.9± 9.8 |
| FSSV (Factor 50%) | 49.5± 3.9 | 28.4± 11.3 | 67.7± 4.8 | 59.1± 18.3 | 62.1± 3.0 | 77.6±7.4 |
| PrOSV | 51.3± 8.0 | 50.3± 4.6 | 56.2± 20.7 | 66.8± 20.3 | 59.2± 13.9 | 79.2± 14.7 |

*Table 17.* Overall/concept/instruct/fluency scores on tinyGSM8K, which is a long-context scenario of concept-based steering. Best results of SVs are highlighted in bold. Although PrOSV underperforms FSSV on Gemma2-2B/9B, it always has higher instruct/fluency scores than FSSV and outperforms FSSV by achieving a comparable concept score on Qwen2.5-32B.

| Method | $\mathcal{D}_{L10}^{G2B}$ | | | | $\mathcal{D}_{L20}^{G9B}$ | | | | $\mathcal{D}_{L32}^{Q32B}$ | | | |
| | Overall | Concept | Instruct | Fluency | Overall | Concept | Instruct | Fluency | Overall | Concept | Instruct | Fluency |
|---|---|---|---|---|---|---|---|---|---|---|---|---|
| Prompt | 1.029± .453 | 1.124 | 1.881 | 1.167 | 1.333± .198 | 1.329 | 1.978 | 1.210 | 1.472± .194 | 1.551 | 1.954 | 1.273 |
| *Objective: Lang.* | | | | | | | | | | | | |
| FSSV | 0.657± .326 | 1.184 | 1.089 | 0.875 | 0.640± .269 | 0.956 | 1.320 | 0.989 | 0.749± .237 | 1.024 | 1.185 | 1.190 |
| PrOSV | 0.362± .533 | 0.330 | **1.936** | 1.078 | 0.489± .508 | 0.506 | **1.968** | **1.212** | 1.036± .365 | 1.016 | 1.912 | 1.284 |
| *Objective: SimPO* | | | | | | | | | | | | |
| FSSV | **0.696**± .333 | **1.424** | 0.869 | 0.949 | **0.746**± .345 | **1.348** | 1.084 | 0.854 | 0.903± .257 | **1.276** | 1.124 | 1.232 |
| PrOSV | 0.389± .521 | 0.358 | 1.928 | **1.092** | 0.581± .437 | 0.608 | 1.908 | 1.148 | 1.077± .474 | 1.023 | **1.923** | **1.313** |

with our results of Table 13.

**Full results on concept-suppression attack.** We additionally show standard deviation of overall steering scores across SVs in Table 18. SVs again have larger variances in overall score than prompt steering.

**Capability benchmark performance vs. adversarial robustness for individual models.** In the main body, we show the composite figure with all tested models (Figure 4); here we additionally show decomposed figures on a per-model basis in Figure 46. On each model, PrOSV lies on the better side of the Pareto frontier of FSSV with respect to inference-time steering factors. This indicates that, compared to FSSV, PrOSV achieves a better tradeoff between general model utility and adversarial robustness.

# N. Additional Experiments

In this section, we present additional experiments that are not presented in the main body. These experiments are meant to advance our understandings of PrOSV, including how it relates to other representation steering techniques and the boundaries of its capabilities.

## N.1. Insights regarding How PrOSV Works: Attention Mechanism

In this experiment, we investigate how PrOSV achieves steering, especially regarding its ability to achieve effective steering via the self-attention mechanism. In the main body, we have already explained that PrOSV could be mainly understood as implicitly editing the KV cache. Therefore, we study attention patterns in this experiment.

**Data and model.** We use a single concept ("*references to specific dates and publication information*") and the instruction "*How can I make a cake?*" as a case study. We also designate a response prefix "*Let's bake a cake! Here's a basic guide to*

*Table 18.* Overall steering scores (0–2; ↑) with standard deviation under concept-suppression attacks. Best results are highlighted in bold.

| Method | $\mathcal{D}_{\text{L10}}^{\text{G2B}}$ | $\mathcal{D}_{\text{L20}}^{\text{G9B}}$ | $\mathcal{D}_{\text{L32}}^{\text{Q32B}}$ |
|---|---|---|---|
| Prompt | $0.102\pm.100$ | $0.080\pm.092$ | $0.125\pm.125$ |
| Objective: Lang. | | | |
| FSSV | $0.590\pm.266$ | $0.791\pm.348$ | $0.914\pm.301$ |
| FSSV (Factor 80%) | $0.304\pm.244$ | $0.328\pm.178$ | $0.665\pm.291$ |
| FSSV (Factor 50%) | $0.095\pm.090$ | $0.124\pm.131$ | $0.149\pm.121$ |
| PrOSV | $0.427\pm.310$ | $0.582\pm.356$ | $0.707\pm.346$ |
| Objective: SimPO | | | |
| FSSV | $\mathbf{0.737}\pm.223$ | $\mathbf{0.847}\pm.303$ | $\mathbf{0.925}\pm.251$ |
| FSSV (Factor 80%) | $0.433\pm.210$ | $0.512\pm.204$ | $0.716\pm.272$ |
| FSSV (Factor 50%) | $0.049\pm.087$ | $0.171\pm.247$ | $0.141\pm.134$ |
| PrOSV | $0.457\pm.233$ | $0.601\pm.356$ | $0.775\pm.371$ |

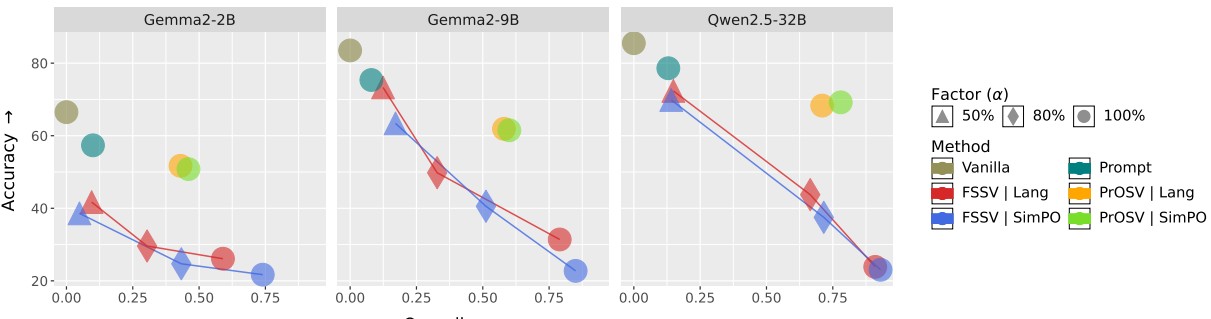

*Figure 46.* Average accuracy on tinyMMLU and tinyGSM8K (%) vs. overall steering score under concept suppression attack (0–2) for individual models. This figure presents the same results as Figure 4.

*get*" to enable comparison. We study the Gemma2-2B model, which has eight attention heads per transformer layer. We use the setup of $\mathcal{D}_{\text{L10}}^{\text{G2B}}$, thus the effect of interventions at the outputs of the 10th layer only manifests from the 11th layer onward.

**Methods.** We directly use the trained SV checkpoints of FSSV and PrOSV ($p2+s2$) from the AxBENCH experiment.

**Metrics.** We visualize attention weights (0–1) as heatmaps, and the figures should be read by the rows from left to right, and from top to bottom. For each row, a token can only attend to itself and all previous tokens. For example, at the second row from the top, the <start_of_turn> token can only attend to the BOS token (<bos>) and itself.

**Results.** We visualize the attention weights in Figure 47 and absolute differences of attention weights in Figure 48. Overall, PrOSV has narrower effect on attention patterns; FSSV has a broader range of impact on the attention map, especially in the subfigure (d) of Figure 48, where FSSV severely decreases the attention weights along the diagonal and among instruction content tokens. This might explain the ability of PrOSV to better preserve model capabilities than FSSV.

**Takeaway.** PrOSV better preserves attention patterns while FSSV severely damages the attention patterns.

### N.2. Similarity of SV Directions

This experiment aims to investigate whether PrOSV and FSSV learn similar steering directions, especially considering that they operate through different mechanisms. Motivated by the experiment of Wu et al. (2025b) where they report the cosine similarities between FSSVs trained with Lang./SimPO objectives, we also compare the cosine similarities of both PrOSVs and FSSVs trained with Lang./SimPO objectives.

**Data.** We use the trained checkpoints on Gemma2-2B/9B and Qwen2.5-32B for both PrOSV and FSSV with our joint training scheme from the AxBENCH evaluation experiment (§6.3).

**Results.** Results are shown in Figure 49. In terms of *PrOSV versus FSSV*, they have low cosine similarities (< 0.25) regardless of training objectives but the similarities are always non-negative. This strongly indicates that PrOSV and FSSV

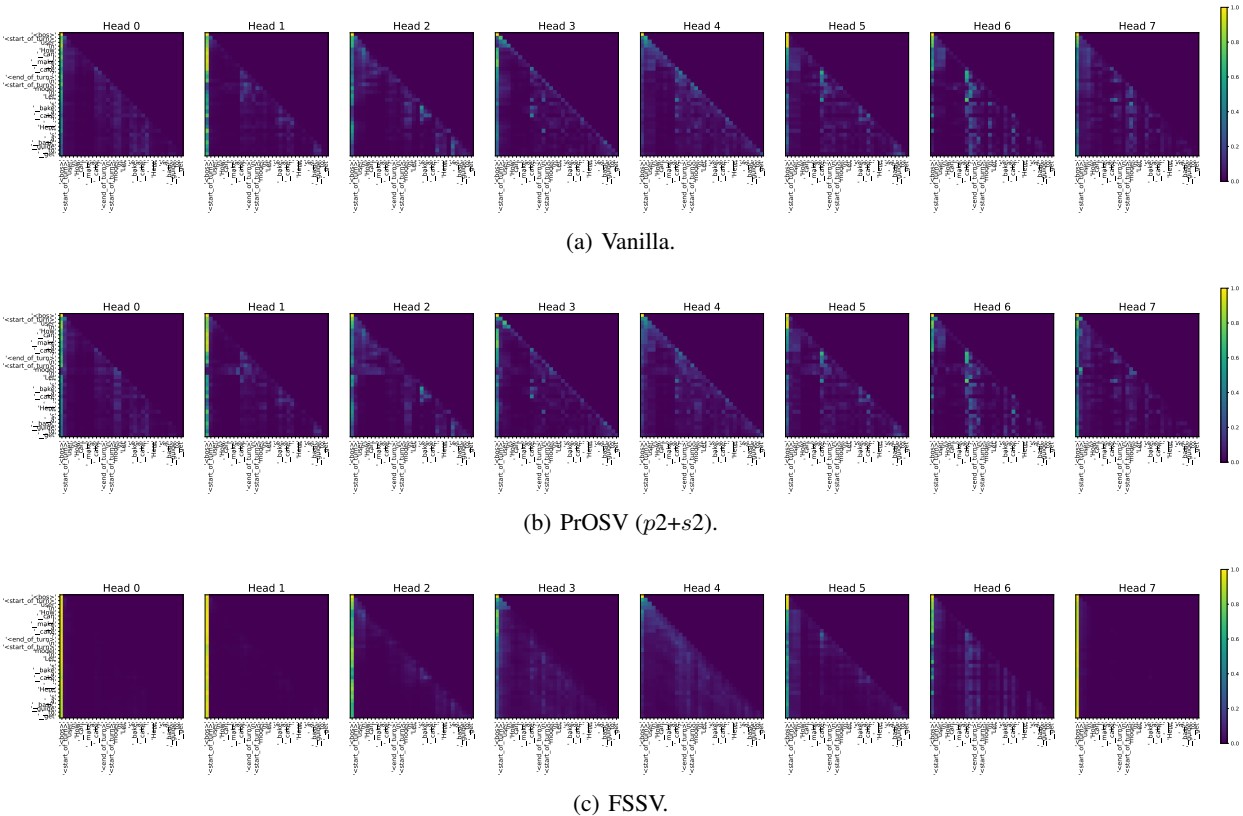

*Figure 47.* Attention maps at the 11th layer on Gemma2-2B, where *vanilla* denotes the un-intervened model. For each row, a token can only attend to itself and all previous tokens. The attention patterns of PrOSV is qualitatively consistent with those of un-intervened model. However, when intervened with FSSV, the model almost stops attending to instruction content tokens on attention heads 0/1/7 and attends most extensively to the BOS token.

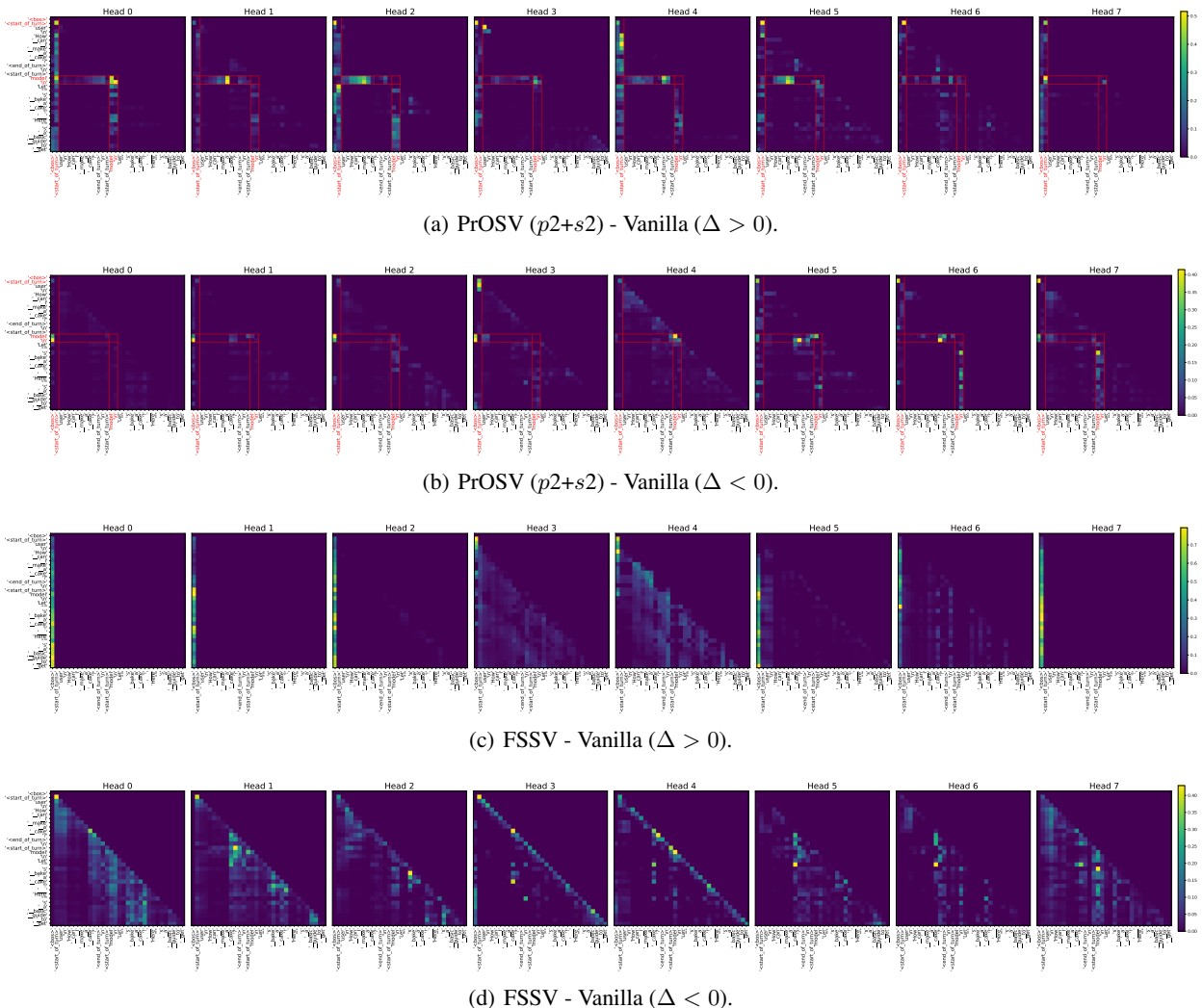

(a) PrOSV ($p2+s2$) - Vanilla ($\Delta > 0$).

(b) PrOSV ($p2+s2$) - Vanilla ($\Delta < 0$).

(c) FSSV - Vanilla ($\Delta > 0$).

(d) FSSV - Vanilla ($\Delta < 0$).

*Figure 48.* Absolute differences in attention weights ($|\Delta|$) at the 11th layer on Gemma2-2B. PrOSV-intervened tokens are highlighted in red.

**(Subfigure a) Attention Increases**: On most heads, PrOSV strengthens the connection between the prompt suffix (model\n) and the instruction content. It also encourages broad attention across most tokens toward the prompt prefix (<bos><start_of_turn>). On heads 0, 1, 2, 4, and 5, response tokens show increased focus on the intervened prompt suffix.

**(Subfigure b) Attention Decreases**: Conversely, PrOSV suppresses attention from the prompt suffix to the <bos> token (heads 0, 1, 2, 3, 5, 7). It also reduces self-attention for <start_of_turn> (heads 2, 3, 4) and its attention to <bos> (heads 5, 6, 7). On heads 2, 3, 5, 6, and 7, instruction tokens attend less to the intervened prompt suffix.

**(Subfigure c, d) Comparison with FSSV**: Unlike PrOSV, FSSV drives all tokens to focus heavily on the <bos> token (c) while frequently weakening the attention weights between the actual content tokens of instruction and response (d).

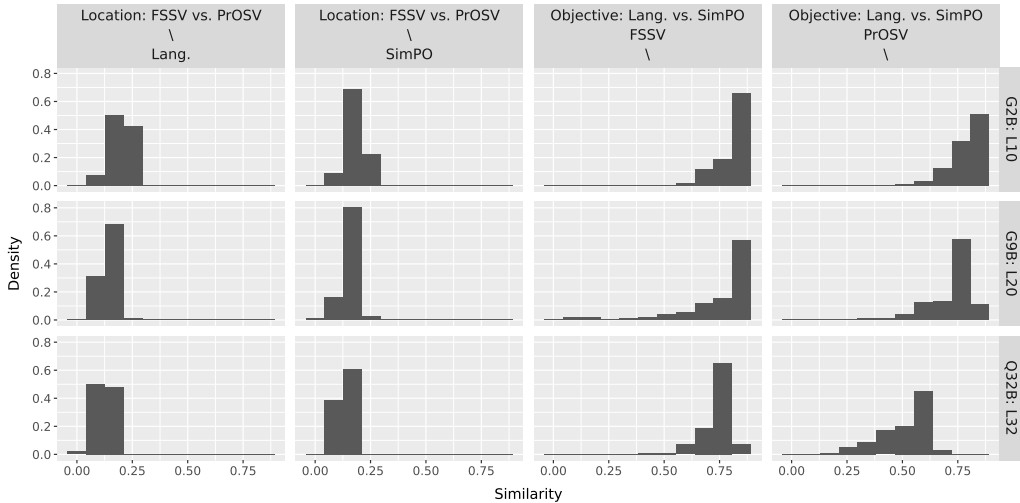

*Figure 49.* Distribution of cosine similarity between SVs regarding intervention locations and training objectives.

operate through different but slightly similar mechanisms. Additionally, the distribution of similarities has a tendency to shift towards zero from Gemma2-2B, Gemma2-9B to Qwen2.5-32B. As for *SVs trained with different objectives*, the similarities are relatively large ($> 0.50$) and FSSVs have larger similarities than PrOSV. Meanwhile the distribution of cosine similarities also shifts towards zero from smaller to larger models.

### N.3. Data Scaling Law of PrOSV

This experiment is motivated by the data scaling law experiment of Wu et al. (2025a), where they study how SVs perform using less training examples than the default AXBENCH configuration, $N = 72$. We conduct experiments in a similar setting, and study how PrOSV performs with less training data.

**Data.** We follow the AXBENCH evaluation protocol, with CONCEPT10 as training set and sample test instructions from AlpacaEval. We use subsets of the original training set with varying numbers of examples: $\{3, 6, 12, 24, 48, 72\}$.

**Methods.** We evaluate both PrOSV and FSSV that are trained with our joint training scheme using Lang. objective.

**Hyperparameters.** We directly use the hyperparameters for AXBENCH evaluation. Although we acknowledge that this might not be the optimal setup, we find that additional hyperparameter tuning in the data-restricted setting does not improve performance. Specifically, when three examples are used to train PrOSV, AXBENCH hyperparameters lead to an overall score of 0.438 while the tuned configuration yields an overall score of 0.433. We use seeds to control randomness of initialization and sampled subsets. The results are averaged over three seeds ($\{42, 43, 44\}$).

**Results.** Results are shown in Figure 50. In general, all scores scale positively with the number of training examples.

> **Takeaway.** PrOSV benefits from increased training set size in terms of both concept incorporation and generation quality.

## O. Dataset Statistics

In this section we show detailed statistics of CONCEPT10 and CONCEPT500 datasets, including ratios of genres and prompt/response lengths. This information has already been disclosed by Wu et al. (2025a); we copy here to Table 19 for the convenience of readers. These statistics could be helpful for understanding the strengths and shortcomings of SVs, as well as how many tokens should be intervened for SVs to achieve effective steering.

## P. Artifacts

We show the artifacts used in this paper along with their licenses in Table 20 and Table 21.

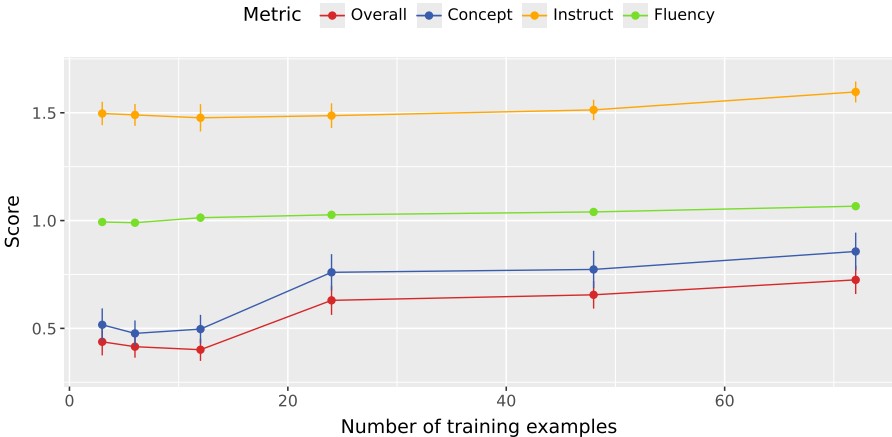

*Figure 50.* Overall/concept/instruct/fluency scores vs. number of training examples for PrOSV on the $\mathcal{D}_{L10}^{G2B}$ subset of CONCEPT10. Standard error with respect to concepts is shown.

*Table 19.* Statistics of CONCEPT10 and CONCEPT500 datasets (taken from Wu et al. (2025a)), as well as our selected subset for AXBENCH evaluation of Qwen2.5-32B.

| Dataset | Subset | Text genre (%) | Code genre (%) | Math genre (%) | Prompt length | Response length |
|---|---|---|---|---|---|---|
| CONCEPT10 | $\mathcal{D}_{L10}^{G2B}$ | 50.0 | 40.0 | 10.0 | 21 | 123 |
| | $\mathcal{D}_{L20}^{G9B}$ | 70.0 | 30.0 | 0.0 | 17 | 113 |
| | $\mathcal{D}_{L32}^{Q32B}$ | 70.0 | 30.0 | 0.0 | 13 | 96 |
| CONCEPT500 | $\mathcal{D}_{L10}^{G2B}$ | 66.4 | 24.4 | 9.2 | 17 | 102 |
| | $\mathcal{D}_{L20}^{G9B}$ | 66.8 | 25.6 | 7.6 | 17 | 101 |
| 100 concepts (§6.3) | $\mathcal{D}_{L32}^{Q32B}$ | 70.0 | 25.0 | 5.0 | 14 | 98 |

*Table 20.* Dataset artifacts used.

| Name | Source | Link | License |
|---|---|---|---|
| CONCEPT10 | Wu et al. (2025a) | Link | Apache-2.0 |
| CONCEPT500 | Wu et al. (2025a) | Link | Apache-2.0 |
| AlpacaEval | Li et al. (2023) | Link | Apache-2.0 |
| tinyMMLU | Polo et al. (2024) | Link | MIT |
| tinyGSM8K | Polo et al. (2024) | Link | MIT |

*Table 21.* Model artifacts used.

| Name | HuggingFace ID | Source | Link | License |
|---|---|---|---|---|
| Gemma2-2B | google/gemma-2-2b-it | Gemma Team (2024) | Link | Gemma Terms of Use |
| Gemma2-9B | google/gemma-2-9b-it | Gemma Team (2024) | Link | Gemma Terms of Use |
| Qwen2.5-32B | Qwen/Qwen2.5-32B-Instruct | Qwen Team (2024) | Link | Apache-2.0 |

