# OpenReview forum: "Towards Steering without Sacrifice: Principled Training of Steering Vectors for Prompt-only Interventions"
_ICML.cc/2026/Conference — ICML 2026 regular_

### Official Review · Reviewer_Bk6Z · 2026-03-09

**Soundness:** 3
**Presentation:** 3
**Significance:** 3
**Originality:** 3
**Overall Recommendation:** 4
**Confidence:** 4

**Summary:**

This paper introduces Prompt-Only Steering Vector (PrOSV), a novel method for efficient concept-based steering in LLMs. PrOSV challenges traditional paradigms through two key innovations: (1) theoretically analyzing training dynamics via neural network scaling theory to propose joint training of steering factors and directions, eliminating post-hoc factor selection; (2) applying interventions only at prefill stage on minimal prompt tokens. Experiments show PrOSV outperforms traditional FSSVs on AXBENCH while better balancing capability preservation and adversarial robustness.

**Compliance With Llm Reviewing Policy:**

Affirmed.

**Final Justification:**

I keep my positive evaluation.

**Key Questions For Authors:**

1. In the §6.1 verification experiments, authors fix $η_v$ = 0.04 and search over $λ ∈ {1,8}$. What is the rationale behind these specific value choices?

2. The paper derives the core condition $$\gamma[\eta_v] + \gamma[\eta_{\alpha}] = 0$$ for joint training, but does performance degrade as expected when this condition is violated? Experiments may be needed to demonstrate this.

3. Why does PrOSV underperform FSSV in long-context robustness on Gemma2-2B/9B, yet outperform it on Qwen2.5-32B? "Benefiting from increased model scale" describes the phenomenon rather than explaining the mechanism. What specific mechanism leads to this scale dependence?

4. The paper states that "PrOSV functions mainly by implicitly editing the KV cache." It would be helpful to provide a more detailed explanation of this implicit KV cache editing.

**Limitations:**

yes

**Strengths And Weaknesses:**

### Strengths:

1. The paper is generally clear and well-written.
2. The theoretical analysis is thorough.
3. Systematic comparisons against multiple baselines are conducted on the AxBENCH benchmark, demonstrating the state-of-the-art performance of PrOSV.

### Weaknesses:

1. PrOSV improves the utility–robustness tradeoff, but it generally underperforms full-sequence SVs in raw concept incorporation strength. This suggests that the proposed method mitigates harm by weakening intervention rather than fundamentally improving controllability. The paper does not clearly disentangle whether gains arise from better parameterization or simply from reduced intervention scope.

2. All major evaluations are conducted on concept-based steering (AXBENCH). It remains unclear whether joint training and prompt-only intervention generalize to other steering paradigms such as safety alignment, persona control, reasoning calibration, or multi-objective steering. This narrows the demonstrated applicability of the approach.

3. Figure 1 is difficult to interpret, as it combines multiple concepts (FSSV vs. PrOSV and factor selection vs. joint training) into a single dense illustration without a clear visual hierarchy, which makes the core contribution hard to grasp at first glance.

---

> ### Author Rebuttal · Authors · 2026-03-28
>
> Thank you for your acknowledgement of our work and valuable feedback.
> We would like to address the raised concerns as follows.
>
> > **W1**. Gains may come from weaker intervention, not better controllability.
>
> We agree that PrOSV typically yields lower concept scores than FSSV *when we are tuning for optimal overall scores*, as is shown in Table 1.
> However, this is often not the case across diverse hyperparameter setups (i.e. initialization size and learning rate).
> Please first kindly refer to Figure 32, where we show how SVs perform across hyperparameter search grids.
> We found that **PrOSV yields higher concept incorporation than FSSV on average** and resides on a better Pareto Frontier between concept incorporation and instruction fulfillment.
> These results indicate that PrOSV benefits from better parameterization rather than merely weakened intervention.
>
> > **W2**. Limited to concept steering; unclear applicability to other steering paradigms.
>
> Our paper already covers 1100 concepts, providing broad empirical scope.
> Beyond this, we expect joint training and PrOSV to excel at high-level concepts.
> Prior work shows that DiffMean already excels at such concepts (e.g., safety, persona) [R1,R2] but underperforms on content concepts of AxBench, where our method excels.
>
> **Additional evidence.**
> We evaluate the safety refusal concept on Gemma2-2B, where the model is steered to refuse even harmless prompts [R1], following AxBench protocol.
> Using joint training scheme and Lang. objective:
> - PrOSV (p2+s2): 1.37
> - FSSV: 1.29
> - DiffMean: 1.01
>
> Results suggest that joint training and PrOSV have potential to generalize to other steering paradigms.
>
> > **W3**. Figure 1 is difficult to interpret.
>
> We will revise Figure 1 with clearer separation of (a) FSSV vs. PrOSV and (b) factor selection vs. joint training, and improve labeling.
>
> > **Q1**. In the §6.1 verification experiments, authors fix $\eta_v=0.04$ and search over $\lambda\in\\{1,8\\}$. What is the rationale behind these specific value choices?
>
> We chose $\eta_v=0.04$ following the hyperparameter choice of [R3,R4].
> The values $\lambda\in\\{1,8\\}$ were selected to cover a meaningful range of relative scaling between direction and factor learning, corresponding to under- and over-regularized regimes predicted by our scaling analysis.
> We will clarify that these values are representative in future revisions.
>
> > **Q2**. Does performance degrade as expected when $\gamma[\eta_v]+\gamma[\eta_\alpha]=0$ is violated?
>
> Yes, we observe that violating the condition leads to degraded performance.
> In particular, please refer to Figure 2, where FSSV suffers from high $\eta_\alpha$ (e.g., $\eta_\alpha \geq 10.0$ in (a) and (f)), while PrOSV yields suboptimal performance with low $\eta_\alpha$ (e.g., $\eta_\alpha\leq20.0$ in (b)).
> This supports our theoretical prediction that balanced scaling is critical for stable joint training.
> We will make this more explicit in the next revision.
>
> > **Q3**. Long-context performance varies by model size; mechanism unclear.
>
> We believe the gap on Gemma2-2B/9B is primarily due to train-test context mismatch rather than an inherent limitation of PrOSV.
> SVs are trained on shorter prompts (17--20 tokens) but tested on tinyGSM8K (1K tokens), which is challenging for PrOSV.
>
> To test this, we train and evaluate on long prompts (2WikiMultihopQA split of LongBench [R5], ~5K tokens) on Gemma2-2B.
> In this setting, **PrOSV outperforms FSSV (0.87 vs. 0.79)**, reversing the result from the short-training/long-testing setting.
> This shows PrOSV remains effective in long contexts when trained accordingly.
>
> Our tentative mechanistic explanation for scale dependence is that PrOSV relies on prompt-conditioned representations whose influence must persist over long horizons.
> Smaller models are more prone to context dilution, making them sensitive to mismatch.
> This scale-dependent stability has also been noted in prior work [R6].
>
> We will revise Section 6.4 to clarify the role of train-test mismatch, include new results, and present the mechanistic hypothesis.
>
> > **Q4**. KV-cache editing explanation is underspecified.
>
> In Appendix M.1 and Figures 46--47, we show how PrOSV and FSSV influence the attention mechanism with attention maps.
> We find that PrOSV preserves attention patterns and incorporate concepts in responses via attention to intervened tokens.
> In contrast, FSSV severely harms attention to instruction content tokens, hence poor utility preservation.
>
>
> ---
> References:\
> [R1] Refusal in Language Models Is Mediated by a Single Direction\
> [R2] PERSONA: Dynamic and Compositional Inference-Time Personality Control via Activation Vector Algebra\
> [R3] Improved Representation Steering for Language Models\
> [R4] AxBench: Steering LLMs? Even Simple Baselines Outperform Sparse Autoencoders\
> [R5] LongBench: A Bilingual, Multitask Benchmark for Long Context Understanding\
> [R6] Steering off Course: Reliability Challenges in Steering Language Models

---

> > ### Author Rebuttal · Reviewer_Bk6Z · 2026-04-02
> >
> > Thank you for your reply. I will keep the positive rating.

---

> > > ### Author Response · Authors · 2026-04-05
> > >
> > > Thank you for your response and for maintaining your positive assessment.
> > > We will incorporate the key improvements from the rebuttal, including
> > > clarification of PrOSV advantage over FSSV and its potential applicability to other steering paradigms,
> > > explanation for the rationale behind hyperparameter choice,
> > > clarification on long-context robustness of PrOSV,
> > > explanation on how PrOSV works via the attention mechanism,
> > > as well as improved writing and presentation.
> > >
> > > Again, we sincerely appreciate your support and your contributions to the review process.

---

### Official Review · Reviewer_92wR · 2026-03-09

**Soundness:** 3
**Presentation:** 2
**Significance:** 3
**Originality:** 3
**Overall Recommendation:** 4
**Confidence:** 4

**Summary:**

This paper addresses two challenges in steering vector methods for LLMs: the costly search for steering factors and the tradeoff between steering strength and general model utility. To address these issues, the paper proposes a principled framework that jointly learns steering directions and factors, along with a prompt-only intervention design (PrOSV) that applies steering to only a small set of prompt tokens. Results show that the proposed method improves over conventional factor-selection pipelines, while PrOSV often achieves a better balance between concept control and generation quality than full-sequence steering.

**Compliance With Llm Reviewing Policy:**

Affirmed.

**Final Justification:**

The response clarifies some issues. I update positive score.

**Key Questions For Authors:**

1. The intervention-location study is useful, but the search space is still fairly limited to prefix/suffix-based patterns. Since intervention location is a central design choice of PrOSV, why were more general token-selection strategies not considered, or to discuss whether p2+s2 is best only within this restricted design space?

2. The main empirical conclusions rely heavily on LLM-judge-based evaluation, yet the paper itself notes substantial variance in fair scores and sensitivity to generation length. How robust are the reported PrOSV gains to evaluation noise and judge preferences?

**Limitations:**

yes

**Strengths And Weaknesses:**

Strengths:
1. The paper provides a mathematical derivation utilizing scaling laws to address the instability issues typically encountered during the joint training of multiple factors.
2. The experimental setup is comprehensive and well-structured, providing solid empirical support for the proposed method.
3. The ablation study is comprehensive.


Weaknesses:
1. Although the paper evaluates long-context robustness, the results suggest that PrOSV is not consistently robust across model families, underperforming FSSV on Gemma2-2B/9B and only surpassing it on Qwen2.5-32B. This raises the question of whether prompt-only interventions remain reliably influential as generation length grows, and what mechanism explains the apparent model-scale dependence.

2. The motivation emphasizes reducing hyperparameter tuning burden, but the practical picture seems more nuanced. While joint training removes inference-time factor search, the appendix shows that it still requires nontrivial offline tuning over several hyperparameters, and the amortized benefit only materializes when enough SVs are trained.

3. The claim that p2+s2 is a model-agnostic choice seems somewhat strong given the current evidence. This conclusion is drawn from three models on relatively short prompts in CONCEPT10, while the long-context results later show clear model-dependent behavior. It would help if the paper clarified the intended scope of this claim or provided additional evidence.

---

> ### Author Rebuttal · Authors · 2026-03-28
>
> Thank you for the constructive feedback.
> We would like to clarify the concerns below and will revise the paper accordingly.
>
> > **W1**. Model-scale dependence of long-context robustness and mechanism.
>
> We hypothesize that the gap on Gemma2-2B/9B arises from train-test mismatch rather than an inherent limitation of PrOSV.
> SVs are trained on shorter prompts (17--20 tokens) but tested on tinyGSM8K (1K tokens), which is challenging for PrOSV.
>
> To test this, we train and evaluate on long prompts (2WikiMultihopQA split of LongBench [R3], ~5K tokens) on Gemma2-2B, precisely the regime where PrOSV underperformed.
> In this setting, **PrOSV outperforms FSSV (0.87 vs. 0.79)**, reversing the result from the short-training/long-testing setting.
> This shows PrOSV remains effective in long contexts when trained accordingly.
>
> Mechanistically, our tentative explanation for scale dependence is that larger models preserve prompt-conditioned features over long horizons more effectively, making PrOSV more robust even under mismatch;
> similar behavior has been noted by prior work [R4] and is not specific to PrOSV.
>
> We will revise Section 6.4 to present this as a hypothesis and frame the current long-context result as a stress test, while keeping our main claim focused on short-context steering, where PrOSV outperforms FSSV (Table 2).
>
> > **W2**. Reduced hyperparameter tuning claim and practical benefit.
>
> Thank you for prompting this clarification.
> Our claim is not zero tuning, but shifting tuning from per-SV inference-time search to one-off offline tuning;
> the benefit of joint training is *amortized efficiency*.
> In contrast, prior methods require per-SV inference-time factor selection, which scales linearly with the number of SVs.
> We show in Appendix F.2 that **after about 55 SVs, joint training becomes amortized-beneficial**; AxBench evaluates at least 100 concepts.
> We will make this framing more explicit in the main body.
>
> > **W3**. Clarify the scope of the "p2+s2 is model-agnostic" claim.
>
> We agree that the original phrasing is too strong.
> We will revise it to:
> "p2+s2 is the best-performing configuration within the prefix/suffix design space, which transfers consistently across the tested model families and scales."
>
> > **Q1**. Intervention location search space is limited to prefix/suffix.
>
> Our focus on prefix/suffix interventions is **principled, not merely a computational convenience**.
> These regions correspond to the two distinct functional roles:
> prefixes (especially BOS token) provide broad contextual modulation and are attended to by most tokens, while suffixes strong influence generation.
> Our attention map analysis of Figures 46--47 directly confirms this functional distinction.
> Therefore, restricting to prefix/suffix interventions allows us to study where to intervene under a fixed budget in a controlled manner.
>
> Empirically, p2+s2 outperforms p4 and s4 (in Table 1), confirming that combining prefix/suffix interventions is beneficial.
> We also observe *diminishing returns beyond 4 tokens*: p4+s4 < p2+s2.
> This indicates that our search space already captures dominant effects, and exhaustive combinatorial search is unlikely to yield large gains.
>
> We will clarify that p2+s2 is optimal within our search space, not globally optimal, and highlight more general intervention locations as future work.
>
> > **Q2**. Sensitivity to LLM-judge noise and evaluation variance.
>
> We address evaluation robustness from three angles.
>
> **(a) Noise magnitude**.
> Following [R1,R2], results are averaged over ≥100 concepts, substantially reducing variance.
> Table 13 shows standard errors of 0.017 (Gemma2-2B/9B) and 0.037 (Qwen2.5-32B) (<4% of the average score).
> Importantly, the performance gaps between PrOSV and baselines are generally larger than this noise level, indicating statistical significance.
>
> **(b) Removing a major variance source**.
> A key source of variance in prior SV methods is factor selection.
> The fair score protocol (see Appendix K.2) relies on validation/test splits for selecting factors, which introduces substantial variance due to random partitioning.
> Our joint training scheme removes factor selection entirely and directly evaluates on the test set with learned factors, eliminating this source of variance.
>
> **(c) Controlling judge variability**.
> We reduce judge noise via consistent AxBench setups and low temperature.
> While systematic judge preferences may remain, our conclusions rely on *relative rankings* between methods, not judge-invariant absolute scores.
> The rankings are stable across models, objectives and large concept sets, making it less likely that the gains of PrOSV are due to judge preferences alone.
>
>
> ---
> References:\
> [R1] Improved Representation Steering for Language Models\
> [R2] AxBench: Steering LLMs? Even Simple Baselines Outperform Sparse Autoencoders\
> [R3] LongBench: A Bilingual, Multitask Benchmark for Long Context Understanding\
> [R4] Steering off Course: Reliability Challenges in Steering Language Models

---

> > ### Author Rebuttal · Reviewer_92wR · 2026-04-01
> >
> > Thank you for the response, which clarified some issues. I have updated my scores accordingly.

---

> > > ### Author Response · Authors · 2026-04-05
> > >
> > > Thank you for your thoughtful follow-up and for taking the time to reassess our submission. We're glad that our clarifications were helpful and appreciate your updated evaluation.
> > > We will incorporate the key improvements from the rebuttal, including explanation and new results on model-scale dependence of long-context robustness, clarification of the practical benefit of one-off hyperparameter tuning, explanation on intervention location search space and LLM-judge noise/variance.
> > >
> > > Thank you again for your engagement and valuable feedback throughout the review process.

---

### Official Review · Reviewer_dwm1 · 2026-03-13

**Soundness:** 3
**Presentation:** 3
**Significance:** 3
**Originality:** 3
**Overall Recommendation:** 5
**Confidence:** 3

**Summary:**

In activation steering, the steering factor is selected in a costly procedure after the steering vector is generated. This is addressed through a new unified procedure based on scaling theory where both the steering vector and factor are learned together. Additionally, a new method is introduced to use steering vectors in prompt tokens only so as to have less of an effect on fluency and coherence.

**Compliance With Llm Reviewing Policy:**

Affirmed.

**Final Justification:**

Overall, I recommend acceptance -- I feel there is a clear motivation for integrating the factor selection into steering vector training, as well as intervening at less tokens with the prompt-only steering vectors. Additionally, there is sufficient motivation and experimental results to back this up. The rebuttal also clarified confusions and added more experiments regarding FSSV's and PrOSVs as well as long context experiments, which was enough to increase my score.

**Key Questions For Authors:**

See weaknesses above. Happy to increase my score if these are addressed.

**Limitations:**

yes

**Strengths And Weaknesses:**

Strengths
1. Clear motivation to learn the factor with the vector, as the factor selection procedure is expensive and not obtained through a continuous optimization procedure.
2. Extensive experiments with a variety of baselines and hyperparameter choices.

Weaknesses
1. The introduction of the joint training scheme introduces more hyperparameters, which can seemingly lead to expensive grid search style runs. A better discussion of the tradeoffs of this approach in terms of computational cost in the main body is missing, e.g., if you are producing only a certain number of steering vectors might it be worth it to just do the grid-search? Also, are $\beta$ and $\lambda$ and $\eta$’s the only hyperparameters that need searching? What about the intervention budget, which seems to have a less predictable effect (Section 6.2)?
    1. In practice, what values would be suggested for users to start with? What is the cost of the grid search here? Can you give more explanation as to why the choice of hyperparameters need only be done once then generalizes to any SV? Since it generalizes can I use the parameters you found best in the paper if I were to use the same models and layers?
2. For tinyGSM8k, I would be hesitant to say 1k is necessarily ‘long context’ since it is still relatively small; though there isn’t a fixed definition for what long context is, it has expanded as models have longer context windows, e.g., see [1] tests context as 32k up to 128k.
    - tinyGSM8k might also have a somewhat different distribution since it is full of few-shot examples, so may be off if we just steer the front and back and not take a more sophisticated approach? It is not necessarily a natural long context prompt like other datasets might be.
3. In Section 5 it is said that “FSSVs (…) cannot be trivially converted into PrOSVs” — this part is confusing to me. It seems to me that the difference between FSSV and PrOSVs are mainly in their intervention locations. Why can’t we take an existing steering vector generated through any method and apply it only on the prompt? Why is factor selection required during the attempt? There is some analysis in Appendix M.2 regarding direction similarity, but not the naive approach of intervening with the FSSV vector but only on the given tokens in the prompt. Though the result may be trivial it would be good to see this for completeness.

Suggestions
1. Can you provide a short description of the Lang and SimPO objectives in the main body to provide more context?
2. I think the dataset notation in Table 1 is a little confusing — it might seem like these datasets are interchangeable when they’re actually unique representing different models. This is specified in the superscript but still could be unclear.
3. In Section 6.3 Methods, can you clarify the baselines more? E.g., for prompting baseline add a little reference in parentheses like ‘adding concept elicitation information to the prompt’? Also, in Table 1 we have ‘Full prompt’ then in Table 2 we have ‘Prompt’, which might be a little confusing.
4. In general when referring to FSSV’s by default they are for fixed unit factor? But an also be used for joint training? Why does SimPO in Table 2 not have fixed unit factor for FSSV? In general, it might be helpful to define if by default, when referenced, FSSV are taken using the joint objective or the factor selection procedure.

[1] Bai, Yushi, et al. "Longbench v2: Towards deeper understanding and reasoning on realistic long-context multitasks." Proceedings of the 63rd Annual Meeting of the Association for Computational Linguistics (Volume 1: Long Papers). 2025.

---

> ### Author Rebuttal · Authors · 2026-03-28
>
> We sincerely thank you for your acknowledgement of our work and valuable feedback.
> We would like to address the raised concerns as follows.
>
> > **W1.1**. Hyperparameter complexity; missing discussion of computational cost tradeoffs.
>
> Our key point is that joint training changes the structure of tuning rather than eliminating it.
> Prior approaches require per-SV inference-time factor selection, which scales linearly with the number of SV instances.
> In contrast, joint training performs **one-time offline tuning** of a set of global hyperparameters, after which all SVs can be trained and used without further search.
> Although we have discussed hyperparameter tuning costs in Appendix F.2, we will clarify the tradeoff in the main body in the revision.
>
> > **W1.2**. Unclear full hyperparameter set.
>
> Four hyperparameters need tuning: factor/direction initialization sizes and learning rates (details in Appendix F.2).
> Intervention budget generally does not need tuning, since our preliminary analysis present p2+s2 as a decent choice that transfers well across model families and scales.
>
> > **W1.3**. Lack of practical guidance.
>
> **Cost of grid search.**
> We discuss hyperparameter search in Appendix F.2 in detail, and we briefly explain the cost here for your convenience.
> As is shown in Table 10, the grid search for AxBench takes 768 training runs when we evaluate a total of 1100 concepts.
>
> **Suggested choices.**
> In practice, we recommend the following default settings:
> - Roughly tune $\lambda\in\\{1,8\\}$;
> - Fix $\eta_v$ in a stable mid-range (e.g., 0.01--0.08);
> - Tune $\beta,\eta_\alpha$ for steering factors carefully as we do in the paper;
> - Prioritize prefix+suffix interventions and start from p2+s2.
>
> We will add a dedicated section in the revision.
>
> > **W1.4**. Why hyperparameter tuning generalizes to any SV?
>
> The tuned hyperparameters depend on properties of the target model/layer (e.g., gradient scale, width) to keep SV training in a stable and efficient regime.
> These properties are shared across SVs trained on that same model/layer.
>
> > **W1.5**. Can I use the parameters you found best if I were to use the same models and layers?
>
> Yes, and we recommend that you start with our configuration.
>
> > **W2**. 1K tokens not convincing as long context; tinyGSM8k distribution not representative of natural long contexts.
>
> We agree that 1K tokens may not meet modern definitions of "long context."
> We will revise the wording to "extended context" and clarify the scope.
>
> We also acknowledge that tinyGSM8K is a different distribution from natural prompts.
> We will clarify that it represents a specific type of extended-context prompt and that more natural settings would be a direction for future work.
>
> **Additional evidence.**
> To address both prompt length and distribution, we include results on LongBench [R1].
> When trained *and* tested on the 2WikiMultihopQA split (~5K tokens, 5 concepts of Concept10, Gemma2-2B), PrOSV no longer underperforms FSSV (overall scores **0.87 vs. 0.79**, both Lang. objective).
> These results suggest that PrOSV *can* work well under extended contexts on smaller model when also trained on extended contexts.
>
> > **W3**. Non-trivial convertibility between FSSV and PrOSV
>
> While it is possible to apply an FSSV direction only to prompt tokens, this typically leads to suboptimal performance.
> The **key reasons** are that
> - FSSV and PrOSV have different effective directions (Figure 48 shows low similarity between PrOSV/FSSV directions);
> - Using FSSV directions as PrOSV would require additional factor selection (Figure 41 shows PrOSVs need higher factors than FSSVs).
>
> We present a **new experiment** showing that FSSVs fail to serve as effective PrOSVs.
> We test SVs at 10th layer of Gemma2-2B on Concept10, and find that reusing FSSV as PrOSV yields overall scores of 0.17 after factor tuning and 0.04 with FSSV factors, when PrOSV yields a score of 0.70.
> We will add a detailed discussion regarding FSSV/PrOSV convertibility in the revision.
>
> > **S1--S3**. Brief explanation of Lang / SimPO objectives; dataset notation in Table 1; baseline descriptions.
>
> Thank you for the suggestions! We will:
> - move description of Lang./SimPO from Appendix E.4 to the main body;
> - change from dataset notation to model/layer-specific notation like "2B;L10";
> - explain the baselines more clearly.
>
> > **S4.1**. Ambiguity in default definition of FSSV.
>
> We are sorry for the confusion.
> FSSV describes intervention location which is orthogonal to choice of factors.
> By default (except for Section 6.3), by "FSSV" we refer to FSSVs trained with our joint training scheme.
>
> > **S4.2**. Why does SimPO in Table 2 not have fixed unit factor for FSSV?
>
> We only show FSSV (Lang.) with unit factor in Table 2 since these results are taken from prior work, and we did not test SimPO objective when FSSV (Lang.) already underperforms.
>
> ---
> References:\
> [R1] LongBench: A Bilingual, Multitask Benchmark for Long Context Understanding

---

> > ### Author Rebuttal · Reviewer_dwm1 · 2026-04-03
> >
> > Thanks for the clarifications and additional experiments -- I will raise my score to a 5.

---

> > > ### Author Response · Authors · 2026-04-05
> > >
> > > Thank you for your positive feedback and for taking the time to consider our rebuttal. We're glad that the additional clarifications and experiments helped address your concerns, and we sincerely appreciate your decision to raise the score.
> > > We will incorporate the key improvements from the rebuttal, including but not limited to: clarification of computational cost tradeoffs, practical guidance on grid search, clarification of experiment scope, baseline formulation and improved writing.
> > >
> > > Thank you again for your thoughtful engagement and valuable contributions to the review process.

---

### Official Review · Reviewer_yLYF · 2026-03-13

**Soundness:** 3
**Presentation:** 3
**Significance:** 3
**Originality:** 3
**Overall Recommendation:** 5
**Confidence:** 3

**Summary:**

The authors identify two drawbacks of standard fine-tuned steering vectors: (1) steering strength or steering factors need to be carefully selected (2) steering vectors naturally compress and operate across a full sequence (i.e. there exists a single vector for a sequence). The authors propose a method to jointly train steering factors and directions to tackle (1) and propose a prompt-only steering vector that tackles (2). Taken together, their work shows their approach outperforms full sequence steering vectors on AxBench.

**Compliance With Llm Reviewing Policy:**

Affirmed.

**Final Justification:**

I think the paper is a small focused contribution without any major weaknesses. Without a large expansion of models, datasets, and analysis its impossible to vault this paper to get a very strong rating of 6, so I maintain my score of a 5. The presentation and flow related challenges that I mentioned given the author response will likely be cleared up for a camera ready, so I vote for the paper to be accepted (overall assessment 5).

**Key Questions For Authors:**

- How does prefix tuning relate to PrOSV? To me this is quite relevant and probably worth citing.
- Can PrOSV be thought of as a task vector or function vector? If so, can you do auxiliary evaluations that demonstrate effectiveness at steering a model to do better at a specific task?
- I'm confused about the long context analysis (see weakness 2).

**Limitations:**

yes

**Strengths And Weaknesses:**

Strengths:
- Introduction and motivation is well-written and clear.
- The appendix is incredibly thorough; especially about how different steering methods relate to one another including recent approaches on dynamic steering vectors and KV cache steering.
- There are many different ablations shown in the appendix.
- For all intents and purposes PrOSV is an improvement over full sequence steering vectors and other approaches, getting state of the art on AxBench.

Weaknesses:
- Expansion to stronger models and more model families would strengthen the work.
- The long-context analysis seems incomplete. I don't understand why you can conclude this in section 6.4 under insights. I'm putting this under weaknesses for now.
- The presentation of the results could be improved. The heatmaps are difficult to parse. I wonder if further analysis could be done on layer number
- Section 4 is very hard to parse and difficult to follow.

---

> ### Author Rebuttal · Authors · 2026-03-28
>
> We sincerely thank you for your acknowledgement and constructive suggestions.
> We would like to address the concerns as follows.
>
> > **W1**. Expansion to stronger models and more model families would strengthen the work.
>
> We agree that extending experiments to larger models and more model families would strengthen our work.
> Due to computational constraints, we focused on a representative set spanning different scales and architectures.
> We will clarify this limitation and position our results as a strong initial validation, with broader coverage as future work.
>
> > **W2 & Q3**. Incomplete long-context analysis causes confusion.
>
> Our analysis of SVs in long-context setting is motivated by a hypothesis:
> since PrOSV only intervenes on a few prompt tokens, it might fail to generalize to long prompts when only trained on short prompts.
> In our current paper, we take PrOSVs from Section 6.3 trained on AxBench (a short-prompt setting, ~20 tokens) and test it on tinyGSM8K (a long-prompt setting, ~1K tokens).
> Table 3 shows that although PrOSV underperforms FSSV on Gemma2-2B/9B, it surprisingly outperforms FSSV on Qwen2.5-32B.
>
> Mechanistically, our tentative explanation for the scale dependence under train-test mismatch is that larger models better fulfill prompts over long horizons, allowing PrOSV to remain influential even under mismatch.
> Similar phenomena have been noted by prior work on steering LLMs with FSSVs [R1] and are not specific to PrOSV.
>
> To make long-context analysis more complete, we conduct an **additional experiment**.
> We train *and* evaluate SVs on longer prompts (~5K tokens) using the 2WikiMultihopQA split of LongBench [R2], with 5 concepts from Concept10 on Gemma2-2B, precisely the small-model regime where PrOSV underperformed in Table 3.
> PrOSV now outperforms FSSV, reversing the result from the short-training, long-testing setting.
> Therefore, the scale-dependent gap in Table 3 is a data coverage issue, which could be resolved by training on longer prompts.
>
> |Method|Overall score|
> |-|:-:|
> |Prompt|1.19|
> |PrOSV|0.87|
> |FSSV|0.79|
>
> We will revise Section 6.4 to clearly separate empirical observations from interpretation, to present our hypothesis rather than a conclusion and to explicitly discuss the limitations and model-scale dependence.
>
> > **W3.1 & W4**. Readability of Section 4 and presentation of Figure 2 heatmap.
>
> We will focus on improving the reader-friendliness of theoretical results and readability of empirical results.
> Specifically, we will further explain the theoretical results of Section 4 with plain language and elaborate on practical implications in addition to current Section 4.4.
> We will also add visual aids to Figure 2 pointing out which hyperparameter setups support our theoretical results.
>
> > **W3.2**. I wonder if further analysis could be done on layer number.
>
> In this paper, we intervene on layers where previous works have found to be effective for FSSV steering, e.g., 10th layer of Gemma2-2B.
> We find that both ProSV and FSSV are sensitive to layers.
> Specifically, PrOSV (p2+s2, Lang.) yields an overall score of 0.679 on Concept10 at 10th layer of Gemma2-2B but drops to 0.374 at 20th layer;
> FSSV (Lang.) yields an overall score of 0.648 at 10th layer but 0.471 at 20th layer.
>
> > **Q1**. How does prefix tuning relate to PrOSV?
>
> We agree that the connection to prefix tuning is relevant and worth discussing.
> We will incorporate a dedicated paragraph in the Related Work section.
>
> In terms of *similarity*, both PrOSV and prefix tuning steer frozen LMs via optimizable vectors.
> The *difference* between the two lies in how the vectors interact with model representations.
> In prefix tuning, the optimized vectors *condition* prompt representations and are prepended to original attention keys and values.
> In contrast, PrOSV *directly edit* prompt representations by adding vectors onto prompt representations.
>
> > **Q2**. Can PrOSV be thought of as a task vector or function vector?
>
> Good point!
> PrOSV could indeed be used as a function vector [R3].
> However, the task of function vector is different from the task we aim to solve in this paper.
> As is explained at lines 125--135, we study the task of concept-based steering, which aims for the dual goal of instruction fulfillment and concept incorporation.
> In contrast, the function vector is only intended for the target task.
>
> In response to your interest, we steer Llama-7B to answer BoolQ questions, where we train PrOSV (p2+s2) at all layers.
> FSSV and LoRA (rank 8) are also tested as baselines.
> Results show that PrOSV outperforms FSSV but underperforms LoRA, which is expected since PrOSV is not designed for task adaptation.
>
> |Method|BoolQ|
> |-|:-:|
> |Vanilla|38|
> |FSSV|56|
> |PrOSV|63|
> |LoRA|68.9|
>
> ---
> References:\
> [R1] Steering off Course: Reliability Challenges in Steering Language Models\
> [R2] LongBench: A Bilingual, Multitask Benchmark for Long Context Understanding\
> [R3] Function vectors in large language models

---

> > ### Author Rebuttal · Reviewer_yLYF · 2026-04-01
> >
> > Thanks for running the small studies you did and clarifying some of my questions. I think the paper is more or less fine; my original assessment was a 5 and I'll maintain that score. Without a large revision incorporating more models from different model families at different scales, careful analysis on more axes, and additional explanation contextualizing with respect to other approaches (e.g. the prefix tuning and function vector relationships discussed in the author response are a nice start), I can't increase my score further.

---

> > > ### Author Response · Authors · 2026-04-05
> > >
> > > Thank you for your thoughtful follow-up and for maintaining your assessment.
> > >
> > > We appreciate your recognition of our additional experiments and clarifications. In the revision, we will incorporate the key improvements from the rebuttal, including: clearer long-context analysis, integration of the new experimental results, improved writing and figure presentation, and better contextualization with related approaches such as prefix tuning and function vectors.
> > >
> > > We also agree that extending the evaluation to more models, scales, and model families, along with deeper comparative analysis, would further strengthen the work. We will highlight these as important directions for future work in the discussion.
> > >
> > > Thank you again for your constructive feedback and engagement throughout the review process.

---

### Decision · Program_Chairs · 2026-04-30

**Decision:**

Accept (regular)

**Comment:**

Overview of the paper:

This paper addresses two primary challenges associated with steering vectors in large language models: the computational overhead of per-vector factor selection and the degradation of generation quality from full-sequence interventions. To resolve these issues, the authors propose an approach that jointly learns steering directions and factors, guided by neural network scaling theory. Additionally, they introduce Prompt-Only Steering Vectors (PrOSV), a method that restricts interventions to a subset of prompt tokens rather than the entire sequence. Experimental evaluations on the AxBench benchmark indicate that this approach balances concept control and model utility more efficiently than conventional full-sequence methods.

Strengths:

- The problem is well-motivated, and the proposed joint training mechanism is supported by a clear theoretical derivation utilizing scaling laws.
- The experimental setup is systematic and comprehensive, featuring a variety of baselines, robust ablation studies, and evaluation across multiple models.
- Empirical results demonstrate that the prompt-only steering approach successfully reduces unintended harms to model fluency, achieving a favorable tradeoff between concept incorporation and general capability preservation.
- The supplementary material is thorough, providing details on hyperparameter effects and contextualizing the approach alongside related dynamic steering and KV cache methods.

Areas for improvement:

- Although the joint training scheme eliminates inference-time factor search, it still necessitates nontrivial offline tuning for multiple global hyperparameters (e.g., initialization sizes and learning rates).
- The design space for the prompt-only interventions is primarily restricted to prefix and suffix token patterns, leaving room for future exploration into more generalized token-selection strategies.
- Reviewers noted that several visual and presentational elements, such as the interpretability of initial diagrams, the clarity of the heatmaps, and the readability of the theoretical sections, required structural refinement to improve reader comprehension.